# CLOOB: MODERN HOPFIELD NETWORKS WITH INFOLOOB OUTPERFORM CLIP

## ABSTRACT

Contrastive learning with the InfoNCE objective is exceptionally successful in various self-supervised learning tasks. Recently, the CLIP model yielded impressive results on zero-shot transfer learning when using InfoNCE for learning visual representations from natural language supervision. However, InfoNCE as a lower bound on the mutual information has been shown to perform poorly for high mutual information. In contrast, the InfoLOOB upper bound (leave one out bound) works well for high mutual information but suffers from large variance and instabilities. We introduce "Contrastive Leave One Out Boost" (CLOOB), where modern Hopfield networks boost learning with the InfoLOOB objective. Modern Hopfield networks replace the original embeddings by retrieved embeddings in the InfoLOOB objective. The retrieved embeddings give InfoLOOB two assets. Firstly, the retrieved embeddings stabilize InfoLOOB, since they are less noisy and more similar to one another than the original embeddings. Secondly, they are enriched by correlations, since the covariance structure of embeddings is reinforced through retrievals. We compare CLOOB to CLIP after learning on the Conceptual Captions and the YFCC dataset with respect to their zero-shot transfer learning performance on other datasets. CLOOB consistently outperforms CLIP at zero-shot transfer learning across all considered architectures and datasets.

## 1 INTRODUCTION

With the advent of large corpora of unlabeled data in vision and language, self-supervised learning via contrastive learning has become highly successful. Some contrastive learning objectives, such as those of BYOL (Grill et al., 2020) and SimSiam (Chen & He, 2021), do not require negative samples. However, the most popular objective for contrastive learning is InfoNCE (van den Oord et al., 2018), in which for an anchor sample, a positive sample is contrasted with negative samples.

The idea to use objectives with negative samples is well known in deep learning (Gutmann & Hyvärinen, 2010; Chen et al., 2017; Mikolov et al., 2013). For contrastive learning, the most successful objective is InfoNCE, which has been introduced as Contrastive Predictive Coding (CPC) (van den Oord et al., 2018). InfoNCE has been applied to transfer learning (Hénaff et al., 2019), to natural language response suggestion (Henderson et al., 2017), to learning sentence representations from unlabelled data (Logeswaran & Lee, 2018), and to unsupervised feature learning by maximizing distinctions between instances (Wu et al., 2018). InfoNCE has been used for learning visual representations in Pretext-Invariant Representation Learning (PIRL) (Misra & vanDerMaaten, 2020), in Momentum Contrast (MoCo) (He et al., 2020), and in SimCLR (Chen et al., 2020). SimCLR became well known as is was highly effective for transfer learning. Zero-shot transfer learning (Lampert et al., 2009) is one of the most ambitious goals in vision, since it would improve various real-world downstream applications. Current models in natural language processing and vision perform very well on standard benchmarks, but they fail at new data, new applications, deployments in the wild, and stress tests (D'Amour et al., 2020; Recht et al., 2019; Taori et al., 2020; Lapuschkin et al., 2019; Geirhos et al., 2020). A model with high zero-shot transfer learning performance will not fail on such data, therefore will be trusted by practitioners.

Contrastive Language-Image Pre-training (CLIP) based on the InfoNCE objective yielded very impressive results at zero-shot transfer learning (Radford et al., 2021). CLIP learns expressive image embeddings directly from raw text, thereby leverages a much richer source of supervision than just

labels. A plethora of CLIP follow-up work has already been published (see Appendix Section A.5). The CLIP model is considered as an important foundation model (Bommasani et al., 2021). Though CLIP excels at zero-shot transfer learning, it can be improved.

CLIP training suffers from an "explaining away" problem (Wellman & Henrion, 1993), which leads to "shortcut learning" (Geirhos et al., 2020) or the Clever Hans phenomenon (Lapuschkin et al., 2019). Explaining away impedes the increase of the similarity between a text and a corresponding image, since learning focuses on only one common aspect and does not exploit the full covariance structure of the data. If one common aspect is sufficient for high similarity, the InfoNCE objective saturates, since it has the form $a/(a + b)$ with $a$ giving the similarity of a matched pair and $b$ giving the average similarity of unmatched pairs. For a large similarity $a$, the objective saturates and increasing $a$ has a small effect. Contrary to InfoNCE, the leave-one-out ("InfoLOOB") bound (Poole et al., 2019) is of the form $a/b$ which does not saturate. However, so far the InfoLOOB bound was not used as an objective in contrastive learning. We justify the maximization of the InfoLOOB bound for contrastive learning in Appendix Section A.1.3. We show that maximizing the InfoLOOB bound leads to a good approximation of the mutual information, in particular for high mutual information. A problem of InfoLOOB is that it has high variance for small $b$.

Even when InfoLOOB avoids saturation, CLIP insufficiently extracts the covariance structure in the data. The covariance originates from co-occurrences of related words in text or from co-occurrences of objects, textures, or colors in images. CLIP's problem of insufficiently extracting the covariance structure of the data is tackled by modern Hopfield networks. Hopfield networks are energy-based, binary associative memories, which popularized artificial neural networks in the 1980s (Hopfield, 1982; 1984). Associative memory networks have been designed to store and retrieve samples. Their storage capacity can be considerably increased by polynomial terms in the energy function (Chen et al., 1986; Psaltis & Cheol, 1986; Baldi & Venkatesh, 1987; Gardner, 1987; Abbott & Arian, 1987; Horn & Usher, 1988; Caputo & Niemann, 2002; Krotov & Hopfield, 2016). In contrast to these binary memory networks, we use continuous associative memory networks with very high storage capacity. These modern Hopfield networks for deep learning architectures have an energy function with continuous states and can retrieve samples with only one update (Ramsauer et al., 2021; 2020). Modern Hopfield Networks have already been successfully applied to immune repertoire classification (Widrich et al., 2020) and chemical reaction prediction (Seidl et al., 2021). Modern Hopfield networks reinforce the covariance structure in the data and stabilize the InfoLOOB objective by increasing $b$. The covariance structure of retrieved embeddings is amplified through co-occurrences of embedding features in the memory. Additionally, the retrieved embeddings are less noisy and more similar to one another which leads to a larger $b$. We introduce "Contrastive Leave One Out Boost" (CLOOB) which overcomes CLIP's problems of (i) "explaining away" with saturation and (ii) insufficiently extracting the covariance structure of the data. CLOOB uses the leave-one-out ("InfoLOOB") bound (Poole et al., 2019) as the objective in combination with modern Hopfield networks.

Our contributions are:
(a) we introduce a new contrastive learning method called CLOOB,
(b) we propose InfoLOOB as an objective for contrastive learning,
(c) we propose to use modern Hopfield networks to reinforce covariance structures,
(d) we show theoretical properties of the InfoLOOB objective and loss function.

## 2  INFOLOOB VS. INFONCE

We discuss and analyse known bounds on the mutual information $I(X ; Y)$ between random variables $X$ and $Y$, which are distributed according to $p(\boldsymbol{x}, \boldsymbol{y})$:

$$I(X ; Y) = \mathrm{E}_{p(\boldsymbol{x},\boldsymbol{y})}\left[\ln \frac{p(\boldsymbol{x}, \boldsymbol{y})}{p(\boldsymbol{x})\, p(\boldsymbol{y})}\right] = \mathrm{E}_{p(\boldsymbol{x},\boldsymbol{y})}\left[\ln \frac{p(\boldsymbol{x} \mid \boldsymbol{y})}{p(\boldsymbol{x})}\right] = \mathrm{E}_{p(\boldsymbol{x},\boldsymbol{y})}\left[\ln \frac{p(\boldsymbol{y} \mid \boldsymbol{x})}{p(\boldsymbol{y})}\right] . \quad (1)$$

We consider the multi-sample lower bound "InfoNCE" (van den Oord et al., 2018). A pair of an anchor sample $\boldsymbol{y}$ and a positive sample $\boldsymbol{x}_1$ is drawn via the joint distribution $p(\boldsymbol{x}_1, \boldsymbol{y})$. The negative samples $\tilde{X} = \{\boldsymbol{x}_2, \ldots, \boldsymbol{x}_N\}$ are drawn iid according to the marginal distribution $p(\boldsymbol{x})$. Using $X = \{\boldsymbol{x}_1, \boldsymbol{x}_2, \ldots, \boldsymbol{x}_N\}$, the probabilities of the datasets are $p(\tilde{X}) = \prod_{i=2}^{N} p(\boldsymbol{x}_i)$, $p(X \mid \boldsymbol{y}) =$

$p(\boldsymbol{x}_1 \mid \boldsymbol{y}) \prod_{i=2}^{N} p(\boldsymbol{x}_i)$, and $p(X) = \prod_{i=1}^{N} p(\boldsymbol{x}_i)$. The InfoNCE with score function $f(\boldsymbol{x}, \boldsymbol{y})$ is

$$\mathrm{I}_{\mathrm{InfoNCE}}(X_1 \; ; \; Y) \;=\; \mathrm{E}_{p(\boldsymbol{y})}\left[\mathrm{E}_{p(X|\boldsymbol{y})}\left[\ln\left(\frac{f(\boldsymbol{x}_1, \boldsymbol{y})}{\frac{1}{N}\sum_{i=1}^{N} f(\boldsymbol{x}_i, \boldsymbol{y})}\right)\right]\right] \;, \tag{2}$$

using the factor $1/N$ as in Poole et al. (2019); Tschannen et al. (2019); Cheng et al. (2020); Chen et al. (2021). For $f(\boldsymbol{x}, \boldsymbol{y}) = p(\boldsymbol{y} \mid \boldsymbol{x})$, we obtain the InfoNCE with probabilities. The InfoNCE is a lower bound on the mutual information (Poole et al., 2019), which is stated in the next theorem.

**Theorem 1** (InfoNCE lower bound). *InfoNCE with score function $f(\boldsymbol{x}, \boldsymbol{y})$ is a lower bound on the mutual information:*

$$\mathrm{I}(X_1 \; ; \; Y) \;\geq\; \mathrm{E}_{p(\boldsymbol{y})}\left[\mathrm{E}_{p(X|\boldsymbol{y})}\left[\ln\left(\frac{f(\boldsymbol{x}_1, \boldsymbol{y})}{\frac{1}{N}\sum_{i=1}^{N} f(\boldsymbol{x}_i, \boldsymbol{y})}\right)\right]\right] \;=\; \mathrm{I}_{\mathrm{InfoNCE}}(X_1 \; ; \; Y) \,. \tag{3}$$

*In particular, the bound holds for InfoNCE with probabilities, i.e. for $f(\boldsymbol{x}, \boldsymbol{y}) = p(\boldsymbol{y} \mid \boldsymbol{x})$.*

For a proof see Poole et al. (2019) and the proof of Theorem A1 in the Appendix.

The "Leave one out upper bound" (Poole et al., 2019) on the mutual information was called "L1Out" in Cheng et al. (2020), while we call it "InfoLOOB" (LOOB for "Leave One Out Bound"). InfoLOOB is the same as InfoNCE (Eq. (3)), but without the positive sample $x_1$ in the denominator. Contrastive Log-ratio Upper Bound (CLUB), another upper bound on the mutual information, was only used for minimizing it (Cheng et al., 2020). Maximizing CLUB failed in experiments, because the embedding distribution was not uniform as known for similar objectives (Wang & Liu, 2021). Uniform embedding distributions are required for successful contrastive learning (Wang & Isola, 2020).

We use InfoLOOB as an objective, since it approximates high mutual information better than InfoNCE. Maximizing an upper bound on the mutual information might be counter-intuitive. Therefore, we justify the maximization of the InfoLOOB bound for contrastive learning in Appendix Section A.1.3. We show that maximizing the InfoLOOB bound approximates the mutual information, the better the higher it is. Recently, InfoLOOB was independently introduced for and successfully applied to image-to-image contrastive learning (Yeh et al., 2021).

The InfoLOOB with score function $f(\boldsymbol{x}, \boldsymbol{y})$ is defined in the following, where we obtain the InfoLOOB with probabilities for $f(\boldsymbol{x}, \boldsymbol{y}) = p(\boldsymbol{y} \mid \boldsymbol{x})$:

$$\mathrm{I}_{\mathrm{InfoLOOB}}(X_1 \; ; \; Y) \;=\; \mathrm{E}_{p(\boldsymbol{y})}\left[\mathrm{E}_{\tilde{p}(X|\boldsymbol{y})}\left[\ln\left(\frac{f(\boldsymbol{x}_1, \boldsymbol{y})}{\frac{1}{N-1}\sum_{i=2}^{N} f(\boldsymbol{x}_i, \boldsymbol{y})}\right)\right]\right] \,. \tag{4}$$

Before we show that InfoLOOB with a score function is an upper bound on the mutual information, we need some definitions. $\tilde{p}(\boldsymbol{x} \mid \boldsymbol{y})$ draws the positives for $\boldsymbol{y}$ with lower probability than $p(\boldsymbol{x})$, that is, the positives are under-sampled. $Z(\boldsymbol{y}) = \mathrm{E}_{\tilde{p}(\boldsymbol{x}|\boldsymbol{y})}[f(\boldsymbol{x}, \boldsymbol{y})]$ gives the average score $f(\boldsymbol{x}, \boldsymbol{y})$, if under-sampling via $\tilde{p}(\boldsymbol{x} \mid \boldsymbol{y})$, while $Z^*(\boldsymbol{y}) = \mathrm{E}_{p(\boldsymbol{x})}[f(\boldsymbol{x}, \boldsymbol{y})]$ average score $f(\boldsymbol{x}, \boldsymbol{y})$ if sampling from $p(\boldsymbol{x})$. We define the variational distribution $q(\boldsymbol{x} \mid \boldsymbol{y}) = \frac{p(\boldsymbol{x})f(\boldsymbol{x},\boldsymbol{y})}{Z^*(\boldsymbol{y})}$. Our main assumption is expressed by the log-ratio of the averages $Z(\boldsymbol{y})$ and $Z^*(\boldsymbol{y})$:

$$\mathrm{E}_{p(\boldsymbol{y})}\left[\mathrm{KL}(p(\boldsymbol{x} \mid \boldsymbol{y}) \parallel q(\boldsymbol{x} \mid \boldsymbol{y}))\right] \;\leqslant\; \mathrm{E}_{p(\boldsymbol{y})}\left[\ln Z^*(\boldsymbol{y}) \;-\; \ln Z(\boldsymbol{y})\right] \,, \tag{5}$$

which ensures that the positives $\boldsymbol{x}$ are sufficiently under-sampled via $p(\boldsymbol{x} \mid \boldsymbol{y})$. The Kullback-Leibler divergence gives the minimal difference between averaging $f(\boldsymbol{x}, \boldsymbol{y})$ via $p(\boldsymbol{x})$ and via $\tilde{p}(\boldsymbol{x} \mid \boldsymbol{y})$. The next theorem shows that InfoLOOB is an upper bound on the mutual information.

**Theorem 2** (InfoLOOB upper bound). *If $\tilde{X} = \{\boldsymbol{x}_2, \ldots, \boldsymbol{x}_N\}$ are drawn iid according to $\tilde{p}(\boldsymbol{x} \mid \boldsymbol{y})$ and if the main assumption Eq. (5) holds, then InfoLOOB with score function $f(\boldsymbol{x}, \boldsymbol{y})$ is an upper bound on the mutual information:*

$$\mathrm{I}(X_1 \; ; \; Y) \;\leqslant\; \mathrm{E}_{p(\boldsymbol{y})}\left[\mathrm{E}_{\tilde{p}(X|\boldsymbol{y})}\left[\ln\left(\frac{f(\boldsymbol{x}_1, \boldsymbol{y})}{\frac{1}{N-1}\sum_{i=2}^{N} f(\boldsymbol{x}_i, \boldsymbol{y})}\right)\right]\right] \;=\; \mathrm{I}_{\mathrm{InfoLOOB}}(X_1 \; ; \; Y) \,. \tag{6}$$

*The bound is valid for InfoLOOB with probabilities (without under-sampling), where the negative samples $\tilde{X} = \{\boldsymbol{x}_2, \ldots, \boldsymbol{x}_N\}$ are drawn iid according to $p(\boldsymbol{x})$ and $f(\boldsymbol{x}, \boldsymbol{y}) = p(\boldsymbol{y} \mid \boldsymbol{x})$.*

The proof for this theorem is given as proof for Theorem A2 in the Appendix.

**Loss functions and their gradients.** The training set $\{(\boldsymbol{x}_1, \boldsymbol{y}_1), (\boldsymbol{x}_2, \boldsymbol{y}_2), \ldots, (\boldsymbol{x}_N, \boldsymbol{y}_N)\}$ consists of $N$ samples that are drawn iid from $p(\boldsymbol{x}, \boldsymbol{y})$. InfoNCE uses the matrix $\boldsymbol{X} = (\boldsymbol{x}_1, \ldots, \boldsymbol{x}_N)$, while InfoLOOB uses $\tilde{\boldsymbol{X}} = (\boldsymbol{x}_2, \ldots, \boldsymbol{x}_N)$. The matrices differ by the positive sample $\boldsymbol{x}_1$. For the score function $f(\boldsymbol{x}, \boldsymbol{y})$, we use $f(\boldsymbol{x}, \boldsymbol{y}) = \exp(\tau^{-1}\text{sim}(\boldsymbol{x}, \boldsymbol{y}))$ with the similarity $\text{sim}(\boldsymbol{x}, \boldsymbol{y}) = \boldsymbol{y}^T \boldsymbol{x}$ and $\tau$ as the temperature. We have the InfoNCE and InfoLOOB loss functions:

$$\text{L}_{\text{InfoNCE}} = -\frac{1}{N} \sum_{i=1}^{N} \ln \frac{\exp(\tau^{-1} \boldsymbol{x}_i^T \boldsymbol{y}_i)}{\sum_{j=1}^{N} \exp(\tau^{-1} \boldsymbol{x}_i^T \boldsymbol{y}_j)} - \frac{1}{N} \sum_{i=1}^{N} \ln \frac{\exp(\tau^{-1} \boldsymbol{x}_i^T \boldsymbol{y}_i)}{\sum_{j=1}^{N} \exp(\tau^{-1} \boldsymbol{x}_j^T \boldsymbol{y}_i)} , \quad (7)$$

$$\text{L}_{\text{InfoLOOB}} = -\frac{1}{N} \sum_{i=1}^{N} \ln \frac{\exp(\tau^{-1} \boldsymbol{x}_i^T \boldsymbol{y}_i)}{\sum_{j \neq i} \exp(\tau^{-1} \boldsymbol{x}_i^T \boldsymbol{y}_j)} - \frac{1}{N} \sum_{i=1}^{N} \ln \frac{\exp(\tau^{-1} \boldsymbol{x}_i^T \boldsymbol{y}_i)}{\sum_{j \neq i} \exp(\tau^{-1} \boldsymbol{x}_j^T \boldsymbol{y}_i)} . \quad (8)$$

In the second sum of the losses in Eq. 7 and Eq. 8, we consider only the first term. For simplicity, we abbreviate $\boldsymbol{y} = \boldsymbol{y}_1$ leading to the pair $(\boldsymbol{x}_1, \boldsymbol{y})$ and the negatives $\tilde{\boldsymbol{X}} = (\boldsymbol{x}_2, \ldots, \boldsymbol{x}_N)$.

$$\text{L}_{\text{InfoNCE}}(\boldsymbol{y}) = -\ln \frac{\exp(\tau^{-1} \boldsymbol{x}_1^T \boldsymbol{y})}{\sum_{j=1}^{N} \exp(\tau^{-1} \boldsymbol{x}_j^T \boldsymbol{y})} , \quad \text{L}_{\text{InfoLOOB}}(\boldsymbol{y}) = -\ln \frac{\exp(\tau^{-1} \boldsymbol{x}_1^T \boldsymbol{y})}{\sum_{j=2}^{N} \exp(\tau^{-1} \boldsymbol{x}_j^T \boldsymbol{y})} .$$

These loss terms can be simplified to $\text{L}_{\text{InfoNCE}}(\boldsymbol{y}) = -\tau^{-1}\boldsymbol{y}^T\boldsymbol{x}_1 + \tau^{-1}\text{lse}(\tau^{-1}, \boldsymbol{X}^T\boldsymbol{y})$ and $\text{L}_{\text{InfoLOOB}}(\boldsymbol{y}) = -\tau^{-1}\boldsymbol{y}^T\boldsymbol{x}_1 + \tau^{-1}\text{lse}(\tau^{-1}, \tilde{\boldsymbol{X}}^T\boldsymbol{y})$, where lse is the log-sum-exp function (see Eq. (A103) in the Appendix). The gradient of the InfoNCE loss with respect to $\boldsymbol{y}$ is $-\tau^{-1}\boldsymbol{x}_1 + \tau^{-1}\boldsymbol{X}\text{softmax}(\tau^{-1}\boldsymbol{X}^T\boldsymbol{y})$ and the gradient of the InfoLOOB loss is $-\tau^{-1}\boldsymbol{x}_1 + \tau^{-1}\tilde{\boldsymbol{X}}\text{softmax}(\tau^{-1}\tilde{\boldsymbol{X}}^T\boldsymbol{y})$. Using $\boldsymbol{p} = (p_1, \ldots, p_N)^T = \text{softmax}(\tau^{-1}\boldsymbol{X}^T\boldsymbol{y})$, the gradient of InfoNCE with respect to $\boldsymbol{y}$ is $-\tau^{-1}(1 - p_1)(\boldsymbol{x}_1 - \tilde{\boldsymbol{X}}\text{softmax}(\tau^{-1}\tilde{\boldsymbol{X}}^T\boldsymbol{y}))$ and its gradient with respect to $\boldsymbol{x}_1$ is $-\tau^{-1}(1 - p_1)\boldsymbol{y}$ (see Appendix Subsection A.1.4).

By and large, the gradient of InfoNCE is scaled by $(1 - p_1)$ compared to the gradient of InfoLOOB, where $p_1$ is softmax similarity between the anchor $\boldsymbol{y}$ and positive sample $\boldsymbol{x}_1$. Consequently, InfoNCE saturates and learning stalls when anchor and positive sample become similar to each other.

## 3  CLOOB: INFOLOOB WITH MODERN HOPFIELD NETWORKS

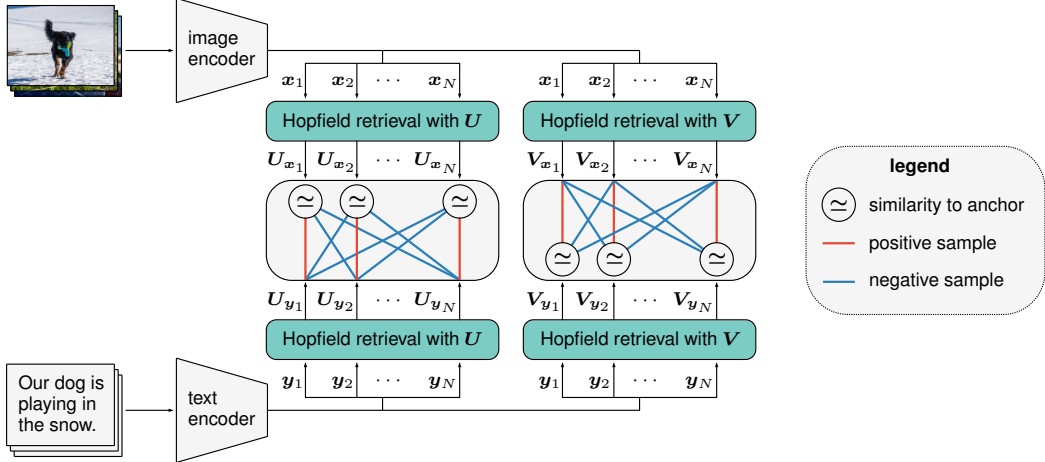

Figure 1: The CLOOB architecture for image-text pairs. The image embedding $\boldsymbol{x}_i$ and the text embedding $\boldsymbol{y}_i$ retrieve the embeddings $\boldsymbol{U}_{\boldsymbol{x}_i}$ and $\boldsymbol{U}_{\boldsymbol{y}_i}$, respectively, from a modern Hopfield network that stores image embeddings $\boldsymbol{U} = (\boldsymbol{u}_1, \ldots, \boldsymbol{u}_M)$ (green boxes at the left). The image-retrieved image embedding $\boldsymbol{U}_{\boldsymbol{x}_i}$ serves as anchor in order to contrast the positive text-retrieved image embedding $\boldsymbol{U}_{\boldsymbol{y}_i}$ with the negative text-retrieved image embedding $\boldsymbol{U}_{\boldsymbol{y}_j}$ for $j \neq i$. Analog, for the second modern Hopfield network that stores text embeddings $\boldsymbol{V} = (\boldsymbol{v}_1, \ldots, \boldsymbol{v}_K)$ (green boxes at the right).

**CLOOB for contrastive learning.** Our novel Contrastive Leave One Out Boost (CLOOB) combines the InfoLOOB objective with modern Hopfield networks. Modern Hopfield networks substitute the original by retrieved embeddings, thereby reduce the variance of InfoLOOB and reinforce the covariance structure in the data. Figure 1 sketches the CLOOB architecture for image-text pairs.

The training set consists of $N$ pairs of embeddings $\{(\boldsymbol{x}_1, \boldsymbol{y}_1), \ldots, (\boldsymbol{x}_N, \boldsymbol{y}_N)\}$, $M$ stored embeddings $\boldsymbol{U} = (\boldsymbol{u}_1, \ldots, \boldsymbol{u}_M)$, and $K$ stored embeddings $\boldsymbol{V} = (\boldsymbol{v}_1, \ldots, \boldsymbol{v}_K)$. The state or query embeddings $\boldsymbol{x}_i$ and $\boldsymbol{y}_i$ retrieve $\boldsymbol{U}_{\boldsymbol{x}_i}$ and $\boldsymbol{U}_{\boldsymbol{y}_i}$, respectively, from $\boldsymbol{U}$ — analogous notation for retrievals from $\boldsymbol{V}$. All samples are normalized: $\|\boldsymbol{x}_i\| = \|\boldsymbol{y}_i\| = \|\boldsymbol{u}_i\| = \|\boldsymbol{v}_i\| = 1$. The following vectors are retrieved from modern Hopfield networks (Ramsauer et al., 2021):

$$\boldsymbol{U}_{\boldsymbol{x}_i} = \boldsymbol{U} \operatorname{softmax}(\beta\, \boldsymbol{U}^T \boldsymbol{x}_i)\,, \quad \boldsymbol{U}_{\boldsymbol{y}_i} = \boldsymbol{U} \operatorname{softmax}(\beta\, \boldsymbol{U}^T \boldsymbol{y}_i)\,, \tag{9}$$

$$\boldsymbol{V}_{\boldsymbol{x}_i} = \boldsymbol{V} \operatorname{softmax}(\beta\, \boldsymbol{V}^T \boldsymbol{x}_i)\,, \quad \boldsymbol{V}_{\boldsymbol{y}_i} = \boldsymbol{V} \operatorname{softmax}(\beta\, \boldsymbol{V}^T \boldsymbol{y}_i) \tag{10}$$

where $\boldsymbol{U}_{\boldsymbol{x}_i}$ denotes an image-retrieved image embedding, $\boldsymbol{U}_{\boldsymbol{y}_i}$ a text-retrieved image embedding, $\boldsymbol{V}_{\boldsymbol{x}_i}$ an image-retrieved text embedding and $\boldsymbol{V}_{\boldsymbol{y}_i}$ a text-retrieved text embedding. The hyperparameter $\beta$ corresponds to the inverse temperature: $\beta = 0$ retrieves the average of the stored pattern, while large $\beta$ retrieves the stored pattern that is most similar to the state pattern (query).

In InfoLOOB, CLOOB substitutes the embedded samples $\boldsymbol{x}_i$ and $\boldsymbol{y}_i$ by the retrieved embedded samples. In the first term, $\boldsymbol{x}_i$ and $\boldsymbol{y}_i$ are substituted by $\boldsymbol{U}_{\boldsymbol{x}_i}$ and $\boldsymbol{U}_{\boldsymbol{y}_i}$, respectively, while in the second term by $\boldsymbol{V}_{\boldsymbol{x}_i}$ and $\boldsymbol{V}_{\boldsymbol{y}_i}$. All retrieved samples are normalized, $\|\boldsymbol{U}_{\boldsymbol{x}_i}\| = \|\boldsymbol{U}_{\boldsymbol{y}_i}\| = \|\boldsymbol{V}_{\boldsymbol{x}_i}\| = \|\boldsymbol{V}_{\boldsymbol{y}_i}\| = 1$. We obtain the InfoLOOB loss function that is used by CLOOB:

$$\mathrm{L}_{\mathrm{InfoLOOB}} = -\frac{1}{N} \sum_{i=1}^{N} \ln \frac{\exp(\tau^{-1}\, \boldsymbol{U}_{\boldsymbol{x}_i}^T \boldsymbol{U}_{\boldsymbol{y}_i})}{\sum_{j \neq i}^{N} \exp(\tau^{-1}\, \boldsymbol{U}_{\boldsymbol{x}_i}^T \boldsymbol{U}_{\boldsymbol{y}_j})} - \frac{1}{N} \sum_{i=1}^{N} \ln \frac{\exp(\tau^{-1}\, \boldsymbol{V}_{\boldsymbol{x}_i}^T \boldsymbol{V}_{\boldsymbol{y}_i})}{\sum_{j \neq i}^{N} \exp(\tau^{-1}\, \boldsymbol{V}_{\boldsymbol{x}_j}^T \boldsymbol{V}_{\boldsymbol{y}_i})}\,. \tag{11}$$

**Modern Hopfield Networks reduce high variance of InfoLOOB.** CLOOB uses InfoLOOB as objective, since it estimates the mutual information (MI) better than InfoNCE, in particular, for large MI. Cheng et al. (2020, Fig. 1 and Fig. 2) show that InfoLOOB is a better estimator for the MI than InfoNCE (van den Oord et al., 2018), MINE (Belghazi et al., 2018), and NWJ (Nguyen et al., 2010). We experimentally confirmed that InfoLOOB better estimates the mutual information than InfoNCE.

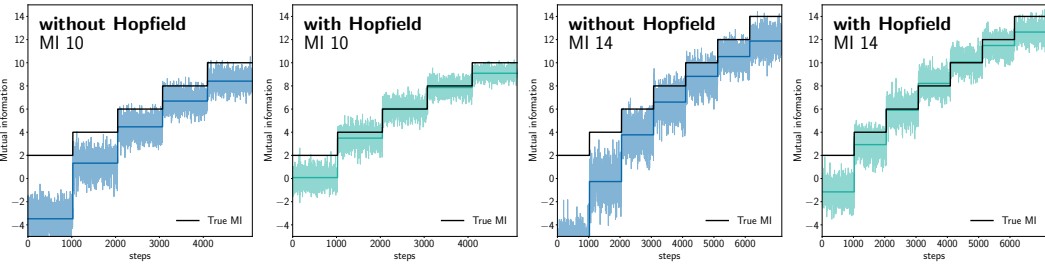

Figure 2: Variance reduction of InfoLOOB by modern Hopfield networks. From left to right: without Hopfield for MI 10, with Hopfield for MI 10, without Hopfield for MI 14, with Hopfield for MI 14. Modern Hopfield networks reduce the variance of the InfoLOOB loss.

However, InfoLOOB has higher variance than lower bounds on MI like InfoNCE, which considerably hampers learning (Cheng et al., 2020, Fig. 1 and Fig. 2), see also Appendix Section A.2. The InfoNCE objective has the form $a/(a + b)$ while InfoLOOB has the form $a/b$ with $a$ giving the anchor-to-positive similarity and $b$ the average anchor-to-negative similarity. For small $b$, we observe high variance and instability of InfoLOOB. Modern Hopfield networks (Ramsauer et al., 2021) are a remedy for the high variance. Modern Hopfield networks substitute the original patterns by retrieved patterns, which are an average over the stored patterns. We tested the variance of MI estimators/bounds on toy tasks, with samples drawn from Gaussian distributions following (Belghazi et al., 2018; Poole et al., 2019; Cheng et al., 2020). With the InfoLOOB objective, we train deep learning architectures with and without modern Hopfield networks on top, where the current learning batch is stored in the modern Hopfield networks. We used training data with mutual information of

10 and 14, where the parameters were optimized for the best performance on a validation set. We test the final model on different levels of mutual information. Figure 2 shows that modern Hopfield networks reduce the variance of the model. The average variances are reduced from $0.67$ to $0.33$ for MI 10 and from $1.00$ to $0.48$ for MI 14 (more details in Appendix A.2).

**Modern Hopfield Networks amplify the covariance structure in the data.** The covariance structure is extracted by the retrieved embeddings $\boldsymbol{U}_{\boldsymbol{x}_i}^T \boldsymbol{U}_{\boldsymbol{y}_i}$ and $\boldsymbol{V}_{\boldsymbol{x}_i}^T \boldsymbol{V}_{\boldsymbol{y}_i}$. The Jacobian J of the softmax $\boldsymbol{p} = \mathrm{softmax}(\beta \boldsymbol{a})$ is $\mathrm{J}(\beta \boldsymbol{a}) = \beta \left( \mathrm{diag}(\boldsymbol{p}) - \boldsymbol{p}\boldsymbol{p}^T \right)$. We define the *weighted covariance* $\mathrm{Cov}(\boldsymbol{U})$, where sample $\boldsymbol{u}_i$ is drawn with probability $p_i$, as $[\mathrm{Cov}(\boldsymbol{U})]_{kl} = \left[ \boldsymbol{U}\mathrm{J}(\beta \boldsymbol{a})\boldsymbol{U}^T \right]_{kl} = \beta(\sum_{i=1}^M p_i u_{ik} u_{il} - \sum_{i=1}^M p_i u_{ik} \sum_{i=1}^M p_i u_{il})$. The formula of the weighted covariance differs from the standard empirical covariance, since the factor $1/M$ is replaced by $p_i$. Thus $\boldsymbol{u}_i$ is sampled with probability $p_i$ instead of being sampled uniformly with probability $1/M$.

We apply the mean value theorem to the softmax function with mean Jacobian matrix $\mathrm{J}^{\mathrm{m}}(\beta \boldsymbol{a}) = \int_0^1 \mathrm{J}(\lambda \beta \boldsymbol{a}) \, \mathrm{d}\lambda$. The mean Jacobian $\mathrm{J}^{\mathrm{m}}(\beta \boldsymbol{a})$ is a symmetric, diagonally dominant, positive semi-definite matrix with one eigenvalue of zero for eigenvector $\boldsymbol{1}$ and spectral norm bounded by $\|\mathrm{J}^{\mathrm{m}}\|_2 \leqslant 0.5\beta$ (see Appendix Lemma A1). We can express $\boldsymbol{U}_{\boldsymbol{x}_i}^T \boldsymbol{U}_{\boldsymbol{y}_i}$ as (see Appendix Theorem A3):

$$\boldsymbol{U}_{\boldsymbol{x}_i}^T \boldsymbol{U}_{\boldsymbol{y}_i} = \left( \bar{\boldsymbol{u}} + \mathrm{Cov}(\boldsymbol{U}, \boldsymbol{x}_i) \, \boldsymbol{x}_i \right)^T \left( \bar{\boldsymbol{u}} + \mathrm{Cov}(\boldsymbol{U}, \boldsymbol{y}_i) \, \boldsymbol{y}_i \right) , \tag{12}$$

where the mean is $\bar{\boldsymbol{u}} = 1/M \boldsymbol{U}\boldsymbol{1}$ and the weighted covariances are $\mathrm{Cov}(\boldsymbol{U}, \boldsymbol{x}_i) = \boldsymbol{U}\mathrm{J}^{\mathrm{m}}(\beta \boldsymbol{U}^T \boldsymbol{x}_i)\boldsymbol{U}^T$ and $\mathrm{Cov}(\boldsymbol{U}, \boldsymbol{y}_i) = \boldsymbol{U}\mathrm{J}^{\mathrm{m}}(\beta \boldsymbol{U}^T \boldsymbol{y}_i)\boldsymbol{U}^T$. The weighted covariance $\mathrm{Cov}(\boldsymbol{U}, .)$ is the covariance if the stored pattern $\boldsymbol{u}_i$ is drawn according to an averaged $p_i$ given by $\mathrm{J}^{\mathrm{m}}(.)$. When maximizing the dot product $\boldsymbol{U}_{\boldsymbol{x}_i}^T \boldsymbol{U}_{\boldsymbol{y}_i}$, the normalized vectors $\boldsymbol{x}_i$ and $\boldsymbol{y}_i$ are encouraged to agree on drawing the patterns $\boldsymbol{u}_i$ with the same probability $p_i$ in order to generate similar weighted covariance matrices $\mathrm{Cov}(\boldsymbol{U}, .)$. If subsets of $\boldsymbol{U}$ have a strong covariance structure, then it can be exploited to produce large weighted covariances and, in turn, large dot products of $\boldsymbol{U}_{\boldsymbol{x}_i}^T \boldsymbol{U}_{\boldsymbol{y}_i}$. Furthermore, for a large dot product $\boldsymbol{U}_{\boldsymbol{x}_i}^T \boldsymbol{U}_{\boldsymbol{y}_i}$, $\boldsymbol{x}_i$ and $\boldsymbol{y}_i$ have to be similar to each other to extract the same direction from the covariance matrices. Above considerations for $\boldsymbol{U}_{\boldsymbol{x}_i}^T \boldsymbol{U}_{\boldsymbol{y}_i}$ analogously apply to $\boldsymbol{V}_{\boldsymbol{x}_i}^T \boldsymbol{V}_{\boldsymbol{y}_i}$.

We did not use a loss function that contains dot products like $\boldsymbol{U}_{\boldsymbol{x}_i}^T \boldsymbol{V}_{\boldsymbol{y}_i}$, because these dot products have higher variance than the ones we have used. The dot product $\boldsymbol{U}_{\boldsymbol{x}_i}^T \boldsymbol{V}_{\boldsymbol{y}_i}$ has higher variance, since it uses $M + K$ stored patterns, whereas $\boldsymbol{U}_{\boldsymbol{x}_i}^T \boldsymbol{U}_{\boldsymbol{y}_i}$ and $\boldsymbol{V}_{\boldsymbol{x}_i}^T \boldsymbol{V}_{\boldsymbol{y}_i}$ use $M$ and $K$, respectively.

**Modern Hopfield Networks can reuse training samples as stored patterns.** We use the training samples as the stored patterns in the modern Hopfield network. Hence, we set $\boldsymbol{u}_i = \boldsymbol{x}_i$ and $\boldsymbol{v}_i = \boldsymbol{y}_i$, that is, $\boldsymbol{U} = \boldsymbol{X}$ and $\boldsymbol{V} = \boldsymbol{Y}$. Consequently, we store the learning batch in the modern Hopfield networks as $\boldsymbol{U}$ and $\boldsymbol{V}$. In particular this means that $\boldsymbol{x}_i$ can retrieve itself from $\boldsymbol{U} = \boldsymbol{X}$ but not from $\boldsymbol{V} = \boldsymbol{Y}$. Analogously, $\boldsymbol{y}_i$ can retrieve itself from $\boldsymbol{V} = \boldsymbol{Y}$ but not from $\boldsymbol{U} = \boldsymbol{X}$.

**Modern Hopfield networks allow the usage of retrieved embeddings.** After learning, both the model embeddings $\boldsymbol{x}$ and $\boldsymbol{y}$ as well as the retrieved embeddings $\boldsymbol{U}_{\boldsymbol{x}}, \boldsymbol{U}_{\boldsymbol{y}}, \boldsymbol{V}_{\boldsymbol{x}}$, and $\boldsymbol{V}_{\boldsymbol{y}}$ may serve for the downstream tasks, e.g. for zero-shot transfer learning. When using the retrieved embeddings, the modern Hopfield networks can store random samples, prototypes, templates, or proprietary samples. Therefore, particular embedding features can be amplified according to the task at hand.

**Modern Hopfield networks is a new concept for contrastive learning.** In bioinformatics the covariance structure in a sequence is reinforced by first retrieving similar sequences from a database and then aligning them. Conserved regions are characterized by high local covariance in the alignment (Dickson & Gloor, 2012; Kreth & Fodor, 2014). Modern Hopfield networks detect high covariances of embedded features, which is conveyed by the retrieved sample that corresponds to an alignment.

## 4 EXPERIMENTS

On two pretraining datasets, we compare our new CLOOB to CLIP (Radford et al., 2021) with respect to their capability of zero-shot transfer learning. The first dataset, Conceptual Captions (CC) (Sharma et al., 2018), has a very rich textual description of images but only three million image-text pairs. The second dataset, a subset of YFCC100M (Thomee et al., 2016), has 15 million image-text pairs but the textual description is less rich than for CC and often vacuous. For both pretraining datasets, the downstream zero-shot transfer learning performance is tested on seven image classification datasets.

Table 1: Zero-shot results for models trained on CC with ResNet-50 vision encoders for two different checkpoints. Results are given as mean accuracy over 5 runs. Statistically significant results are shown in bold. CLIP and CLOOB were trained for 31 epochs while CLIP* and CLOOB* were trained for 128 epochs. In the majority of tasks CLOOB significantly outperforms CLIP.

| Dataset | CLIP RN-50 | CLOOB RN-50 | CLIP* RN-50 | CLOOB* RN-50 |
|---|---|---|---|---|
| Birdsnap | $2.26 \pm 0.20$ | $\mathbf{3.06 \pm 0.30}$ | $2.8 \pm 0.16$ | $\mathbf{3.24 \pm 0.31}$ |
| Country211 | $0.67 \pm 0.11$ | $0.67 \pm 0.05$ | $0.7 \pm 0.04$ | $0.73 \pm 0.05$ |
| Flowers102 | $12.56 \pm 0.38$ | $13.45 \pm 1.19$ | $13.32 \pm 0.43$ | $14.36 \pm 1.17$ |
| GTSRB | $7.66 \pm 1.07$ | $6.38 \pm 2.11$ | $8.96 \pm 1.70$ | $7.03 \pm 1.22$ |
| UCF101 | $20.98 \pm 1.55$ | $22.26 \pm 0.72$ | $21.63 \pm 0.65$ | $\mathbf{23.03 \pm 0.85}$ |
| Stanford Cars | $0.91 \pm 0.10$ | $\mathbf{1.23 \pm 0.10}$ | $0.99 \pm 0.16$ | $\mathbf{1.41 \pm 0.32}$ |
| ImageNet | $20.33 \pm 0.28$ | $\mathbf{23.97 \pm 0.15}$ | $21.3 \pm 0.42$ | $\mathbf{25.67 \pm 0.22}$ |
| ImageNet V2 | $20.24 \pm 0.50$ | $\mathbf{23.59 \pm 0.15}$ | $21.24 \pm 0.22$ | $\mathbf{25.49 \pm 0.11}$ |

## 4.1 Conceptual Captions Pretraining

**Pretraining dataset.** The Conceptual Captions (CC) (Sharma et al., 2018) dataset consists of 2.9 million images with high-quality captions. Images and their captions have been gathered via an automated process from the web and therefore represent a wide variety of content. Raw descriptions of images are collected from the *alt-text* HTML attribute. Both images and texts are filtered for high quality image-text pairs.

**Methods compared.** We compare our new CLOOB to CLIP (Radford et al., 2021). The CLOOB implementation is based on OpenCLIP (Ilharco et al., 2021), which achieves results equivalent to CLIP on the YFCC dataset (see Section 4.2). OpenCLIP also reports results on the CC dataset. As CLIP does not train models on CC we report results from this reimplementation as baseline. Analogously to Radford et al. (2021, Section 2.4), we use the modified ResNet (He et al., 2016) and BERT (Devlin et al., 2018; 2019) architectures to encode image and text input. We use the ResNet encoders ResNet-50, ResNet-101, and ResNet-50x4.

**Hyperparameter selection and learning schedule.** We use the hyperparameter values of OpenCLIP, concretely, a learning rate of $1 \times 10^{-3}$ and a weight decay of $0.1$ for the Adam optimizer (Kingma et al., 2014) with decoupled weight decay regularization (Loshchilov & Hutter, 2019). Deviating from OpenCLIP, we use a batch size of 512 due to computational restraints, which did not change the performance. The learning rate scheduler for all experiments is cosine annealing with warmup and hard restarts (Loshchilov & Hutter, 2017). We report the hyperparameter $\tau$ (default 0.07) from CLIP as $\tau^{-1}$ of 14.3 to be in the same regime as the hyperparameter $\beta$ for the modern Hopfield networks. The main hyperparameter search for CLOOB (also for YFCC pretraining in the next section) was done with ResNet-50 as the vision encoder. Learnable $\tau^{-1}$ in combination with the InfoLOOB loss results in undesired learning behavior (see Appendix Section A.1.4). Therefore, we set $\tau^{-1}$ to a fixed value of 30, which was determined via hyperparameter search (see Appendix Section A.3.2). For modern Hopfield networks, the hyperparameter $\beta$ was set to 8. Further we scale the loss in Eq. (11) with $\tau$ to remove the factor $\tau^{-1}$ from the gradients (see Appendix Section A.1.4) resulting in the loss function $\tau \mathrm{L}_{\mathrm{InfoLOOB}}$.

**Evaluation metrics: Zero-shot transfer learning.** We evaluate and compare both CLIP and CLOOB on their zero-shot transfer learning capabilities on the following downstream image classification tasks. Birdsnap (Berg et al., 2014) contains images of 500 different North American bird species. The Country211 (Radford et al., 2021) dataset consists of photos across 211 countries and is designed to test the geolocalization capability of visual representations. Flowers102 (Nilsback & Zisserman, 2008) is a dataset containing images of 102 flower species. GTSRB (Stallkamp et al., 2011) contains images for classification of German traffic signs. UCF101 (Soomro et al., 2012) is a video dataset with short clips for action recognition. For UCF101 we follow the procedure reported in CLIP and extract the middle frame of every video to assemble the dataset. Stanford Cars (Krause et al., 2013) contains images of 196 types of cars. ImageNet (Deng et al., 2009) is a large scale image classification dataset with images across 1,000 classes. ImageNetv2 (Recht et al., 2019) consists of

Table 2: Performance with InfoLOOB vs. InfoNCE objective and with vs. without Hopfield retrieval. InfoLOOB increases the performance of CLIP in most of the tasks. Hopfield with InfoLOOB strongly improves the performance in 7 out of 8 datasets compared to both CLIP models.

| Dataset | CLIP | | Hopfield | |
| --- | --- | --- | --- | --- |
| | InfoNCE | InfoLOOB | InfoNCE | InfoLOOB |
| Birdsnap | 1.94 | 2.37 | 1.67 | **2.53** |
| Country211 | 0.62 | 0.63 | 0.54 | **0.76** |
| Flowers102 | 13.04 | 13.03 | 11.53 | **14.24** |
| GTSRB | **7.28** | 4.39 | 5.76 | 5.86 |
| UCF101 | 21.00 | 19.14 | 20.56 | **22.29** |
| Stanford Cars | 0.90 | 1.33 | 1.24 | **1.37** |
| ImageNet | 20.31 | 22.13 | 19.04 | **24.21** |
| ImageNetV2 | 20.63 | 21.65 | 18.97 | **23.80** |

three new test sets with 10,000 images each for the ImageNet benchmark. For further details see Appendix Section A.3.3.

**Results.** We employ the same evaluation strategy and use the prompt templates as published in CLIP (see Appendix Section A.3.3). We report zero-shot results from two checkpoints in Table 1. CLIP and CLOOB were trained for a comparable number of epochs used in CLIP (see Appendix Section A.3.2) while CLIP* and CLOOB* were trained until evaluation performance plateaued (epoch 128). In both cases CLOOB significantly outperforms CLIP on the majority of tasks or matches its performance. Statistical significance of these results was assessed by an unpaired Wilcoxon test on a $5\%$ level.

**Ablation studies.** CLOOB has two new major components compared to CLIP: (1) the InfoLOOB objective instead of the InfoNCE objective and (2) the modern Hopfield networks. To assess which of the new major components of CLOOB has led to the performance increase over CLIP, we performed ablation studies on CC. First, we enhanced CLIP by replacing the InfoNCE objective with InfoLOOB. Table 2 shows that the InfoLOOB objective increases the performance of CLIP in the majority of the datasets. The reason is that InfoLOOB suffers less than InfoNCE from the "explaining away" problem. However, InfoLOOB is more effective for higher mutual information, that is, for a richer covariance structure. Hopfield networks amplify the covariance structure by retrieved embeddings. For InfoLOOB, however, this amplification is disadvantageous as the saturation effect is increased by higher similarity between anchor and positive. Thus, combining modern Hopfield networks with InfoNCE leads to a performance drop. Combining Hopfield and InfoLOOB into CLOOB strongly improves the performance on 7 out of 8 zero-shot transfer learning tasks. An additional ablation considers the learning rate scheduler. For more details see in Appendix Section A.3.1.

## 4.2 YFCC PRETRAINING

**Pretraining dataset.** To be comparable to the CLIP results, we use the same subset of 15 million samples from the YFCC100M dataset (Thomee et al., 2016) as in Radford et al. (2021), which we refer to as YFCC. YFCC was created by filtering YFCC100M for images which contain natural language descriptions and/or titles in English. It was not filtered by quality of the captions, therefore the textual descriptions are less rich and contain superfluous information. The dataset with 400 million samples used to train the CLIP models in Radford et al. (2021) has not been released and, thus, is not available for comparison. Due to limited computational resources we are unable to compare CLOOB to CLIP on other datasets of this size.

**Methods compared and evaluation.** In addition to the comparison of CLOOB and CLIP based on the OpenCLIP reimplementation (Ilharco et al., 2021), we include the original CLIP results (Radford et al., 2021, Table 12).

**Hyperparameter selection.** We use the hyperparameters selected at the Conceptual Captions dataset, except learning rate, batch size, and $\beta$. For modern Hopfield networks, the hyperparameter $\beta$ is set to $14.3$, which is the default parameter of $\tau^{-1}$ for the InfoNCE objective in Radford et al. (2021). Furthermore, the learning rate is set to $5 \times 10^{-4}$ and a batch size of $1024$ as in OpenCLIP of Ilharco et al. (2021). For further details see Appendix Section A.3.2.

**Evaluation metrics.** As in the previous experiment, methods are again evaluated at their zero-shot transfer learning capabilities on downstream tasks.

**Results.** Table 3 provides results of the original CLIP and CLOOB trained on YFCC. The results on zero-shot downstream tasks show that CLOOB outperforms the results of CLIP on all 7 tasks (ImageNet V2 results have not been reported in Radford et al. (2021)). Similarly, CLOOB outperforms CLIP on 6 out of 7 tasks for linear probing. Results of the comparison of CLOOB an the CLIP reimplementation of OpenCLIP are given in Table 4. CLOOB exceeds the CLIP reimplementation in 7 out of 8 tasks for zero-shot classification using ResNet-50 encoders. With larger ResNet encoders, CLOOB outperforms CLIP on all tasks. Furthermore, the experiments with larger vision encoder networks show that CLOOB performance increases with network size. Visualizations of predictions of CLOOB zero-shot classifiers from all datasets are shown in Appendix Section A.3.4.

Table 3: Results of CLIP and CLOOB trained on YFCC with ResNet-50 encoder. Except for one linear probing dataset, CLOOB consistently outperforms CLIP across all tasks.

| Dataset | Linear Probing | | Zero-Shot | |
| | CLIP (OpenAI) | CLOOB (ours) | CLIP (OpenAI) | CLOOB (ours) |
|---|---|---|---|---|
| Birdsnap | 47.4 | **56.2** | 19.9 | **28.9** |
| Country211 | **23.1** | 20.6 | 5.2 | **7.9** |
| Flowers102 | 94.4 | **96.1** | 48.6 | **55.1** |
| GTSRB | 66.8 | **78.9** | 6.9 | **8.1** |
| UCF101 | 69.2 | **72.3** | 22.9 | **25.3** |
| Stanford Cars | 31.4 | **37.7** | 3.8 | **4.1** |
| ImageNet | 62.0 | **65.7** | 31.3 | **35.7** |
| ImageNet V2 | - | 58.7 | - | 34.6 |

Table 4: Zero-shot results for the CLIP reimplementation and CLOOB using different ResNet architectures trained on YFCC. CLOOB outperforms CLIP in 7 out of 8 tasks using ResNet-50 encoders. With larger ResNet encoders CLOOB outperforms CLIP on all tasks. The performance of CLOOB scales with increased encoder size.

| Dataset | CLIP RN-50 | CLOOB RN-50 | CLIP RN-101 | CLOOB RN-101 | CLIP RN-50x4 | CLOOB RN-50x4 |
|---|---|---|---|---|---|---|
| Birdsnap | 21.8 | **28.9** | 22.6 | **30.3** | 20.8 | **32.0** |
| Country211 | 6.9 | **7.9** | 7.8 | **8.5** | 8.1 | **9.3** |
| Flowers102 | 48.0 | **55.1** | 48.0 | **55.3** | 50.1 | **54.3** |
| GTSRB | 7.9 | **8.1** | 7.4 | **11.6** | 9.4 | **11.8** |
| UCF101 | **27.2** | 25.3 | 28.6 | **28.8** | 31.0 | **31.9** |
| Stanford Cars | 3.7 | **4.1** | 3.8 | **5.5** | 3.5 | **6.1** |
| ImageNet | 34.6 | **35.7** | 35.3 | **37.1** | 37.7 | **39.0** |
| ImageNet V2 | 33.4 | **34.6** | 34.1 | **35.6** | 35.9 | **37.3** |

## 5 CONCLUSION

For constrastive learning, we have introduced "Contrastive Leave One Out Boost" (CLOOB), for which modern Hopfield networks boost learning with the InfoLOOB objective. Modern Hopfield networks both increase the stability of InfoLOOB and reinforce the covariance structure of the data. We have shown theoretical properties of the InfoLOOB bound and objective. Our results suggest InfoLOOB as an alternative to InfoNCE in contrastive learning. An ablation study shows that both, the InfoLOOB objective and modern Hopfield networks, are necessary to yield high performance. At seven zero-shot transfer learning tasks, the novel CLOOB is compared to CLIP after pretraining on Conceptual Captions and the YFCC dataset. CLOOB consistently outperforms CLIP at zero-shot transfer learning across all considered architectures and datasets.

## REPRODUCIBILITY STATEMENT

We will publish the source code after the reviewing period. This will ensure that the results are reproducible in their entirety. The datasets used for training our models as well as for the downstream tasks are publicly available.

## ETHICAL CONSIDERATIONS

**Impact on ML and related scientific fields.** Our research has the potential to positively impact a wide variety of fields of life due to its general applicability. Most importantly, it has the potential to reduce the cost for training other AI systems, which could lead to a reduction of compute costs and carbon dioxide emissions.

However, any new development in machine learning can be applied for good or for bad. Our system can be used for medical applications where it could save lives but might also be used for surveillance and malevolent systems.

**Impact on society.** A potential danger could arise from an application of our approach in which users rely overly on the outcomes. For example, in a medical setting, physicians might rely on the technical system and shift the liability towards the machine. This might also happen in the domain of self-driving cars, when drivers start paying less attention to the traffic because of an AI-based driving system. Finally, our method may also be deployed in companies to automate various simple tasks, which might lead to a reduced need for particular jobs in production systems.

**Consequences of failures of the method.** Depending on the application area, a failure of this method might be of lesser concern, such as a failed execution of a computer program. If our method is employed within a larger automation system, a failure could result in damages such as a car accident or errors of a production system. However, this holds for almost all machine learning methods, and their usage and testing depends on the application area.

**Leveraging of biases in the data and potential discrimination.** Our proposed method relies on human-annotated data and thereby human decisions, which are usually strongly biased. The undesirable biases contained in dataset are learned and may propagate to downstream applications. Therefore, the responsible use of our method depends on a careful selection of the training data and awareness of the potential biases within those.

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

## A APPENDIX

This appendix consists of four sections (A.1–A.4). Section A.1 provides the theoretical properties of the InfoLOOB and InfoNCE. It is shown how to derive that InfoNCE is a lower bound on mutual information. Further it is shown how to derive that InfoLOOB is an upper bound on mutual information. The proposed loss function $\mathrm{L_{InfoLOOB}}$ and its gradients are discussed. In Section A.2 we discuss the estimation of mutual information for a toy example. Section A.3 provides details on the experiments for Section 4. Section A.4 briefly reviews continuous modern Hopfield networks. Section A.5 discusses further related work.

## CONTENTS OF THE APPENDIX

## LIST OF THEOREMS

## LIST OF DEFINITIONS

## LIST OF FIGURES

## LIST OF TABLES

### A.1 INFOLOOB VS. INFONCE

### A.1.1 INFONCE: LOWER BOUND ON MUTUAL INFORMATION

We derive a lower bound on the mutual information between random variables $X$ and $Y$ distributed according to $p(\boldsymbol{x}, \boldsymbol{y})$. The mutual information $\mathrm{I}(X ; Y)$ between random variables $X$ and $Y$ is

$$\mathrm{I}(X ; Y) = \mathrm{E}_{p(\boldsymbol{x},\boldsymbol{y})}\left[\ln \frac{p(\boldsymbol{x},\boldsymbol{y})}{p(\boldsymbol{x})\,p(\boldsymbol{y})}\right] = \mathrm{E}_{p(\boldsymbol{x},\boldsymbol{y})}\left[\ln \frac{p(\boldsymbol{x}\mid\boldsymbol{y})}{p(\boldsymbol{x})}\right] = \mathrm{E}_{p(\boldsymbol{x},\boldsymbol{y})}\left[\ln \frac{p(\boldsymbol{y}\mid\boldsymbol{x})}{p(\boldsymbol{y})}\right] . \tag{A1}$$

"InfoNCE" has been introduced in van den Oord et al. (2018) and is a *multi-sample bound*. In the setting introduced in van den Oord et al. (2018), we have an anchor sample $\boldsymbol{y}$ given. For the anchor sample $\boldsymbol{y}$ we draw a positive sample $\boldsymbol{x}_1$ according to $p(\boldsymbol{x}_1 \mid \boldsymbol{y})$. Next, we draw a set $\tilde{X} = \{\boldsymbol{x}_2, \ldots, \boldsymbol{x}_N\}$ according to $p(\tilde{X})$, which are $n-1$ negative samples drawn iid according to $p(\boldsymbol{x})$. We have drawn a set $X = \{\boldsymbol{x}_1, \boldsymbol{x}_2, \ldots, \boldsymbol{x}_N\}$ according to $p(X \mid \boldsymbol{y})$, which is one positive sample $\boldsymbol{x}_1$ drawn by $p(\boldsymbol{x}_1 \mid \boldsymbol{y})$ and $N-1$ negative samples $\{\boldsymbol{x}_2, \ldots, \boldsymbol{x}_N\}$ drawn iid according to $p(\boldsymbol{x})$.

The InfoNCE with probabilities is

$$\mathrm{I}_{\mathrm{InfoNCE}}(X_1 ; Y) = \mathrm{E}_{p(\boldsymbol{y})}\left[\mathrm{E}_{p(X|\boldsymbol{y})}\left[\ln\left(\frac{p(\boldsymbol{y}\mid\boldsymbol{x}_1)}{\frac{1}{N}\sum_{i=1}^{N}p(\boldsymbol{y}\mid\boldsymbol{x}_i)}\right)\right]\right] , \tag{A2}$$

where we inserted the factor $\frac{1}{N}$ in contrast to the original version in van den Oord et al. (2018), where we followed Poole et al. (2019); Tschannen et al. (2019); Cheng et al. (2020); Chen et al. (2021).

The InfoNCE with score function $f(\boldsymbol{x}, \boldsymbol{y})$ is

$$\mathrm{I}_{\mathrm{InfoNCE}}(X_1 ; Y) = \mathrm{E}_{p(\boldsymbol{y})}\left[\mathrm{E}_{p(X|\boldsymbol{y})}\left[\ln\left(\frac{f(\boldsymbol{x}_1,\boldsymbol{y})}{\frac{1}{N}\sum_{i=1}^{N}f(\boldsymbol{x}_i,\boldsymbol{y})}\right)\right]\right] . \tag{A3}$$

The InfoNCE with probabilities can be rewritten as:

$$\begin{aligned}
\mathrm{I}_{\mathrm{InfoNCE}}(X_1 ; Y) &= \mathrm{E}_{p(\boldsymbol{y})}\left[\mathrm{E}_{p(X|\boldsymbol{y})}\left[\ln\left(\frac{p(\boldsymbol{y}\mid\boldsymbol{x}_1)}{\frac{1}{N}\sum_{i=1}^{N}p(\boldsymbol{y}\mid\boldsymbol{x}_i)}\right)\right]\right] \\
&= \mathrm{E}_{p(\boldsymbol{y})}\left[\mathrm{E}_{p(X|\boldsymbol{y})}\left[\ln\left(\frac{\frac{p(\boldsymbol{y}|\boldsymbol{x}_1)}{p(\boldsymbol{y})}}{\frac{1}{N}\sum_{i=1}^{N}\frac{p(\boldsymbol{y}|\boldsymbol{x}_i)}{p(\boldsymbol{y})}}\right)\right]\right] \\
&= \mathrm{E}_{p(\boldsymbol{y})}\left[\mathrm{E}_{p(X|\boldsymbol{y})}\left[\ln\left(\frac{\frac{p(\boldsymbol{x}_1|\boldsymbol{y})}{p(\boldsymbol{x}_1)}}{\frac{1}{N}\sum_{i=1}^{N}\frac{p(\boldsymbol{x}_i|\boldsymbol{y})}{p(\boldsymbol{x}_i)}}\right)\right]\right] .
\end{aligned} \tag{A4}$$

This is the InfoNCE with $f(\boldsymbol{x}, \boldsymbol{y}) = p(\boldsymbol{y} \mid \boldsymbol{x})$.

**Set of pairs.** The InfoNCE can be written in a different setting Poole et al. (2019), which is used in most implementations. We sample $N$ pairs independently from $p(\boldsymbol{x}, \boldsymbol{y})$, which gives $Z = \{(\boldsymbol{x}_1, \boldsymbol{y}_1), (\boldsymbol{x}_2, \boldsymbol{y}_2), \ldots, (\boldsymbol{x}_N, \boldsymbol{y}_N)\}$. The InfoNCE is then

$$\mathrm{I}_{\mathrm{InfoNCE}}(X ; Y) = \mathrm{E}_{p(X|\boldsymbol{y})}\left[\frac{1}{N}\sum_{i=1}^{N}\ln\left(\frac{f(\boldsymbol{x}_i,\boldsymbol{y}_i)}{\frac{1}{N}\sum_{j=1}^{N}f(\boldsymbol{x}_j,\boldsymbol{y}_i)}\right)\right] . \tag{A5}$$

Following van den Oord et al. (2018) we have

$$
\begin{aligned}
\mathrm{I}_{\mathrm{InfoNCE}}(X_1 \; ; \; Y) \; &= \; \mathrm{E}_{p(\boldsymbol{y})} \left[ \mathrm{E}_{p(X|\boldsymbol{y})} \left[ \ln \left( \frac{\frac{p(\boldsymbol{y}|\boldsymbol{x}_1)}{p(\boldsymbol{y})}}{\frac{1}{N} \sum_{i=1}^{N} \frac{p(\boldsymbol{y}|\boldsymbol{x}_i)}{p(\boldsymbol{y})}} \right) \right] \right] \qquad (\text{A6}) \\
&= \; \mathrm{E}_{p(\boldsymbol{y})} \left[ \mathrm{E}_{p(X|\boldsymbol{y})} \left[ \ln \left( \frac{\frac{p(\boldsymbol{x}_1|\boldsymbol{y})}{p(\boldsymbol{x}_1)}}{\frac{1}{N} \sum_{i=1}^{N} \frac{p(\boldsymbol{x}_i|\boldsymbol{y})}{p(\boldsymbol{x}_i)}} \right) \right] \right] \\
&= \; \mathrm{E}_{p(\boldsymbol{y})} \left[ \mathrm{E}_{p(X|\boldsymbol{y})} \left[ \ln \left( \frac{p(\boldsymbol{x}_1 \mid \boldsymbol{y}) \prod_{l=2}^{N} p(\boldsymbol{x}_l)}{\sum_{i=1}^{N} p(\boldsymbol{x}_i \mid \boldsymbol{y}) \prod_{l \neq i} p(\boldsymbol{x}_l)} \right) \right] \right] + \ln(N) \\
&= \; \mathrm{E}_{p(\boldsymbol{y})} \left[ \mathrm{E}_{p(X|\boldsymbol{y})} \left[ \ln p(i = 1 \mid X, \boldsymbol{y}) \right] \right] + \ln(N) \, ,
\end{aligned}
$$

where $p(i = 1 \mid X, \boldsymbol{y})$ is the probability that sample $\boldsymbol{x}_1$ is the positive sample if we know there exists exactly one positive sample in $X$.

The InfoNCE is a lower bound on the mutual information. The following inequality is from van den Oord et al. (2018):

$$
\begin{aligned}
\mathrm{I}(X_1 \; ; \; Y) \; &= \; \mathrm{E}_{p(\boldsymbol{y})} \left[ \mathrm{E}_{p(\boldsymbol{x}_1|\boldsymbol{y})} \left[ \ln \left( \frac{p(\boldsymbol{x}_1 \mid \boldsymbol{y})}{p(\boldsymbol{x}_1)} \right) \right] \right] \qquad (\text{A7}) \\
&= \; \mathrm{E}_{p(\boldsymbol{y})} \left[ \mathrm{E}_{p(\boldsymbol{x}_1|\boldsymbol{y})} \left[ - \ln \left( \frac{p(\boldsymbol{x}_1)}{p(\boldsymbol{x}_1 \mid \boldsymbol{y})} \right) \right] \right] \\
&\geq \; \mathrm{E}_{p(\boldsymbol{y})} \left[ \mathrm{E}_{p(\boldsymbol{x}_1|\boldsymbol{y})} \left[ - \ln \left( \frac{1}{N} + \frac{p(\boldsymbol{x}_1)}{p(\boldsymbol{x}_1 \mid \boldsymbol{y})} \right) \right] \right] \\
&\approx \; \mathrm{E}_{p(\boldsymbol{y})} \left[ \mathrm{E}_{p(X|\boldsymbol{y})} \left[ - \ln \left( \frac{1}{N} + \frac{1}{N} \frac{p(\boldsymbol{x}_1)}{p(\boldsymbol{x}_1 \mid \boldsymbol{y})} \sum_{i=2}^{N} \frac{p(\boldsymbol{x}_i \mid \boldsymbol{y})}{p(\boldsymbol{x}_i)} \right) \right] \right] \\
&= \; \mathrm{E}_{p(\boldsymbol{y})} \left[ \mathrm{E}_{p(X|\boldsymbol{y})} \left[ \ln \left( \frac{\frac{p(\boldsymbol{x}_1|\boldsymbol{y})}{p(\boldsymbol{x}_1)}}{\frac{1}{N} \frac{p(\boldsymbol{x}_1|\boldsymbol{y})}{p(\boldsymbol{x}_1)} + \frac{1}{N} \sum_{i=2}^{N} \frac{p(\boldsymbol{x}_i|\boldsymbol{y})}{p(\boldsymbol{x}_i)}} \right) \right] \right] \\
&= \; \mathrm{I}_{\mathrm{InfoNCE}}(X_1 \; ; \; Y) \, ,
\end{aligned}
$$

where the "$\geq$" is obtained by bounding $\ln(1/N + a)$ by $\ln(a)$, which gives a bound that is not very tight, since $a = \frac{p(\boldsymbol{x}_1)}{p(\boldsymbol{x}_1|\boldsymbol{y})}$ can become small. However for the "$\approx$" van den Oord et al. (2018) have to assume

$$
\frac{1}{N} \sum_{i=2}^{N} \frac{p(\boldsymbol{x}_i \mid \boldsymbol{y})}{p(\boldsymbol{x}_i)} \; = \; \frac{1}{N} \sum_{i=2}^{N} \frac{p(\boldsymbol{y} \mid \boldsymbol{x}_i)}{p(\boldsymbol{y})} \; \geq \; 1 \, , \qquad (\text{A8})
$$

which is unclear how to ensure.

For a proof of this bound see Poole et al. (2019).

We assumed that for the anchor sample $\boldsymbol{y}$ a positive sample $\boldsymbol{x}_1$ has been drawn according to $p(\boldsymbol{x}_1 \mid \boldsymbol{y})$. A set $\tilde{X} = \{\boldsymbol{x}_2, \ldots, \boldsymbol{x}_N\}$ of negative samples is drawn according to $p(\boldsymbol{x})$. Therefore, we have a set $X = \{\boldsymbol{x}_1, \boldsymbol{x}_2, \ldots, \boldsymbol{x}_N\}$ that is drawn with one positive sample $\boldsymbol{x}_1$ and $N - 1$ negative samples $\tilde{X} = \{\boldsymbol{x}_2, \ldots, \boldsymbol{x}_N\}$. We have

$$
p(\tilde{X}) \; = \; \prod_{i=2}^{N} p(\boldsymbol{x}_i) \, , \qquad (\text{A9})
$$

$$
p(X \mid \boldsymbol{y}) \; = \; p(\boldsymbol{x}_1 \mid \boldsymbol{y}) \prod_{i=2}^{N} p(\boldsymbol{x}_i) \, , \qquad (\text{A10})
$$

$$
p(X) \; = \; \prod_{i=1}^{N} p(\boldsymbol{x}_i) \, . \qquad (\text{A11})
$$

Next, we present a theorem that shows this bound, where we largely follow Poole et al. (2019) in the proof. In contrast to Poole et al. (2019), we do not use the NWJ bound Nguyen et al. (2010). The mutual information is

$$\mathrm{I}(X_1 \; ; \; Y) \; = \; \mathrm{E}_{p(\boldsymbol{x}_1, \boldsymbol{y})} \left[ \ln \left( \frac{p(\boldsymbol{x}_1 \mid \boldsymbol{y})}{p(\boldsymbol{x}_1)} \right) \right] \, . \tag{A12}$$

**Theorem A1** (InfoNCE lower bound). *InfoNCE with score function $f(\boldsymbol{x}, \boldsymbol{y})$ according to Eq. (A3) is a lower bound on the mutual information.*

$$\mathrm{I}(X_1 \; ; \; Y) \; \geq \; \mathrm{E}_{p(\boldsymbol{y})p(X|\boldsymbol{y})} \left[ \ln \left( \frac{f(\boldsymbol{x}_1, \boldsymbol{y})}{\frac{1}{N} \sum_{i=1}^{N} f(\boldsymbol{x}_i, \boldsymbol{y})} \right) \right] \; = \; \mathrm{I}_{\mathrm{InfoNCE}}(X_1 \; ; \; Y) \, . \tag{A13}$$

*InfoNCE with probabilities according to Eq. (A2) is a lower bound on the mutual information.*

$$\mathrm{I}(X_1 \; ; \; Y) \; \geq \; \mathrm{E}_{p(\boldsymbol{y})p(X|\boldsymbol{y})} \left[ \ln \left( \frac{p(\boldsymbol{y} \mid \boldsymbol{x}_1)}{\frac{1}{N} \sum_{i=1}^{N} p(\boldsymbol{y} \mid \boldsymbol{x}_i)} \right) \right] \; = \; \mathrm{I}_{\mathrm{InfoNCE}}(X_1 \; ; \; Y) \, . \tag{A14}$$

*The second bound Eq. (A14) is a special case of the first bound Eq. (A13).*

*Proof.* **Part (I)**: Lower bound with score function $f(\boldsymbol{x}, \boldsymbol{y})$.

For each set $\tilde{X} = \{\boldsymbol{x}_2, \ldots, \boldsymbol{x}_N\}$, we define as data-dependent (depending on $\tilde{X}$) score function $g(\boldsymbol{x}_1, \boldsymbol{y}, \tilde{X})$ that is based on the score function $f(\boldsymbol{x}, \boldsymbol{y})$. Therefore we have for each $\tilde{X}$ a different data-dependent score function $g$ based on $f$. We will derive a bound on the InfoNCE, which is the expectation of a lower bond on the mutual information over the score functions. For score function $g(\boldsymbol{x}_1, \boldsymbol{y}, \tilde{X})$, we define a variational distribution $q(\boldsymbol{x}_1 \mid \boldsymbol{y}, \tilde{X})$ over $\boldsymbol{x}_1$:

$$q(\boldsymbol{x}_1 \mid \boldsymbol{y}, \tilde{X}) \; = \; \frac{p(\boldsymbol{x}_1) \, g(\boldsymbol{x}_1, \boldsymbol{y}, \tilde{X})}{Z(\boldsymbol{y}, \tilde{X})} \, , \tag{A15}$$

$$Z(\boldsymbol{y}, \tilde{X}) \; = \; \mathrm{E}_{p(\boldsymbol{x}_1)} \left[ g(\boldsymbol{x}_1, \boldsymbol{y}, \tilde{X}) \right] \, , \tag{A16}$$

which ensures

$$\int q(\boldsymbol{x}_1 \mid \boldsymbol{y}, \tilde{X}) \, \mathrm{d}\boldsymbol{x}_1 \; = \; 1 \, . \tag{A17}$$

We have

$$\frac{q(\boldsymbol{x}_1 \mid \boldsymbol{y}, \tilde{X})}{p(\boldsymbol{x}_1)} \; = \; \frac{g(\boldsymbol{x}_1, \boldsymbol{y}, \tilde{X})}{Z(\boldsymbol{y}, \tilde{X})} \, . \tag{A18}$$

For the function $g$, we set

$$g(\boldsymbol{x}_1, \boldsymbol{y}, \tilde{X}) \; = \; \frac{f(\boldsymbol{x}_1, \boldsymbol{y})}{\frac{1}{N} \sum_{i=1}^{N} f(\boldsymbol{x}_i, \boldsymbol{y})} \, , \tag{A19}$$

For the function $f$ we use

$$f(\boldsymbol{x}_1, \boldsymbol{y}) \; = \; \exp(\tau^{-1} \operatorname{sim}(\boldsymbol{x}_1, \boldsymbol{y})) \, , \tag{A20}$$

where $\operatorname{sim}(\boldsymbol{x}, \boldsymbol{y})$ is typically the cosine similarity.

We next show that InfoNCE is a lower bound on the mutual information.

$$\mathrm{I}(X_1 \; ; \; Y) \; = \; \mathrm{E}_{p(\tilde{X})}\left[\mathrm{I}(X_1 \; ; \; Y)\right] \; = \; \mathrm{E}_{p(\tilde{X})}\left[\mathrm{E}_{p(\boldsymbol{x}_1,\boldsymbol{y})}\left[\ln \frac{p(\boldsymbol{x}_1 \mid \boldsymbol{y})}{p(\boldsymbol{x}_1)}\right]\right] \tag{A21}$$

$$= \; \mathrm{E}_{p(\tilde{X})}\left[\mathrm{E}_{p(\boldsymbol{x}_1,\boldsymbol{y})}\left[\ln\left(\frac{p(\boldsymbol{x}_1 \mid \boldsymbol{y})}{q(\boldsymbol{x}_1 \mid \boldsymbol{y}, \tilde{X})} \; \frac{q(\boldsymbol{x}_1 \mid \boldsymbol{y}, \tilde{X})}{p(\boldsymbol{x}_1)}\right)\right]\right]$$

$$= \; \mathrm{E}_{p(\tilde{X})}\left[\mathrm{E}_{p(\boldsymbol{x}_1,\boldsymbol{y})}\left[\ln \frac{q(\boldsymbol{x}_1 \mid \boldsymbol{y}, \tilde{X})}{p(\boldsymbol{x}_1)}\right] \; + \; \mathrm{E}_{p(\boldsymbol{y})}\left[\mathrm{KL}(p(\boldsymbol{x}_1 \mid \boldsymbol{y}) \parallel q(\boldsymbol{x}_1 \mid \boldsymbol{y}, \tilde{X}))\right]\right]$$

$$\geq \; \mathrm{E}_{p(\tilde{X})}\left[\mathrm{E}_{p(\boldsymbol{x}_1,\boldsymbol{y})}\left[\ln \frac{q(\boldsymbol{x}_1 \mid \boldsymbol{y}, \tilde{X})}{p(\boldsymbol{x}_1)}\right]\right] \; = \; \mathrm{E}_{p(\tilde{X})}\left[\mathrm{E}_{p(\boldsymbol{x}_1,\boldsymbol{y})}\left[\ln \frac{g(\boldsymbol{x}_1, \boldsymbol{y}, \tilde{X})}{Z(\boldsymbol{y}, \tilde{X})}\right]\right]$$

$$= \; \mathrm{E}_{p(\tilde{X})}\left[\mathrm{E}_{p(\boldsymbol{x}_1,\boldsymbol{y})}\left[\ln g(\boldsymbol{x}_1, \boldsymbol{y}, \tilde{X}) \; - \; \ln\left(\mathrm{E}_{p(\boldsymbol{x}_1)}\left[g(\boldsymbol{x}_1, \boldsymbol{y}, \tilde{X})\right]\right)\right]\right]$$

$$= \; \mathrm{E}_{p(\tilde{X})}\left[\mathrm{E}_{p(\boldsymbol{y})}\left[\mathrm{E}_{p(\boldsymbol{x}_1 \mid \boldsymbol{y})}\left[\ln g(\boldsymbol{x}_1, \boldsymbol{y}, \tilde{X})\right] \; - \; \ln\left(\mathrm{E}_{p(\boldsymbol{x}_1)}\left[g(\boldsymbol{x}_1, \boldsymbol{y}, \tilde{X})\right]\right)\right]\right]$$

$$= \; \mathrm{E}_{p(\tilde{X})}\left[\mathrm{E}_{p(\boldsymbol{y})}\left[\mathrm{E}_{p(\boldsymbol{x}_1 \mid \boldsymbol{y})}\left[\ln g(\boldsymbol{x}_1, \boldsymbol{y}, \tilde{X})\right]\right]\right] \; - \; \mathrm{E}_{p(\tilde{X})}\left[\mathrm{E}_{p(\boldsymbol{y})}\left[\ln\left(\mathrm{E}_{p(\boldsymbol{x}_1)}\left[g(\boldsymbol{x}_1, \boldsymbol{y}, \tilde{X})\right]\right)\right]\right]$$

$$\geq \; \mathrm{E}_{p(\boldsymbol{y})p(X \mid \boldsymbol{y})}\left[\ln g(\boldsymbol{x}_1, \boldsymbol{y}, \tilde{X})\right] \; - \; \mathrm{E}_{p(\tilde{X})}\left[\mathrm{E}_{p(\boldsymbol{y})}\left[\mathrm{E}_{p(\boldsymbol{x}_1)}\left[g(\boldsymbol{x}_1, \boldsymbol{y}, \tilde{X})\right] \; - \; 1\right]\right]$$

$$= \; \mathrm{E}_{p(\boldsymbol{y})p(X \mid \boldsymbol{y})}\left[\ln \frac{f(\boldsymbol{x}_1, \boldsymbol{y})}{\frac{1}{N}\sum_{i=1}^{N} f(\boldsymbol{x}_i, \boldsymbol{y})}\right] \; - \; \mathrm{E}_{p(\boldsymbol{y})}\left[\mathrm{E}_{p(X)}\left[\frac{f(\boldsymbol{x}_1, \boldsymbol{y})}{\frac{1}{N}\sum_{i=1}^{N} f(\boldsymbol{x}_i, \boldsymbol{y})}\right] - 1\right]$$

$$= \; \mathrm{E}_{p(\boldsymbol{y})p(X \mid \boldsymbol{y})}\left[\ln \frac{f(\boldsymbol{x}_1, \boldsymbol{y})}{\frac{1}{N}\sum_{i=1}^{N} f(\boldsymbol{x}_i, \boldsymbol{y})}\right] \; - \; \mathrm{E}_{p(\boldsymbol{y})}\left[\frac{1}{N}\sum_{i=1}^{N}\mathrm{E}_{p(X)}\left[\frac{f(\boldsymbol{x}_i, \boldsymbol{y})}{\frac{1}{N}\sum_{i=1}^{N} f(\boldsymbol{x}_i, \boldsymbol{y})}\right] - 1\right]$$

$$= \; \mathrm{E}_{p(\boldsymbol{y})p(X \mid \boldsymbol{y})}\left[\ln \frac{f(\boldsymbol{x}_1, \boldsymbol{y})}{\frac{1}{N}\sum_{i=1}^{N} f(\boldsymbol{x}_i, \boldsymbol{y})}\right] \; - \; \mathrm{E}_{p(\boldsymbol{y})}\left[\mathrm{E}_{p(X)}\left[\frac{\frac{1}{N}\sum_{i=1}^{N} f(\boldsymbol{x}_i, \boldsymbol{y})}{\frac{1}{N}\sum_{i=1}^{N} f(\boldsymbol{x}_i, \boldsymbol{y})}\right] - 1\right]$$

$$= \; \mathrm{E}_{p(\boldsymbol{y})p(X \mid \boldsymbol{y})}\left[\ln \frac{f(\boldsymbol{x}_1, \boldsymbol{y})}{\frac{1}{N}\sum_{i=1}^{N} f(\boldsymbol{x}_i, \boldsymbol{y})}\right]$$

$$= \; \mathrm{I}_{\mathrm{InfoNCE}}(X_1 \; ; \; Y) \, .$$

For the first "$\geq$" we used that the Kullback-Leibler divergence is non-negative. For the second "$\geq$" we used the inequality $\ln a \leqslant a - 1$ for $a > 0$.

**Part (II)**: Lower bound with probabilities.

If the score function $f$ is

$$f(\boldsymbol{x}, \boldsymbol{y}) \; = \; p(\boldsymbol{y} \mid \boldsymbol{x}) \, , \tag{A22}$$

then the bound is

$$\mathrm{I}(X_1 \; ; \; Y) \; \geq \; \mathrm{E}_{p(\boldsymbol{y})p(X \mid \boldsymbol{y})}\left[\ln\left(\frac{f(\boldsymbol{x}_1, \boldsymbol{y})}{\frac{1}{N}\sum_{i=1}^{N} f(\boldsymbol{x}_i, \boldsymbol{y})}\right)\right] \; = \; \mathrm{E}_{p(\boldsymbol{y})p(X \mid \boldsymbol{y})}\left[\ln\left(\frac{p(\boldsymbol{y} \mid \boldsymbol{x}_1)}{\frac{1}{N}\sum_{i=1}^{N} p(\boldsymbol{y} \mid \boldsymbol{x}_i)}\right)\right] \tag{A23}$$

$$= \; \mathrm{E}_{p(\boldsymbol{y})p(X \mid \boldsymbol{y})}\left[\ln\left(\frac{\frac{p(\boldsymbol{y} \mid \boldsymbol{x}_1)}{p(\boldsymbol{y})}}{\frac{1}{N}\sum_{i=1}^{N} \frac{p(\boldsymbol{y} \mid \boldsymbol{x}_i)}{p(\boldsymbol{y})}}\right)\right] \; = \; \mathrm{I}_{\mathrm{InfoNCE}}(X_1 \; ; \; Y) \, .$$

This is the bound with probabilities in the theorem. $\qquad\square$

### A.1.2  INFOLOOB: UPPER BOUND ON MUTUAL INFORMATION

We derive an upper bound on the mutual information between random variables $X$ and $Y$ distributed according to $p(\boldsymbol{x}, \boldsymbol{y})$. The mutual information $\mathrm{I}(X \; ; \; Y)$ between random variables $X$ and $Y$ is

$$\mathrm{I}(X \; ; \; Y) \; = \; \mathrm{E}_{p(\boldsymbol{x},\boldsymbol{y})}\left[\ln \frac{p(\boldsymbol{x}, \boldsymbol{y})}{p(\boldsymbol{x})\, p(\boldsymbol{y})}\right] \; = \; \mathrm{E}_{p(\boldsymbol{x},\boldsymbol{y})}\left[\ln \frac{p(\boldsymbol{x} \mid \boldsymbol{y})}{p(\boldsymbol{x})}\right] \; = \; \mathrm{E}_{p(\boldsymbol{x},\boldsymbol{y})}\left[\ln \frac{p(\boldsymbol{y} \mid \boldsymbol{x})}{p(\boldsymbol{y})}\right] \; . \tag{A24}$$

In Poole et al. (2019) Eq. (13) introduces a variational upper bound on the mutual information, which has been called "Leave one out upper bound" (called "L1Out" in Cheng et al. (2020)). For simplicity, we call this bound "InfoLOOB", where LOOB is an acronym for "Leave One Out Bound". In contrast to InfoNCE, InfoLOOB is an upper bound on the mutual information. InfoLOOB is analog to InfoNCE except that the negative samples do not contain a positive sample. Fig. 1 and Fig. 2 in Cheng et al. (2020) both show that InfoLOOB is a better estimator for the mutual information than InfoNCE (van den Oord et al., 2018), MINE (Belghazi et al., 2018), and NWJ (Nguyen et al., 2010).

The InfoLOOB with score function $f(\boldsymbol{x}, \boldsymbol{y})$ is defined as

$$\mathrm{I}_{\mathrm{InfoLOOB}}(X_1 \; ; \; Y) \; = \; \mathrm{E}_{p(\boldsymbol{y})}\left[\mathrm{E}_{\tilde{p}(X|\boldsymbol{y})}\left[\ln\left(\frac{f(\boldsymbol{x}_1, \boldsymbol{y})}{\frac{1}{N-1}\sum_{i=2}^{N} f(\boldsymbol{x}_i, \boldsymbol{y})}\right)\right]\right] \; . \tag{A25}$$

The InfoLOOB with probabilities is defined as

$$\mathrm{I}_{\mathrm{InfoLOOB}}(X_1 \; ; \; Y) \; = \; \mathrm{E}_{p(\boldsymbol{y})}\left[\mathrm{E}_{p(X|\boldsymbol{y})}\left[\ln\left(\frac{p(\boldsymbol{y} \mid \boldsymbol{x}_1)}{\frac{1}{N-1}\sum_{i=2}^{N} p(\boldsymbol{y} \mid \boldsymbol{x}_i)}\right)\right]\right] \; . \tag{A26}$$

This is the InfoLOOB with $f(\boldsymbol{x}, \boldsymbol{y}) = p(\boldsymbol{y} \mid \boldsymbol{x})$.

The InfoLOOB with probabilities can be written in different forms:

$$\mathrm{I}_{\mathrm{InfoLOOB}}(X_1 \; ; \; Y) \; = \; \mathrm{E}_{p(\boldsymbol{y})}\left[\mathrm{E}_{p(X|\boldsymbol{y})}\left[\ln\left(\frac{p(\boldsymbol{y} \mid \boldsymbol{x}_1)}{\frac{1}{N-1}\sum_{i=2}^{N} p(\boldsymbol{y} \mid \boldsymbol{x}_i)}\right)\right]\right] \tag{A27}$$

$$= \; \mathrm{E}_{p(\boldsymbol{y})}\left[\mathrm{E}_{p(X|\boldsymbol{y})}\left[\ln\left(\frac{\frac{p(\boldsymbol{y}|\boldsymbol{x}_1)}{p(\boldsymbol{y})}}{\frac{1}{N-1}\sum_{i=2}^{N}\frac{p(\boldsymbol{y}|\boldsymbol{x}_i)}{p(\boldsymbol{y})}}\right)\right]\right] \; = \; \mathrm{E}_{p(\boldsymbol{y})}\left[\mathrm{E}_{p(X|\boldsymbol{y})}\left[\ln\left(\frac{\frac{p(\boldsymbol{x}_1|\boldsymbol{y})}{p(\boldsymbol{x}_1)}}{\frac{1}{N-1}\sum_{i=2}^{N}\frac{p(\boldsymbol{x}_i|\boldsymbol{y})}{p(\boldsymbol{x}_i)}}\right)\right]\right] \; .$$

**Set of pairs.** The InfoLOOB can we written in a different setting (Poole et al., 2019), which will be used in our implementations. We sample $N$ pairs independently from $p(\boldsymbol{x}, \boldsymbol{y})$, which gives $X = \{(\boldsymbol{x}_1, \boldsymbol{y}_1), (\boldsymbol{x}_2, \boldsymbol{y}_2), \ldots, (\boldsymbol{x}_N, \boldsymbol{y}_N)\}$. The InfoLOOB is then

$$\mathrm{I}_{\mathrm{InfoLOOB}}(X \; ; \; Y) \; = \; \mathrm{E}_{p(X|\boldsymbol{y})}\left[\frac{1}{N}\sum_{i=1}^{N}\ln\left(\frac{f(\boldsymbol{x}_i, \boldsymbol{y}_i)}{\frac{1}{N-1}\sum_{j=1,j\neq i}^{N} f(\boldsymbol{x}_j, \boldsymbol{y}_i)}\right)\right] \; . \tag{A28}$$

We assume that an anchor sample $\boldsymbol{y}$ is given. For the anchor sample $\boldsymbol{y}$ we draw a positive sample $\boldsymbol{x}_1$ according to $p(\boldsymbol{x}_1 \mid \boldsymbol{y})$. Next, we draw a set $\tilde{X} = \{\boldsymbol{x}_2, \ldots, \boldsymbol{x}_N\}$ of negative samples according to $\tilde{p}(\boldsymbol{x} \mid \boldsymbol{y})$. **For a given $\boldsymbol{y}$, the $\boldsymbol{x}$ that have a large $p(\boldsymbol{x} \mid \boldsymbol{y})$ are drawn with a lower probability $\tilde{p}(\boldsymbol{x} \mid \boldsymbol{y})$ compared to random drawing via $p(\boldsymbol{x})$.** The negatives are indeed negatives. We have drawn first anchor sample $\boldsymbol{y}$ and then $X = \{\boldsymbol{x}_1, \ldots, \boldsymbol{x}_N\}$, where $\boldsymbol{x}_1$ is drawn according to $p(\boldsymbol{x}_1 \mid \boldsymbol{y})$ and $\tilde{X} = \{\boldsymbol{x}_2, \ldots, \boldsymbol{x}_N\}$ are drawn iid according to $\tilde{p}(\boldsymbol{x} \mid \boldsymbol{y})$. We have

$$\tilde{p}(\tilde{X} \mid \boldsymbol{y}) \; = \; \prod_{i=2}^{N}\tilde{p}(\boldsymbol{x}_i \mid \boldsymbol{y}) \; , \tag{A29}$$

$$\tilde{p}(X \mid \boldsymbol{y}) \; = \; p(\boldsymbol{x}_1 \mid \boldsymbol{y})\prod_{i=2}^{N}\tilde{p}(\boldsymbol{x}_i \mid \boldsymbol{y}) \; , \tag{A30}$$

$$\tilde{p}(\tilde{X} \mid \boldsymbol{y})\, p(\boldsymbol{x}_1) \; = \; p(\boldsymbol{x}_1)\prod_{i=2}^{N}\tilde{p}(\boldsymbol{x}_i \mid \boldsymbol{y}) \; . \tag{A31}$$

We assume for score function $f(\boldsymbol{x}, \boldsymbol{y})$

$$\forall_{\boldsymbol{y}} \forall_{\boldsymbol{x}} : \ 0 \ < \ f(\boldsymbol{x}, \boldsymbol{y}) \ . \tag{A32}$$

We ensure this by using for score function $f$

$$f(\boldsymbol{x}, \boldsymbol{y}) \ = \ \exp(\tau^{-1} \operatorname{sim}(\boldsymbol{x}, \boldsymbol{y})) \ , \tag{A33}$$

where $\operatorname{sim}(\boldsymbol{x}, \boldsymbol{y})$ is typically the cosine similarity.

InfoLOOB with score function $f(\boldsymbol{x}, \boldsymbol{y})$ is

$$\mathrm{I}_{\mathrm{InfoLOOB}}(X \ ; \ Y) \ = \ \mathrm{E}_{p(\boldsymbol{y})} \left[ \mathrm{E}_{p(X|\boldsymbol{y})} \left[ \ln \left( \frac{f(\boldsymbol{x}_1, \boldsymbol{y})}{\frac{1}{N-1} \sum_{i=2}^{N} f(\boldsymbol{x}_i, \boldsymbol{y})} \right) \right] \right] \ . \tag{A34}$$

The reference constant $Z(\boldsymbol{y})$ gives the average score $f(\boldsymbol{x}, \boldsymbol{y})$, if the negatives for $\boldsymbol{y}$ are selected with lower probability via $\tilde{p}(\boldsymbol{x} \mid \boldsymbol{y})$ than with random drawing according to $p(\boldsymbol{x})$.

$$Z(\boldsymbol{y}) \ = \ \mathrm{E}_{\tilde{p}(\boldsymbol{x}|\boldsymbol{y})} \left[ f(\boldsymbol{x}, \boldsymbol{y}) \right] \ . \tag{A35}$$

We define the variational distribution

$$q(\boldsymbol{x} \mid \boldsymbol{y}) \ = \ \frac{p(\boldsymbol{x}) \, f(\boldsymbol{x}, \boldsymbol{y})}{Z^*(\boldsymbol{y})} \ , \quad Z^*(\boldsymbol{y}) \ = \ \mathrm{E}_{p(\boldsymbol{x})} \left[ f(\boldsymbol{x}, \boldsymbol{y}) \right] \ . \tag{A36}$$

With the variational distribution $q(\boldsymbol{x} \mid \boldsymbol{y})$, we express our main assumption. **The main assumption for the bound is:**

$$\mathrm{E}_{p(\boldsymbol{y})} \left[ \mathrm{KL}(p(\boldsymbol{x} \mid \boldsymbol{y}) \parallel q(\boldsymbol{x} \mid \boldsymbol{y})) \right] \ \leqslant \ \mathrm{E}_{p(\boldsymbol{y})} \left[ \ln Z^*(\boldsymbol{y}) \ - \ \ln Z(\boldsymbol{y}) \right] \ . \tag{A37}$$

This assumption can be written as

$$\mathrm{E}_{p(\boldsymbol{y})} \left[ \mathrm{E}_{p(\boldsymbol{x}|\boldsymbol{y})} \left[ \ln \left( \frac{p(\boldsymbol{y} \mid \boldsymbol{x}) \, Z(\boldsymbol{y})}{p(\boldsymbol{y}) \, f(\boldsymbol{x}, \boldsymbol{y})} \right) \right] \right] \ \leqslant \ 0 \ . \tag{A38}$$

This assumption ensures that the $\boldsymbol{x}$ with large $p(\boldsymbol{x} \mid \boldsymbol{y}))$ are selected with lower probability via $\tilde{p}(\boldsymbol{x} \mid \boldsymbol{y})$ than with random drawing according to $p(\boldsymbol{x})$. The negatives are ensured to be real negatives, that is, $p(\boldsymbol{x} \mid \boldsymbol{y})$ is small and so is $f(\boldsymbol{x}, \boldsymbol{y})$. Consequently, we make sure that we draw $\boldsymbol{x}$ with sufficient small $f(\boldsymbol{x}, \boldsymbol{y})$. The Kullback-Leibler gives the minimal required gap between drawing $f(\boldsymbol{x}, \boldsymbol{y})$ via $p(\boldsymbol{x})$ and drawing $f(\boldsymbol{x}, \boldsymbol{y})$ via $\tilde{p}(\boldsymbol{x} \mid \boldsymbol{y})$.

**EXAMPLE.** With $h(\boldsymbol{y}) > 0$, we consider the setting

$$f(\boldsymbol{x}, \boldsymbol{y}) \ = \ \frac{p(\boldsymbol{y} \mid \boldsymbol{x}) \, h(\boldsymbol{y})}{p(\boldsymbol{y})} \ , \tag{A39}$$

$$\tilde{p}(\boldsymbol{x} \mid \boldsymbol{y}) \ = \ \frac{p(\boldsymbol{x}) \, p(\boldsymbol{y})}{h(\boldsymbol{y}) \, p(\boldsymbol{y} \mid \boldsymbol{x}) \, C(\boldsymbol{y})} \ , \quad C(\boldsymbol{y}) \ = \ \mathrm{E}_{p(\boldsymbol{x})} \left[ \left( \frac{p(\boldsymbol{y} \mid \boldsymbol{x}) \, h(\boldsymbol{y})}{p(\boldsymbol{y})} \right)^{-1} \right] \ . \tag{A40}$$

The main assumption becomes

$$\mathrm{E}_{p(\boldsymbol{y})} \left[ \mathrm{E}_{p(\boldsymbol{x}|\boldsymbol{y})} \left[ \ln \frac{Z(\boldsymbol{y})}{h(\boldsymbol{y})} \right] \right] \ \leqslant \ 0 \ . \tag{A41}$$

The main assumption holds since

$$Z(\boldsymbol{y}) \ = \ \mathrm{E}_{\tilde{p}(\boldsymbol{x}|\boldsymbol{y})} \left[ \frac{p(\boldsymbol{y} \mid \boldsymbol{x}) \, h(\boldsymbol{y})}{p(\boldsymbol{y})} \right] \ = \ \int \frac{p(\boldsymbol{x}) \, p(\boldsymbol{y})}{h(\boldsymbol{y}) \, p(\boldsymbol{y} \mid \boldsymbol{x}) \, C(\boldsymbol{y})} \, \frac{p(\boldsymbol{y} \mid \boldsymbol{x}) \, h(\boldsymbol{y})}{p(\boldsymbol{y})} \, \mathrm{d}\boldsymbol{x} \tag{A42}$$

$$= \ \int p(\boldsymbol{x}) \, C(\boldsymbol{y})^{-1} \, \mathrm{d}\boldsymbol{x} \ = \ C(\boldsymbol{y})^{-1} \ = \ \left( \mathrm{E}_{p(\boldsymbol{x})} \left[ \left( \frac{p(\boldsymbol{y} \mid \boldsymbol{x}) \, h(\boldsymbol{y})}{p(\boldsymbol{y})} \right)^{-1} \right] \right)^{-1}$$

$$\leqslant \ \left( \mathrm{E}_{p(\boldsymbol{x})} \left[ \frac{p(\boldsymbol{y} \mid \boldsymbol{x}) \, h(\boldsymbol{y})}{p(\boldsymbol{y})} \right]^{-1} \right)^{-1} \ = \ \mathrm{E}_{p(\boldsymbol{x})} \left[ \frac{p(\boldsymbol{y} \mid \boldsymbol{x}) \, h(\boldsymbol{y})}{p(\boldsymbol{y})} \right]$$

$$= \ \int \frac{p(\boldsymbol{y}, \boldsymbol{x}) \, h(\boldsymbol{y})}{p(\boldsymbol{y})} \, \mathrm{d}\boldsymbol{x} \ = \ h(\boldsymbol{y}) \ ,$$

where we used for the $\leqslant$ Jensen's inequality with the function $f(a) = 1/a$, which is convex for $a > 0$.

For score function $f(\boldsymbol{x}, \boldsymbol{y})$ and distribution $\tilde{p}(\boldsymbol{x} \mid \boldsymbol{y})$ for sampling the negative samples, we have defined:

$$Z(\boldsymbol{y}) = \mathrm{E}_{\tilde{p}(\boldsymbol{x}|\boldsymbol{y})}[f(\boldsymbol{x}, \boldsymbol{y})] \ , \tag{A43}$$

$$Z^*(\boldsymbol{y}) = \mathrm{E}_{p(\boldsymbol{x})}[f(\boldsymbol{x}, \boldsymbol{y})] \ , \tag{A44}$$

$$q(\boldsymbol{x} \mid \boldsymbol{y}) = \frac{p(\boldsymbol{x}) \ f(\boldsymbol{x}, \boldsymbol{y})}{Z^*(\boldsymbol{y})} \ . \tag{A45}$$

Next theorem gives the upper bound of the InfoLOOB on the mutual information, which is

$$\mathrm{I}(X_1 \ ; \ Y) = \mathrm{E}_{p(\boldsymbol{x}_1, \boldsymbol{y})}\left[\ln \frac{p(\boldsymbol{x}_1 \mid \boldsymbol{y})}{p(\boldsymbol{x}_1)}\right] \ . \tag{A46}$$

**Theorem A2** (InfoLOOB upper bound). *If $\tilde{X} = \{\boldsymbol{x}_2, \dots, \boldsymbol{x}_N\}$ are drawn iid according to $\tilde{p}(\boldsymbol{x} \mid \boldsymbol{y})$ and if the main assumption holds:*

$$\mathrm{E}_{p(\boldsymbol{y})}\left[\mathrm{KL}(p(\boldsymbol{x} \mid \boldsymbol{y}) \ \| \ q(\boldsymbol{x} \mid \boldsymbol{y}))\right] \ \leqslant \ \mathrm{E}_{p(\boldsymbol{y})}\left[\ln Z^*(\boldsymbol{y}) \ - \ \ln Z(\boldsymbol{y})\right] \ . \tag{A47}$$

*Then InfoLOOB with score function $f(\boldsymbol{x}, \boldsymbol{y})$ as in Eq. (A25) is an upper bound on the mutual information:*

$$\mathrm{I}(X_1 \ ; \ Y) \ \leqslant \ \mathrm{E}_{p(\boldsymbol{y})}\left[\mathrm{E}_{\tilde{p}(X|\boldsymbol{y})}\left[\ln\left(\frac{f(\boldsymbol{x}_1, \boldsymbol{y})}{\frac{1}{N-1}\sum_{i=2}^{N} f(\boldsymbol{x}_i, \boldsymbol{y})}\right)\right]\right] = \mathrm{I}_{\mathrm{InfoLOOB}}(X_1 \ ; \ Y) \ . \tag{A48}$$

*If the negative samples $\tilde{X} = \{\boldsymbol{x}_2, \dots, \boldsymbol{x}_N\}$ are drawn iid according to $p(\boldsymbol{x})$, then InfoLOOB with probabilities according to Eq. (A26) is an upper bound on the mutual information:*

$$\mathrm{I}(X_1 \ ; \ Y) \ \leqslant \ \mathrm{E}_{p(\boldsymbol{y})}\left[\mathrm{E}_{p(X|\boldsymbol{y})}\left[\ln\left(\frac{p(\boldsymbol{y} \mid \boldsymbol{x}_1)}{\frac{1}{N-1}\sum_{i=2}^{N} p(\boldsymbol{y} \mid \boldsymbol{X}_i)}\right)\right]\right] = \mathrm{I}_{\mathrm{InfoLOOB}}(X_1 \ ; \ Y) \ . \tag{A49}$$

*The second bound Eq. (A49) is a special case of the first bound Eq. (A48).*

*Proof.* **Part (I)**: Upper bound with score function $f(\boldsymbol{x}, \boldsymbol{y})$.

$$I(X_1 ; Y) = E_{p(\boldsymbol{x}_1, \boldsymbol{y})} \left[ \ln \frac{p(\boldsymbol{x}_1 \mid \boldsymbol{y})}{p(\boldsymbol{x}_1)} \right] \tag{A50}$$

$$= E_{p(\boldsymbol{x}_1, \boldsymbol{y})} \left[ \ln \left( \frac{p(\boldsymbol{x}_1 \mid \boldsymbol{y})}{q(\boldsymbol{x}_1 \mid \boldsymbol{y})} \frac{q(\boldsymbol{x}_1 \mid \boldsymbol{y})}{p(\boldsymbol{x}_1)} \right) \right]$$

$$= E_{p(\boldsymbol{x}_1, \boldsymbol{y})} \left[ \ln \frac{q(\boldsymbol{x}_1 \mid \boldsymbol{y})}{p(\boldsymbol{x}_1)} \right] + E_{p(\boldsymbol{y})} \left[ KL(p(\boldsymbol{x}_1 \mid \boldsymbol{y}) \parallel q(\boldsymbol{x}_1 \mid \boldsymbol{y})) \right]$$

$$\leqslant E_{p(\boldsymbol{x}_1, \boldsymbol{y})} \left[ \ln \frac{q(\boldsymbol{x}_1 \mid \boldsymbol{y})}{p(\boldsymbol{x}_1)} \right] + E_{p(\boldsymbol{y})} \left[ \ln E_{p(\boldsymbol{x}_1)} [f(\boldsymbol{x}_1, \boldsymbol{y})] - \ln Z(\boldsymbol{y}) \right]$$

$$= E_{p(\boldsymbol{x}_1, \boldsymbol{y})} \left[ \ln \frac{q(\boldsymbol{x}_1 \mid \boldsymbol{y})}{p(\boldsymbol{x}_1)} + \ln \frac{E_{p(\boldsymbol{x}_1)} [f(\boldsymbol{x}_1, \boldsymbol{y})]}{Z(\boldsymbol{y})} \right]$$

$$= E_{p(\boldsymbol{x}_1, \boldsymbol{y})} \left[ \ln \left( \frac{f(\boldsymbol{x}_1, \boldsymbol{y})}{E_{p(\boldsymbol{x}_1)} [f(\boldsymbol{x}_1, \boldsymbol{y})]} \frac{E_{p(\boldsymbol{x}_1)} [f(\boldsymbol{x}_1, \boldsymbol{y})]}{Z(\boldsymbol{y})} \right) \right]$$

$$= E_{p(\boldsymbol{x}_1, \boldsymbol{y})} \left[ \ln \frac{f(\boldsymbol{x}_1, \boldsymbol{y})}{Z(\boldsymbol{y})} \right]$$

$$= E_{p(\boldsymbol{x}_1, \boldsymbol{y})} \left[ \ln \left( \frac{f(\boldsymbol{x}_1, \boldsymbol{y})}{E_{\tilde{p}(X \mid \boldsymbol{y})} \left[ \frac{1}{N-1} \sum_{i=2}^{N} f(\boldsymbol{x}_i, \boldsymbol{y}) \right]} \right) \right]$$

$$= E_{p(\boldsymbol{x}_1, \boldsymbol{y})} [\ln f(\boldsymbol{x}_1, \boldsymbol{y})] - E_{p(\boldsymbol{y})} \left[ \ln \left( E_{\tilde{p}(X \mid \boldsymbol{y})} \left[ \frac{1}{N-1} \sum_{i=2}^{N} f(\boldsymbol{x}_i, \boldsymbol{y}) \right] \right) \right]$$

$$\leqslant E_{p(\boldsymbol{x}_1, \boldsymbol{y})} [\ln f(\boldsymbol{x}_1, \boldsymbol{y})] - E_{p(\boldsymbol{y})} \left[ E_{\tilde{p}(X \mid \boldsymbol{y})} \left[ \ln \left( \frac{1}{N-1} \sum_{i=2}^{N} f(\boldsymbol{x}_i, \boldsymbol{y}) \right) \right] \right]$$

$$= E_{p(\boldsymbol{y})} \left[ E_{\tilde{p}(X \mid \boldsymbol{y})} \left[ \ln \left( \frac{f(\boldsymbol{x}_1, \boldsymbol{y})}{\frac{1}{N-1} \sum_{i=2}^{N} f(\boldsymbol{x}_i, \boldsymbol{y})} \right) \right] \right]$$

$$= I_{\mathrm{InfoLOOB}}(X_1 ; Y),$$

where the first "$\leqslant$" uses assumption Eq. (A37), while Jensens's inequality was used for the second "$\leqslant$" by exchanging the expectation and the "ln". We also used

$$E_{\tilde{p}(X \mid \boldsymbol{y})} \left[ \frac{1}{N-1} \sum_{i=2}^{N} f(\boldsymbol{x}_i, \boldsymbol{y}) \right] = \frac{1}{N-1} \sum_{i=2}^{N} E_{\tilde{p}(\boldsymbol{x}_i \mid \boldsymbol{y})} [f(\boldsymbol{x}_i, \boldsymbol{y})] = \frac{1}{N-1} \sum_{i=2}^{N} Z(\boldsymbol{y}) = Z(\boldsymbol{y}). \tag{A51}$$

**Part (II)**: Upper bound with probabilities.

If the score function $f$ is

$$f(\boldsymbol{x}, \boldsymbol{y}) = p(\boldsymbol{y} \mid \boldsymbol{x}) \tag{A52}$$

and

$$\tilde{p}(\boldsymbol{x} \mid \boldsymbol{y}) = p(\boldsymbol{x}), \tag{A53}$$

then

$$\tilde{p}(X \mid \boldsymbol{y}) = p(X \mid \boldsymbol{y}), \tag{A54}$$

$$Z(\boldsymbol{y}) = E_{p(\boldsymbol{x})} [p(\boldsymbol{y} \mid \boldsymbol{x})] = p(\boldsymbol{y}), \tag{A55}$$

$$Z^*(\boldsymbol{y}) = E_{p(\boldsymbol{x})} [p(\boldsymbol{y} \mid \boldsymbol{x})] = p(\boldsymbol{y}), \tag{A56}$$

$$q(\boldsymbol{x} \mid \boldsymbol{y}) = \frac{p(\boldsymbol{x}) \, p(\boldsymbol{y} \mid \boldsymbol{x})}{p(\boldsymbol{y})} = p(\boldsymbol{x} \mid \boldsymbol{y}), \tag{A57}$$

$$KL(p(\boldsymbol{x} \mid \boldsymbol{y}) \parallel q(\boldsymbol{x} \mid \boldsymbol{y})) = KL(p(\boldsymbol{x} \mid \boldsymbol{y}) \parallel p(\boldsymbol{x} \mid \boldsymbol{y})) = 0. \tag{A58}$$

Therefore, the main assumption holds, since

$$0 \;=\; \mathrm{E}_{p(\boldsymbol{y})}\left[\mathrm{KL}(p(\boldsymbol{x} \mid \boldsymbol{y}) \parallel q(\boldsymbol{x} \mid \boldsymbol{y}))\right] \;=\; \mathrm{E}_{p(\boldsymbol{y})}\left[\ln Z^*(\boldsymbol{y}) \;-\; \ln Z(\boldsymbol{y})\right] . \tag{A59}$$

The bound becomes

$$\mathrm{I}(X_1 \;;\; Y) \;\leqslant\; \mathrm{E}_{p(\boldsymbol{y})}\left[\mathrm{E}_{p(X|\boldsymbol{y})}\left[\ln\left(\frac{p(\boldsymbol{y} \mid \boldsymbol{x}_1)}{\frac{1}{N-1}\sum_{i=2}^{N} p(\boldsymbol{y} \mid \boldsymbol{x}_i)}\right)\right]\right] \tag{A60}$$

$$= \; \mathrm{E}_{p(\boldsymbol{y})}\left[\mathrm{E}_{p(X|\boldsymbol{y})}\left[\ln\left(\frac{\frac{p(\boldsymbol{y}|\boldsymbol{x}_1)}{p(\boldsymbol{y})}}{\frac{1}{N-1}\sum_{i=2}^{N}\frac{p(\boldsymbol{y}|\boldsymbol{x}_i)}{p(\boldsymbol{y})}}\right)\right]\right] \;=\; \mathrm{I}_{\mathrm{InfoLOOB}}(X_1 \;;\; Y) .$$

An alternative proof is as follows:

$$\mathrm{I}(X_1 \;;\; Y) \;=\; \mathrm{I}(X_1 \;;\; Y) \;-\; \mathrm{E}_{p(\boldsymbol{y})}\left[\ln\left(\frac{1}{N-1}\sum_{i=2}^{N}\frac{p(\boldsymbol{y})}{p(\boldsymbol{y})}\right)\right] \tag{A61}$$

$$= \; \mathrm{I}(X_1 \;;\; Y) \;-\; \mathrm{E}_{p(\boldsymbol{y})}\left[\ln\left(\mathrm{E}_{p(X|\boldsymbol{y})}\left[\frac{1}{N-1}\sum_{i=2}^{N}\frac{p(\boldsymbol{y} \mid \boldsymbol{x}_i)}{p(\boldsymbol{y})}\right]\right)\right]$$

$$\leqslant \; \mathrm{I}(X_1 \;;\; Y) \;-\; \mathrm{E}_{p(\boldsymbol{y})}\left[\mathrm{E}_{p(X|\boldsymbol{y})}\left[\ln\left(\frac{1}{N-1}\sum_{i=2}^{N}\frac{p(\boldsymbol{y} \mid \boldsymbol{x}_i)}{p(\boldsymbol{y})}\right)\right]\right]$$

$$= \; \mathrm{E}_{p(\boldsymbol{y})}\left[\mathrm{E}_{p(\boldsymbol{x}_1|\boldsymbol{y})}\left[\ln\left(\frac{p(\boldsymbol{x}_1 \mid \boldsymbol{y})}{p(\boldsymbol{x}_1)}\right)\right]\right] \;-\; \mathrm{E}_{p(\boldsymbol{y})}\left[\mathrm{E}_{p(X|\boldsymbol{y})}\left[\ln\left(\frac{1}{N-1}\sum_{i=2}^{N}\frac{p(\boldsymbol{x}_i \mid \boldsymbol{y})}{p(\boldsymbol{x}_i)}\right)\right]\right]$$

$$= \; \mathrm{E}_{p(\boldsymbol{y})}\left[\mathrm{E}_{p(X|\boldsymbol{y})}\left[\ln\left(\frac{\frac{p(\boldsymbol{x}_1|\boldsymbol{y})}{p(\boldsymbol{x}_1)}}{\frac{1}{N-1}\sum_{i=2}^{N}\frac{p(\boldsymbol{x}_i|\boldsymbol{y})}{p(\boldsymbol{x}_i)}}\right)\right]\right]$$

$$= \; \mathrm{I}_{\mathrm{InfoLOOB}}(X_1 \;;\; Y) .$$

where we applied Jensens's inequality for the exchanging the expectation and the "ln" to obtain the "$\leqslant$" inequality.

$\square$

Experiments that compare upper and lower bounds as mutual information estimates are provided in Cheng et al. (2020) and in Poole et al. (2019). In Fig. 2 in Cheng et al. (2020) it is shown that InfoLOOB is a good estimator of the mutual information.

### A.1.3 INFOLOOB: ANALYSIS OF THE OBJECTIVE

This subsection justifies the maximization of the InfoLOOB bound for contrastive learning. Maximizing the InfoLOOB bound is not intuitive as it was introduced as an upper bound on the mutual information in the previous subsection. Still maximizing the InfoLOOB bound leads to a good approximation of the mutual information, in particular for high mutual information.

InfoLOOB with a neural network as a scoring function is not an upper bound on the mutual information when not under-sampling. As we use InfoLOOB on training data for which we do not know the sampling procedure, we cannot assume under-sampling. Therefore, we elaborate more on the rationale behind the maximization of the InfoLOOB bound. (I) We show that InfoLOOB with neural networks as scoring function is bounded from above. Therefore, there exists a maximum and the optimization problem is well defined. (II) We show that InfoLOOB with neural networks as scoring function differs by two terms the mutual information. The first term is the Kullback-Leibler divergence between the variational $q(\boldsymbol{x} \mid \boldsymbol{y})$ and the posterior $p(\boldsymbol{x} \mid \boldsymbol{y})$. This divergence is minimal for $q(\boldsymbol{x} \mid \boldsymbol{y}) = p(\boldsymbol{x} \mid \boldsymbol{y})$, which implies $f(\boldsymbol{y} \mid \boldsymbol{x}) = p(\boldsymbol{y} \mid \boldsymbol{x})$. The second term is governed by the difference between the mean $\mathrm{E}[f(\boldsymbol{x}, \boldsymbol{y})]$ and the empirical mean $1/(N-1)\sum_i f(\boldsymbol{x}, \boldsymbol{y})$. Hoeffding's inequality bounds this difference as we demonstrate in this subsection. Therefore, the second term

is negligible for large $N$. In contrast, the KL term is dominant and the relevant term, therefore maximizing InfoLOOB leads to $f(\boldsymbol{y} \mid \boldsymbol{x}) \approx p(\boldsymbol{y} \mid \boldsymbol{x})$.

We assume that an anchor sample $\boldsymbol{y}$ is given. For the anchor sample $\boldsymbol{y}$, we draw a positive sample $\boldsymbol{x}_1$ according to $p(\boldsymbol{x}_1 \mid \boldsymbol{y})$. We define the set $\tilde{X} = \{\boldsymbol{x}_2, \ldots, \boldsymbol{x}_N\}$ of negative samples, which are drawn iid according to $p(\boldsymbol{x})$. We define the set $X = \{\boldsymbol{x}_1, \ldots, \boldsymbol{x}_N\}$.

We have

$$p(\tilde{X}) = \prod_{i=2}^{N} p(\boldsymbol{x}_i) \,, \tag{A62}$$

$$p(X \mid \boldsymbol{y}) = p(\boldsymbol{x}_1 \mid \boldsymbol{y}) \prod_{i=2}^{N} p(\boldsymbol{x}_i) = p(\boldsymbol{x}_1 \mid \boldsymbol{y}) \, p(\tilde{X}) \,, \tag{A63}$$

$$p(X) = \prod_{i=1}^{N} p(\boldsymbol{x}_i) = p(\boldsymbol{x}_1) \, p(\tilde{X}) \,. \tag{A64}$$

We use the score function

$$f(\boldsymbol{x}, \boldsymbol{y}) = \exp(\tau^{-1} \operatorname{sim}(\boldsymbol{x}, \boldsymbol{y})) \,, \tag{A65}$$

where $\operatorname{sim}(\boldsymbol{x}, \boldsymbol{y})$ is typically the cosine similarity.

The InfoLOOB with score function $f(\boldsymbol{x}, \boldsymbol{y})$ is defined as

$$\mathrm{I}_{\mathrm{InfoLOOB}}(X_1 \; ; \; Y) = \mathrm{E}_{p(\boldsymbol{y})} \left[ \mathrm{E}_{p(X \mid \boldsymbol{y})} \left[ \ln \left( \frac{f(\boldsymbol{x}_1, \boldsymbol{y})}{\frac{1}{N-1} \sum_{i=2}^{N} f(\boldsymbol{x}_i, \boldsymbol{y})} \right) \right] \right] \,. \tag{A66}$$

We define the variational distribution

$$q(\boldsymbol{x} \mid \boldsymbol{y}) = \frac{p(\boldsymbol{x}) \, f(\boldsymbol{x}, \boldsymbol{y})}{Z(\boldsymbol{y})} \,, \tag{A67}$$

$$Z(\boldsymbol{y}) = \mathrm{E}_{p(\boldsymbol{x})} \left[ f(\boldsymbol{x}, \boldsymbol{y}) \right] \,. \tag{A68}$$

The next inequality shows the relation between $I(X_1 ; Y)$ and $I_{\text{InfoLOOB}}(X_1 ; Y)$ for random variables $X_1$ and $Y$.

$$I(X_1 ; Y) \; = \; E_{p(\boldsymbol{x}_1, \boldsymbol{y})} \left[ \ln \frac{p(\boldsymbol{x}_1 \mid \boldsymbol{y})}{p(\boldsymbol{x}_1)} \right] \tag{A69}$$

$$= \; E_{p(\boldsymbol{x}_1, \boldsymbol{y})} \left[ \ln \left( \frac{p(\boldsymbol{x}_1 \mid \boldsymbol{y})}{q(\boldsymbol{x}_1 \mid \boldsymbol{y})} \, \frac{q(\boldsymbol{x}_1 \mid \boldsymbol{y})}{p(\boldsymbol{x}_1)} \right) \right]$$

$$= \; E_{p(\boldsymbol{x}_1, \boldsymbol{y})} \left[ \ln \frac{q(\boldsymbol{x}_1 \mid \boldsymbol{y})}{p(\boldsymbol{x}_1)} \right] \; + \; E_{p(\boldsymbol{y})} \left[ \text{KL}(p(\boldsymbol{x}_1 \mid \boldsymbol{y}) \, \| \, q(\boldsymbol{x}_1 \mid \boldsymbol{y})) \right]$$

$$= \; E_{p(\boldsymbol{x}_1, \boldsymbol{y})} \left[ \ln \frac{f(\boldsymbol{x}_1, \boldsymbol{y})}{Z(\boldsymbol{y})} \right] \; + \; E_{p(\boldsymbol{y})} \left[ \text{KL}(p(\boldsymbol{x}_1 \mid \boldsymbol{y}) \, \| \, q(\boldsymbol{x}_1 \mid \boldsymbol{y})) \right]$$

$$= \; E_{p(\boldsymbol{x}_1, \boldsymbol{y})} \left[ \ln \left( \frac{f(\boldsymbol{x}_1, \boldsymbol{y})}{E_{p(X|\boldsymbol{y})} \left[ \frac{1}{N-1} \sum_{i=2}^{N} f(\boldsymbol{x}_i, \boldsymbol{y}) \right]} \right) \right] \; + \; E_{p(\boldsymbol{y})} \left[ \text{KL}(p(\boldsymbol{x}_1 \mid \boldsymbol{y}) \, \| \, q(\boldsymbol{x}_1 \mid \boldsymbol{y})) \right]$$

$$= \; E_{p(\boldsymbol{x}_1, \boldsymbol{y})} \left[ \ln f(\boldsymbol{x}_1, \boldsymbol{y}) \right] \; - \; E_{p(\boldsymbol{y})} \left[ \ln \left( E_{p(X|\boldsymbol{y})} \left[ \frac{1}{N-1} \sum_{i=2}^{N} f(\boldsymbol{x}_i, \boldsymbol{y}) \right] \right) \right]$$
$$+ \; E_{p(\boldsymbol{y})} \left[ \text{KL}(p(\boldsymbol{x}_1 \mid \boldsymbol{y}) \, \| \, q(\boldsymbol{x}_1 \mid \boldsymbol{y})) \right]$$

$$= \; E_{p(\boldsymbol{x}_1, \boldsymbol{y})} \left[ \ln f(\boldsymbol{x}_1, \boldsymbol{y}) \right] \; - \; E_{p(\boldsymbol{y})} \left[ E_{p(X|\boldsymbol{y})} \left[ \ln \left( \frac{1}{N-1} \sum_{i=2}^{N} f(\boldsymbol{x}_i, \boldsymbol{y}) \right) \right] \right]$$
$$+ \; E_{p(\boldsymbol{y})} \left[ E_{p(X|\boldsymbol{y})} \left[ \ln \left( \frac{1}{N-1} \sum_{i=2}^{N} f(\boldsymbol{x}_i, \boldsymbol{y}) \right) \right] \right] \; - \; E_{p(\boldsymbol{y})} \left[ \ln \left( E_{p(X|\boldsymbol{y})} \left[ \frac{1}{N-1} \sum_{i=2}^{N} f(\boldsymbol{x}_i, \boldsymbol{y}) \right] \right) \right]$$
$$+ \; E_{p(\boldsymbol{y})} \left[ \text{KL}(p(\boldsymbol{x}_1 \mid \boldsymbol{y}) \, \| \, q(\boldsymbol{x}_1 \mid \boldsymbol{y})) \right]$$

$$= \; E_{p(\boldsymbol{y})} \left[ E_{p(X|\boldsymbol{y})} \left[ \ln \left( \frac{f(\boldsymbol{x}_1, \boldsymbol{y})}{\frac{1}{N-1} \sum_{i=2}^{N} f(\boldsymbol{x}_i, \boldsymbol{y})} \right) \right] \right] \; + \; E_{p(\boldsymbol{y})} \left[ E_{p(X|\boldsymbol{y})} \left[ \ln \left( \frac{1}{N-1} \sum_{i=2}^{N} f(\boldsymbol{x}_i, \boldsymbol{y}) \right) \right] \right]$$
$$- \; E_{p(\boldsymbol{y})} \left[ \ln \left( E_{p(X|\boldsymbol{y})} \left[ \frac{1}{N-1} \sum_{i=2}^{N} f(\boldsymbol{x}_i, \boldsymbol{y}) \right] \right) \right] \; + \; E_{p(\boldsymbol{y})} \left[ \text{KL}(p(\boldsymbol{x}_1 \mid \boldsymbol{y}) \, \| \, q(\boldsymbol{x}_1 \mid \boldsymbol{y})) \right]$$

$$= \; I_{\text{InfoLOOB}}(X_1 ; Y)$$
$$+ \; E_{p(\boldsymbol{y})} \left[ E_{p(X|\boldsymbol{y})} \left[ \ln \left( \frac{1}{N-1} \sum_{i=2}^{N} f(\boldsymbol{x}_i, \boldsymbol{y}) \right) \right] \right] \; - \; E_{p(\boldsymbol{y})} \left[ \ln \left( E_{p(X|\boldsymbol{y})} \left[ \frac{1}{N-1} \sum_{i=2}^{N} f(\boldsymbol{x}_i, \boldsymbol{y}) \right] \right) \right]$$
$$+ \; E_{p(\boldsymbol{y})} \left[ \text{KL}(p(\boldsymbol{x}_1 \mid \boldsymbol{y}) \, \| \, q(\boldsymbol{x}_1 \mid \boldsymbol{y})) \right]$$

$$= \; I_{\text{InfoLOOB}}(X_1 ; Y)$$
$$+ \; E_{p(\boldsymbol{y})} \left[ E_{p(X|\boldsymbol{y})} \left[ \ln \left( \frac{1}{N-1} \sum_{i=2}^{N} f(\boldsymbol{x}_i, \boldsymbol{y}) \right) \right] \right] \; - \; E_{p(\boldsymbol{y})} \left[ \ln \left( E_{p(\boldsymbol{x}_1)} \left[ f(\boldsymbol{x}_1, \boldsymbol{y}) \right] \right) \right]$$
$$+ \; E_{p(\boldsymbol{y})} \left[ \text{KL}(p(\boldsymbol{x}_1 \mid \boldsymbol{y}) \, \| \, q(\boldsymbol{x}_1 \mid \boldsymbol{y})) \right]$$

$$= \; I_{\text{InfoLOOB}}(X_1 ; Y)$$
$$- \; E_{p(\boldsymbol{y})} \left[ E_{p(\tilde{X})} \left[ \ln \left( \frac{E_{p(\boldsymbol{x}_1)} \left[ f(\boldsymbol{x}_1, \boldsymbol{y}) \right]}{\frac{1}{N-1} \sum_{i=2}^{N} f(\boldsymbol{x}_i, \boldsymbol{y})} \right) \right] \right]$$
$$+ \; E_{p(\boldsymbol{y})} \left[ \text{KL}(p(\boldsymbol{x}_1 \mid \boldsymbol{y}) \, \| \, q(\boldsymbol{x}_1 \mid \boldsymbol{y})) \right]$$

$$= \; I_{\text{InfoLOOB}}(X_1 ; Y) \; - \; \text{DE} \; + \; E_{p(\boldsymbol{y})} \left[ \text{KL}(p(\boldsymbol{x}_1 \mid \boldsymbol{y}) \, \| \, q(\boldsymbol{x}_1 \mid \boldsymbol{y})) \right] \; ,$$

where we used

$$\text{DE} \; = \; E_{p(\boldsymbol{y})} \left[ E_{p(\tilde{X})} \left[ \ln \left( \frac{E_{p(\boldsymbol{x}_1)} \left[ f(\boldsymbol{x}_1, \boldsymbol{y}) \right]}{\frac{1}{N-1} \sum_{i=2}^{N} f(\boldsymbol{x}_i, \boldsymbol{y})} \right) \right] \right] \tag{A70}$$

and

$$Z(\boldsymbol{y}) \; = \; \mathrm{E}_{p(\boldsymbol{x}_1)}\left[f(\boldsymbol{x}_1, \boldsymbol{y})\right] \; = \; \mathrm{E}_{p(\tilde{X})}\left[\frac{1}{N-1}\sum_{i=2}^{N}f(\boldsymbol{x}_i, \boldsymbol{y})\right] \tag{A71}$$

$$= \; \mathrm{E}_{p(X|\boldsymbol{y})}\left[\frac{1}{N-1}\sum_{i=2}^{N}f(\boldsymbol{x}_i, \boldsymbol{y})\right] \; .$$

Since both KL and DE are non-negative (for DE see below), to increase InfoLOOB we have either to decrease KL or to increase DE.

**Bounding** DE. Next we bound DE. We define

$$\mathrm{L} \; = \; \boldsymbol{z}^T \boldsymbol{x} \; - \; \beta^{-1}\sum_{i=1}^{N}z_i \ln z_i \; . \tag{A72}$$

The log-sum-exponential (lse) is the maximum of L on the $N$-dimensional simplex $D$ with $D = \{\boldsymbol{z}\,|\,\sum_i z_i = 1, 0 \leqslant z_i\}$ (Gao & Pavel, 2017):

$$\mathrm{lse}(\beta, \boldsymbol{x}) \; = \; \max_{\boldsymbol{z}\in D}\boldsymbol{z}^T\boldsymbol{x} \; - \; \beta^{-1}\sum_{i=1}^{N}z_i \ln z_i \; . \tag{A73}$$

For some $\boldsymbol{z} \in D$ we have

$$\mathrm{E}_{\boldsymbol{a}}\left[\mathrm{lse}(\beta, \boldsymbol{a})\right] \; \geqslant \; \mathrm{E}_{\boldsymbol{a}}\left[\boldsymbol{z}^T\boldsymbol{a} \; - \; \beta^{-1}\sum_{i=1}^{N}z_i \ln z_i\right] \; = \; \boldsymbol{z}^T\mathrm{E}_{\boldsymbol{a}}\left[\boldsymbol{a}\right] \; - \; \beta^{-1}\sum_{i=1}^{N}z_i \ln z_i \; , \tag{A74}$$

therefore

$$\mathrm{E}_{\boldsymbol{a}}\left[\mathrm{lse}(\beta, \boldsymbol{a})\right] \; \geqslant \; \max_{\boldsymbol{z}\in D}\boldsymbol{z}^T\mathrm{E}_{\boldsymbol{a}}\left[\boldsymbol{a}\right] \; - \; \beta^{-1}\sum_{i=1}^{N}z_i \ln z_i \; = \; \mathrm{lse}(\beta, \mathrm{E}_{\boldsymbol{a}}\left[\boldsymbol{a}\right]) \; . \tag{A75}$$

We obtain

$$\mathrm{E}_{p(\boldsymbol{y})}\left[\mathrm{E}_{p(\tilde{X})}\left[\ln\left(\mathrm{E}_{p(\boldsymbol{x}_1)}\left[\frac{\exp(\tau^{-1}\mathrm{sim}(\boldsymbol{x}_1, \boldsymbol{y}))}{\frac{1}{N-1}\sum_{i=2}^{N}\exp(\tau^{-1}\mathrm{sim}(\boldsymbol{x}_i, \boldsymbol{y}))}\right]\right)\right]\right] \tag{A76}$$

$$\leqslant \; \mathrm{E}_{p(\boldsymbol{y})}\left[\ln\mathrm{E}_{p(\boldsymbol{x}_1)}\left[\exp(\tau^{-1}\mathrm{sim}(\boldsymbol{x}_1, \boldsymbol{y}))\right] \; - \; \ln\left(\frac{1}{N-1}\sum_{i=2}^{N}\exp(\tau^{-1}\mathrm{E}_{p(\boldsymbol{x}_i)}\left[\mathrm{sim}(\boldsymbol{x}_i, \boldsymbol{y})\right])\right)\right]$$

$$= \; \mathrm{E}_{p(\boldsymbol{y})}\left[\ln\mathrm{E}_{p(\boldsymbol{x}_1)}\left[\exp(\tau^{-1}\mathrm{sim}(\boldsymbol{x}_1, \boldsymbol{y}))\right] \; - \; \tau^{-1}\mathrm{E}_{p(\boldsymbol{x}_1)}\left[\mathrm{sim}(\boldsymbol{x}_1, \boldsymbol{y})\right]\right] \; .$$

We obtain via Jensen's inequality

$$\mathrm{E}_{p(\boldsymbol{y})}\left[\mathrm{E}_{p(\tilde{X})}\left[\ln\left(\mathrm{E}_{p(\boldsymbol{x}_1)}\left[\frac{\exp(\tau^{-1}\mathrm{sim}(\boldsymbol{x}_1, \boldsymbol{y}))}{\frac{1}{N-1}\sum_{i=2}^{N}\exp(\tau^{-1}\mathrm{sim}(\boldsymbol{x}_i, \boldsymbol{y}))}\right]\right)\right]\right] \tag{A77}$$

$$\geqslant \; \mathrm{E}_{p(\boldsymbol{y})}\left[\ln\mathrm{E}_{p(\boldsymbol{x}_1)}\left[\exp(\tau^{-1}\mathrm{sim}(\boldsymbol{x}_1, \boldsymbol{y}))\right] \; - \; \ln\left(\frac{1}{N-1}\sum_{i=2}^{N}\mathrm{E}_{p(\boldsymbol{x}_i)}\left[\exp(\tau^{-1}\mathrm{sim}(\boldsymbol{x}_1, \boldsymbol{y}))\right]\right)\right]$$

$$= \; 0 \; .$$

If we combine both previous inequalities, we obtain

$$0 \; \leqslant \; \mathrm{DE} \; \leqslant \; \mathrm{E}_{p(\boldsymbol{y})}\left[\ln\mathrm{E}_{p(\boldsymbol{x}_1)}\left[\exp(\tau^{-1}\mathrm{sim}(\boldsymbol{x}_1, \boldsymbol{y}))\right] \; - \; \tau^{-1}\mathrm{E}_{p(\boldsymbol{x}_1)}\left[\mathrm{sim}(\boldsymbol{x}_1, \boldsymbol{y})\right]\right] \; . \tag{A78}$$

In particular, for bounded $\text{sim}(\boldsymbol{x}_1, \boldsymbol{y})$, we get

$$0 \leqslant \text{DE} \leqslant \tau^{-1} \left( \max_{\boldsymbol{y}, \boldsymbol{x}_1} \text{sim}(\boldsymbol{x}_1, \boldsymbol{y}) - \min_{\boldsymbol{y}, \boldsymbol{x}_1} \text{sim}(\boldsymbol{x}_1, \boldsymbol{y}) \right) , \tag{A79}$$

while Hoeffding's lemma gives

$$0 \leqslant \text{DE} \leqslant \frac{1}{8} \tau^{-2} \left( \max_{\boldsymbol{y}, \boldsymbol{x}_1} \text{sim}(\boldsymbol{x}_1, \boldsymbol{y}) - \min_{\boldsymbol{y}, \boldsymbol{x}_1} \text{sim}(\boldsymbol{x}_1, \boldsymbol{y}) \right)^2 . \tag{A80}$$

Thus, for bounded $\text{sim}(\boldsymbol{x}_1, \boldsymbol{y})$, DE is bounded, therefore also InfoLOOB. For sub-exponential distributions with variance $\sigma^2$, for which Bernstein's condition with $\tau > b$ holds (Eq. (2.16) in Wainwright (2019)), we get (Proposition 2.3 in Wainwright (2019)):

$$0 \leqslant \text{DE} \leqslant \frac{\sigma^2}{2 \left( \tau^2 - b \, \tau \right)} . \tag{A81}$$

Next, we show that DE is small. Hoeffding's inequality states that if $f(\boldsymbol{x}, \boldsymbol{y}) \in [a, b]$ then

$$p \left( \left| \text{E}_{p(\boldsymbol{x}_1)} \left[ f(\boldsymbol{x}_1, \boldsymbol{y}) \right] - \frac{1}{N-1} \sum_{i=2}^{N} f(\boldsymbol{x}_i, \boldsymbol{y}) \right| \geq \epsilon \right) \leqslant 2 \exp \left( - \frac{2 \left( N - 1 \right) \epsilon^2}{(b-a)^2} \right) . \tag{A82}$$

For

$$\text{E}_{p(\boldsymbol{x}_1)} \left[ f(\boldsymbol{x}_1, \boldsymbol{y}) \right] - \frac{1}{N-1} \sum_{i=2}^{N} f(\boldsymbol{x}_i, \boldsymbol{y}) \leqslant \epsilon \tag{A83}$$

we have

$$\ln \left( \frac{\text{E}_{p(\boldsymbol{x}_1)} \left[ f(\boldsymbol{x}_1, \boldsymbol{y}) \right]}{\frac{1}{N-1} \sum_{i=2}^{N} f(\boldsymbol{x}_i, \boldsymbol{y})} \right) \leqslant \ln \left( \frac{\frac{1}{N-1} \sum_{i=2}^{N} f(\boldsymbol{x}_i, \boldsymbol{y}) + \epsilon}{\frac{1}{N-1} \sum_{i=2}^{N} f(\boldsymbol{x}_i, \boldsymbol{y})} \right) \tag{A84}$$

$$\leqslant \frac{\epsilon}{\frac{1}{N-1} \sum_{i=2}^{N} f(\boldsymbol{x}_i, \boldsymbol{y})} \leqslant \frac{\epsilon}{Z - \epsilon} ,$$

where we used $\ln a \leqslant a - 1$ for $0 < a$. Analog for

$$\frac{1}{N-1} \sum_{i=2}^{N} f(\boldsymbol{x}_i, \boldsymbol{y}) - \text{E}_{p(\boldsymbol{x}_1)} \left[ f(\boldsymbol{x}_1, \boldsymbol{y}) \right] \leqslant \epsilon \tag{A85}$$

we have

$$\ln \left( \frac{\text{E}_{p(\boldsymbol{x}_1)} \left[ f(\boldsymbol{x}_1, \boldsymbol{y}) \right]}{\frac{1}{N-1} \sum_{i=2}^{N} f(\boldsymbol{x}_i, \boldsymbol{y})} \right) \geq \ln \left( \frac{\text{E}_{p(\boldsymbol{x}_1)} \left[ f(\boldsymbol{x}_1, \boldsymbol{y}) \right]}{\text{E}_{p(\boldsymbol{x}_1)} \left[ f(\boldsymbol{x}_1, \boldsymbol{y}) \right] + \epsilon} \right) \tag{A86}$$

$$= - \ln \left( \frac{\text{E}_{p(\boldsymbol{x}_1)} \left[ f(\boldsymbol{x}_1, \boldsymbol{y}) \right] + \epsilon}{\text{E}_{p(\boldsymbol{x}_1)} \left[ f(\boldsymbol{x}_1, \boldsymbol{y}) \right]} \right) \geq - \frac{\epsilon}{\text{E}_{p(\boldsymbol{x}_1)} \left[ f(\boldsymbol{x}_1, \boldsymbol{y}) \right]} = - \frac{\epsilon}{Z} ,$$

where we used $- \ln a \geq 1 - a$ for $0 < a$.

In summary, for

$$\left| \text{E}_{p(\boldsymbol{x}_1)} \left[ f(\boldsymbol{x}_1, \boldsymbol{y}) \right] - \frac{1}{N-1} \sum_{i=2}^{N} f(\boldsymbol{x}_i, \boldsymbol{y}) \right| \leqslant \epsilon \tag{A87}$$

we have

$$- \frac{\epsilon}{Z} \leqslant \ln \left( \frac{\text{E}_{p(\boldsymbol{x}_1)} \left[ f(\boldsymbol{x}_1, \boldsymbol{y}) \right]}{\frac{1}{N-1} \sum_{i=2}^{N} f(\boldsymbol{x}_i, \boldsymbol{y})} \right) \leqslant \frac{\epsilon}{Z - \epsilon} . \tag{A88}$$

It follows that

$$- \frac{\epsilon}{Z} \; \leqslant \; \mathrm{DE} \; \leqslant \; \frac{\epsilon}{Z - \epsilon} \; . \tag{A89}$$

DE averages the ln-term over $\boldsymbol{y}$ and $\tilde{X}$, therefore it has an even smaller bound than the bound above on the ln-term. Consequently, for small $b - a$ and large $N$, the term DE is small.

KL is decreased by making the variation distribution $q(\boldsymbol{x}_1 \mid \boldsymbol{y})$ more similar to the posterior $p(\boldsymbol{x}_1 \mid \boldsymbol{y})$. The value DE only depends on the marginal distributions $p(\boldsymbol{y})$ and $p(\boldsymbol{x})$, since $p(\tilde{X}) = \prod_{i=2}^{N} p(\boldsymbol{x}_i)$. The value DE can be changed by adding an offset to $f(\boldsymbol{x}, \boldsymbol{y})$. However, scaling $f(\boldsymbol{x}, \boldsymbol{y})$ by a factor does not change DE. Consequently, DE is difficult to change.

Therefore, increasing InfoLOOB is most effective by making $q(\boldsymbol{x}_1 \mid \boldsymbol{y})$ more similar to the posterior $p(\boldsymbol{x}_1 \mid \boldsymbol{y})$.

**Gradient of InfoLOOB expressed by gradients of** KL **and** DE**.** Assume that the similarity is parametrized by $\boldsymbol{w}$ giving $\mathrm{sim}(\boldsymbol{x}, \boldsymbol{y}; \boldsymbol{w})$.

$$\begin{aligned}
\mathrm{KL}(p(\boldsymbol{x}_1 \mid \boldsymbol{y}) \parallel q(\boldsymbol{x}_1 \mid \boldsymbol{y})) &= \int p(\boldsymbol{x}_1 \mid \boldsymbol{y}) \, \ln \left( \frac{p(\boldsymbol{x}_1 \mid \boldsymbol{y})}{q(\boldsymbol{x}_1 \mid \boldsymbol{y})} \right) \, \mathrm{d}\boldsymbol{x} \\
&= - \tau^{-1} \int p(\boldsymbol{x}_1 \mid \boldsymbol{y}) \, \mathrm{sim}(\boldsymbol{x}_1, \boldsymbol{y}; \boldsymbol{w}) \, \mathrm{d}\boldsymbol{x}_1 \; + \; \ln Z \; + \; C \, ,
\end{aligned} \tag{A90}$$

where $C$ is independent of $\boldsymbol{w}$.

Next, we compute the derivative of KL with respect to parameters $\boldsymbol{w}$.

$$\begin{aligned}
\frac{\partial \mathrm{KL}}{\partial \boldsymbol{w}} & \tag{A91} \\
&= - \tau^{-1} \int p(\boldsymbol{x}_1 \mid \boldsymbol{y}) \frac{\partial \mathrm{sim}(\boldsymbol{x}_1, \boldsymbol{y}; \boldsymbol{w})}{\partial \boldsymbol{w}} \, \mathrm{d}\boldsymbol{x}_1 \; + \; \frac{1}{Z} \int p(\boldsymbol{x}_1) \frac{\exp(\tau^{-1} \mathrm{sim}(\boldsymbol{x}_1, \boldsymbol{y}; \boldsymbol{w}))}{\partial \mathrm{sim}(\boldsymbol{x}_1, \boldsymbol{y}; \boldsymbol{w})} \frac{\partial \mathrm{sim}(\boldsymbol{x}_1, \boldsymbol{y}; \boldsymbol{w})}{\partial \boldsymbol{w}} \, \mathrm{d}\boldsymbol{x}_1 \\
&= - \tau^{-1} \int p(\boldsymbol{x}_1 \mid \boldsymbol{y}) \frac{\partial \mathrm{sim}(\boldsymbol{x}_1, \boldsymbol{y}; \boldsymbol{w})}{\partial \boldsymbol{w}} \, \mathrm{d}\boldsymbol{x}_1 \; + \; \tau^{-1} \int p(\boldsymbol{x}_1) \frac{\exp(\tau^{-1} \mathrm{sim}(\boldsymbol{x}_1, \boldsymbol{y}; \boldsymbol{w}))}{Z} \frac{\partial \mathrm{sim}(\boldsymbol{x}_1, \boldsymbol{y}; \boldsymbol{w})}{\partial \boldsymbol{w}} \, \mathrm{d}\boldsymbol{x}_1 \\
&= - \tau^{-1} \int p(\boldsymbol{x}_1 \mid \boldsymbol{y}) \frac{\partial \mathrm{sim}(\boldsymbol{x}_1, \boldsymbol{y}; \boldsymbol{w})}{\partial \boldsymbol{w}} \, \mathrm{d}\boldsymbol{x}_1 \; + \; \tau^{-1} \int q(\boldsymbol{x}_1 \mid \boldsymbol{y}) \frac{\partial \mathrm{sim}(\boldsymbol{x}_1, \boldsymbol{y}; \boldsymbol{w})}{\partial \boldsymbol{w}} \, \mathrm{d}\boldsymbol{x}_1 \\
&= \tau^{-1} \int (q(\boldsymbol{x}_1 \mid \boldsymbol{y}) \; - \; p(\boldsymbol{x}_1 \mid \boldsymbol{y})) \frac{\partial \mathrm{sim}(\boldsymbol{x}_1, \boldsymbol{y}; \boldsymbol{w})}{\partial \boldsymbol{w}} \, \mathrm{d}\boldsymbol{x}_1 \, .
\end{aligned}$$

The derivative is the average difference between the posterior distribution $p(\boldsymbol{x}_1 \mid \boldsymbol{y})$ and the variational distribution $q(\boldsymbol{x}_1 \mid \boldsymbol{y})$ multiplied by the derivative of the similarity function. If both distribution match, then the derivative vanishes.

Next, we compute the derivative of DE with respect to parameters $\boldsymbol{w}$.

$$\frac{\partial \text{DE}}{\partial \boldsymbol{w}} \tag{A92}$$

$$= \text{E}_{p(\boldsymbol{y})}\left[\frac{\partial \ln Z}{\partial \boldsymbol{w}}\right] - \text{E}_{p(\boldsymbol{y})}\left[\text{E}_{p(\tilde{X})}\left[\frac{\frac{1}{N-1}\sum_{i=2}^{N}\tau^{-1}\exp(\tau^{-1}\text{sim}(\boldsymbol{x}_i,\boldsymbol{y};\boldsymbol{w}))\frac{\partial\text{sim}(\boldsymbol{x}_i,\boldsymbol{y};\boldsymbol{w})}{\partial\boldsymbol{w}}}{\frac{1}{N-1}\sum_{j=2}^{N}f(\boldsymbol{x}_j,\boldsymbol{y})}\right]\right]$$

$$= \text{E}_{p(\boldsymbol{y})}\left[\tau^{-1}\int q(\boldsymbol{x}_1\mid\boldsymbol{y})\frac{\partial\text{sim}(\boldsymbol{x}_1,\boldsymbol{y};\boldsymbol{w})}{\partial\boldsymbol{w}}\,\text{d}\boldsymbol{x}_1\right]$$

$$- \text{E}_{p(\boldsymbol{y})}\left[\text{E}_{p(\tilde{X})}\left[\frac{\frac{1}{N-1}\sum_{i=2}^{N}\tau^{-1}\exp(\tau^{-1}\text{sim}(\boldsymbol{x}_i,\boldsymbol{y};\boldsymbol{w}))\frac{\partial\text{sim}(\boldsymbol{x}_i,\boldsymbol{y};\boldsymbol{w})}{\partial\boldsymbol{w}}}{\frac{1}{N-1}\sum_{j=2}^{N}f(\boldsymbol{x}_j,\boldsymbol{y})}\right]\right]$$

$$= \tau^{-1}\,\text{E}_{p(\boldsymbol{y})}\left[\int q(\boldsymbol{x}_1\mid\boldsymbol{y})\frac{\partial\text{sim}(\boldsymbol{x}_1,\boldsymbol{y};\boldsymbol{w})}{\partial\boldsymbol{w}}\,\text{d}\boldsymbol{x}_1\right]$$

$$- \tau^{-1}\,\text{E}_{p(\boldsymbol{y})}\left[\text{E}_{p(\tilde{X})}\left[\frac{1}{N-1}\sum_{i=2}^{N}\frac{f(\boldsymbol{x}_i,\boldsymbol{y})}{\frac{1}{N-1}\sum_{j=2}^{N}f(\boldsymbol{x}_j,\boldsymbol{y})}\frac{\partial\text{sim}(\boldsymbol{x}_i,\boldsymbol{y};\boldsymbol{w})}{\partial\boldsymbol{w}}\right]\right]$$

$$= \tau^{-1}\,\text{E}_{p(\boldsymbol{y})}\left[\int\frac{p(\boldsymbol{x}_1)\,f(\boldsymbol{x}_1,\boldsymbol{y})}{\text{E}_{p(\boldsymbol{x})}\left[f(\boldsymbol{x},\boldsymbol{y})\right]}\frac{\partial\text{sim}(\boldsymbol{x}_1,\boldsymbol{y};\boldsymbol{w})}{\partial\boldsymbol{w}}\,\text{d}\boldsymbol{x}_1\right]$$

$$- \tau^{-1}\,\text{E}_{p(\boldsymbol{y})}\left[\text{E}_{p(\tilde{X})}\left[\frac{1}{N-1}\sum_{i=2}^{N}\frac{f(\boldsymbol{x}_i,\boldsymbol{y})}{\frac{1}{N-1}\sum_{j=2}^{N}f(\boldsymbol{x}_j,\boldsymbol{y})}\frac{\partial\text{sim}(\boldsymbol{x}_i,\boldsymbol{y};\boldsymbol{w})}{\partial\boldsymbol{w}}\right]\right]$$

$$= \tau^{-1}\,\text{E}_{p(\boldsymbol{y})}\left[\text{E}_{p(\boldsymbol{x}_1)}\left[\frac{f(\boldsymbol{x}_1,\boldsymbol{y})}{\text{E}_{p(\boldsymbol{x})}\left[f(\boldsymbol{x},\boldsymbol{y})\right]}\frac{\partial\text{sim}(\boldsymbol{x}_1,\boldsymbol{y};\boldsymbol{w})}{\partial\boldsymbol{w}}\right]\right]$$

$$- \tau^{-1}\,\text{E}_{p(\boldsymbol{y})}\left[\text{E}_{p(\tilde{X})}\left[\frac{1}{N-1}\sum_{i=2}^{N}\frac{f(\boldsymbol{x}_i,\boldsymbol{y})}{\frac{1}{N-1}\sum_{j=2}^{N}f(\boldsymbol{x}_j,\boldsymbol{y})}\frac{\partial\text{sim}(\boldsymbol{x}_i,\boldsymbol{y};\boldsymbol{w})}{\partial\boldsymbol{w}}\right]\right]$$

$$= \tau^{-1}\,\text{E}_{p(\boldsymbol{y})}\left[\frac{1}{N-1}\sum_{i=2}^{N}\text{E}_{p(\boldsymbol{x}_i)}\left[\frac{f(\boldsymbol{x}_i,\boldsymbol{y})}{\text{E}_{p(\boldsymbol{x})}\left[f(\boldsymbol{x},\boldsymbol{y})\right]}\frac{\partial\text{sim}(\boldsymbol{x}_i,\boldsymbol{y};\boldsymbol{w})}{\partial\boldsymbol{w}}\right]\right]$$

$$- \tau^{-1}\,\text{E}_{p(\boldsymbol{y})}\left[\text{E}_{p(\tilde{X})}\left[\frac{1}{N-1}\sum_{i=2}^{N}\frac{f(\boldsymbol{x}_i,\boldsymbol{y})}{\frac{1}{N-1}\sum_{j=2}^{N}f(\boldsymbol{x}_j,\boldsymbol{y})}\frac{\partial\text{sim}(\boldsymbol{x}_i,\boldsymbol{y};\boldsymbol{w})}{\partial\boldsymbol{w}}\right]\right]$$

$$= \tau^{-1}\,\text{E}_{p(\boldsymbol{y})}\left[\text{E}_{p(\tilde{X})}\left[\frac{1}{N-1}\sum_{i=2}^{N}\frac{f(\boldsymbol{x}_i,\boldsymbol{y})}{\text{E}_{p(\boldsymbol{x})}\left[f(\boldsymbol{x},\boldsymbol{y})\right]}\frac{\partial\text{sim}(\boldsymbol{x}_i,\boldsymbol{y};\boldsymbol{w})}{\partial\boldsymbol{w}}\right]\right]$$

$$- \tau^{-1}\,\text{E}_{p(\boldsymbol{y})}\left[\text{E}_{p(\tilde{X})}\left[\frac{1}{N-1}\sum_{i=2}^{N}\frac{f(\boldsymbol{x}_i,\boldsymbol{y})}{\frac{1}{N-1}\sum_{j=2}^{N}f(\boldsymbol{x}_j,\boldsymbol{y})}\frac{\partial\text{sim}(\boldsymbol{x}_i,\boldsymbol{y};\boldsymbol{w})}{\partial\boldsymbol{w}}\right]\right]$$

$$= \tau^{-1}\,\text{E}_{p(\boldsymbol{y})}\left[\text{E}_{p(\tilde{X})}\left[\frac{1}{N-1}\sum_{i=2}^{N}\left(\frac{1}{\text{E}_{p(\boldsymbol{x})}\left[f(\boldsymbol{x},\boldsymbol{y})\right]}-\frac{1}{\frac{1}{N-1}\sum_{j=2}^{N}f(\boldsymbol{x}_j,\boldsymbol{y})}\right)f(\boldsymbol{x}_i,\boldsymbol{y})\frac{\partial\text{sim}(\boldsymbol{x}_i,\boldsymbol{y};\boldsymbol{w})}{\partial\boldsymbol{w}}\right]\right]$$

$$= \tau^{-1}\,\text{E}_{p(\boldsymbol{y})}\left[\text{E}_{p(\tilde{X})}\left[\frac{1}{N-1}\sum_{i=2}^{N}\left(\frac{1}{Z}-\frac{1}{\frac{1}{N-1}\sum_{j=2}^{N}f(\boldsymbol{x}_j,\boldsymbol{y})}\right)f(\boldsymbol{x}_i,\boldsymbol{y})\frac{\partial\text{sim}(\boldsymbol{x}_i,\boldsymbol{y};\boldsymbol{w})}{\partial\boldsymbol{w}}\right]\right].$$

The derivative is the average of $\frac{1}{Z}-\frac{1}{\frac{1}{N-1}\sum_{j=2}^{N}f(\boldsymbol{x}_j,\boldsymbol{y})}$ multiplied by the score function and the derivative of the similarity function. The average is over $\boldsymbol{y}$ and $\tilde{X}$, therefore the whole derivative becomes even smaller. Consequently, for small $b-a$ and large $N$, the derivative of DE is small.

Note that for

$$\left|\text{E}_{p(\boldsymbol{x}_1)}\left[f(\boldsymbol{x}_1,\boldsymbol{y})\right]-\frac{1}{N-1}\sum_{i=2}^{N}f(\boldsymbol{x}_i,\boldsymbol{y})\right| \leqslant \epsilon \tag{A93}$$

we have

$$\frac{1}{Z} - \frac{1}{\frac{1}{N-1} \sum_{j=2}^{N} f(\boldsymbol{x}_j, \boldsymbol{y})} \leqslant \frac{1}{Z} - \frac{1}{Z+\epsilon} = \frac{\epsilon}{Z(Z+\epsilon)} \, , \tag{A94}$$

$$\frac{1}{Z} - \frac{1}{\frac{1}{N-1} \sum_{j=2}^{N} f(\boldsymbol{x}_j, \boldsymbol{y})} \geq \frac{1}{Z} - \frac{1}{Z-\epsilon} = -\frac{\epsilon}{Z(Z-\epsilon)} \, , \tag{A95}$$

therefore

$$\left| \frac{1}{Z} - \frac{1}{\frac{1}{N-1} \sum_{j=2}^{N} f(\boldsymbol{x}_j, \boldsymbol{y})} \right| \leqslant \frac{\epsilon}{Z(Z-\epsilon)} \, . \tag{A96}$$

If the expectation $Z$ is well approximated by the average $\frac{1}{N-1} \sum_{j=2}^{N} f(\boldsymbol{x}_j, \boldsymbol{y})$, then both DE and its gradient are small.

Derivative of InfoLOOB via KL and DE:

$$\frac{\partial \mathrm{I}_{\mathrm{InfoLOOB}}(X_1 \; ; \; Y)}{\partial \boldsymbol{w}} = \frac{\partial \mathrm{DE}}{\partial \boldsymbol{w}} - \frac{\partial \mathrm{KL}}{\partial \boldsymbol{w}} \, . \tag{A97}$$

In this gradient, the KL term is dominating, therefore $f(\boldsymbol{x}, \boldsymbol{y})$ is pushed to approximate the conditional probability $p(\boldsymbol{y} \mid \boldsymbol{x})$. Modern Hopfield networks lead to larger values of $p(\boldsymbol{y} \mid \boldsymbol{x})$ as the mutual information becomes larger, therefore modern Hopfield networks help to push $f(\boldsymbol{x}, \boldsymbol{y})$ to large values. Furthermore, modern Hopfield networks increase $Z$, which is in the denominator of the bound on DE and its derivative.

### A.1.4 INFONCE AND INFOLOOB: GRADIENTS

We consider the InfoNCE and the InfoLOOB loss function. For computing the loss function, we sample $N$ pairs independently from $p(\boldsymbol{x}, \boldsymbol{y})$, which gives the training set $\{(\boldsymbol{x}_1, \boldsymbol{y}_1), (\boldsymbol{x}_2, \boldsymbol{y}_2), \ldots, (\boldsymbol{x}_N, \boldsymbol{y}_N)\}$. InfoNCE and InfoLOOB only differ in using the positive example in the negatives. More precisely, InfoNCE uses for the matrix of negative samples $\boldsymbol{X} = (\boldsymbol{x}_1, \ldots, \boldsymbol{x}_N)$, while InfoLOOB uses $\tilde{\boldsymbol{X}} = (\boldsymbol{x}_2, \ldots, \boldsymbol{x}_N)$.

**InfoNCE.**

The InfoNCE loss is

$$\mathrm{L}_{\mathrm{InfoNCE}} = -\frac{1}{N} \sum_{i=1}^{N} \ln \left( \frac{f(\boldsymbol{x}_i, \boldsymbol{y}_i)}{\frac{1}{N} \sum_{j=1}^{N} f(\boldsymbol{x}_j, \boldsymbol{y}_i)} \right) = \frac{1}{N} \sum_{i=1}^{N} \mathrm{L}_{\mathrm{InfoNCE}}(\boldsymbol{y}_i) \, , \tag{A98}$$

where we used

$$\mathrm{L}_{\mathrm{InfoNCE}}(\boldsymbol{y}_i) = -\ln \left( \frac{f(\boldsymbol{x}_i, \boldsymbol{y}_i)}{\frac{1}{N} \sum_{j=1}^{N} f(\boldsymbol{x}_j, \boldsymbol{y}_i)} \right) \, . \tag{A99}$$

For the score function $f(\boldsymbol{x}, \boldsymbol{y})$, we use

$$f(\boldsymbol{x}, \boldsymbol{y}) = \exp(\tau^{-1} \mathrm{sim}(\boldsymbol{x}, \boldsymbol{y})) \, , \tag{A100}$$

$$\mathrm{sim}(\boldsymbol{x}, \boldsymbol{y}) = \boldsymbol{y}^T \boldsymbol{x} \tag{A101}$$

with $\tau$ as the temperature.

The loss function for this score function is

$$\mathrm{L}_{\mathrm{InfoNCE}}(\boldsymbol{y}) = -\tau^{-1} \boldsymbol{y}^T \boldsymbol{x}_1 + \tau^{-1} \mathrm{lse}\left(\tau^{-1}, \boldsymbol{X}^T \boldsymbol{y}\right) \, , \tag{A102}$$

where lse is the *log-sum-exp function* (lse):

$$\mathrm{lse}(\beta, \boldsymbol{a}) = \beta^{-1} \log \left( \sum_{i=1}^{N} \exp(\beta a_i) \right) \, , \tag{A103}$$

for $\beta > 0$ and vector $\boldsymbol{a} = (a_1, \ldots, a_N)$.

The gradient with respect to $\boldsymbol{y}$ is

$$\frac{\partial \mathrm{L}_{\mathrm{InfoNCE}}(\boldsymbol{y})}{\partial \boldsymbol{y}} = - \tau^{-1} \, \boldsymbol{x}_1 \, + \, \tau^{-1} \, \boldsymbol{X} \, \mathrm{softmax} \left( \tau^{-1} \boldsymbol{X}^T \boldsymbol{y} \right) , \quad \text{(A104)}$$

which is the positive example $\boldsymbol{x}_1$ that fits to the anchor example $\boldsymbol{y}$ minus the Hopfield network update with state pattern $\boldsymbol{y}$ and stored patterns $\boldsymbol{X}$ and then this difference multiplied by $\tau^{-1}$.

This gradient can be simplified, since the positive example $\boldsymbol{x}_1$ is also in the negative examples. Using $\boldsymbol{p} = (p_1, \ldots, p_N)^T = \mathrm{softmax} \left( \tau^{-1} \boldsymbol{X}^T \boldsymbol{y} \right)$, we obtain

$$\frac{\partial \mathrm{L}_{\mathrm{InfoNCE}}(\boldsymbol{y})}{\partial \boldsymbol{y}} \quad \text{(A105)}$$

$$= - \tau^{-1} \left( 1 - p_1 \right) \left( \boldsymbol{x}_1 - \frac{1}{1 - p_1} \boldsymbol{X} \left( \mathrm{softmax} \left( \tau^{-1} \boldsymbol{X}^T \boldsymbol{y} \right) - (p_1, 0, \ldots, 0)^T \right) \right)$$

$$= - \tau^{-1} \left( 1 - p_1 \right) \left( \boldsymbol{x}_1 - \tilde{\boldsymbol{X}} \, \mathrm{softmax} \left( \tau^{-1} \tilde{\boldsymbol{X}}^T \boldsymbol{y} \right) \right) = \left( 1 - p_1 \right) \frac{\partial \mathrm{L}_{\mathrm{InfoLOOB}}(\boldsymbol{y})}{\partial \boldsymbol{y}} .$$

where

$$\frac{1}{1 - p_1} \boldsymbol{X} \left( \mathrm{softmax} \left( \tau^{-1} \boldsymbol{X}^T \boldsymbol{y} \right) - (p_1, 0, \ldots, 0)^T \right) \quad \text{(A106)}$$

$$= \frac{1}{1 - p_1} \boldsymbol{X} \left( (p_1, p_2, \ldots, p_N)^T - (p_1, 0, \ldots, 0)^T \right)$$

$$= \frac{1}{1 - p_1} \boldsymbol{X} (0, p_2, \ldots, p_N)^T = \frac{1}{1 - p_1} \sum_{i=2}^{N} p_i \, \boldsymbol{x}_i$$

is the softmax average over the negatives $\boldsymbol{x}_i$ for $2 \leqslant i \leqslant N$ without $\boldsymbol{x}_1$. It can be easily seen that $\frac{1}{1-p_1} \sum_{i=2}^{N} p_i = \frac{1-p_1}{1-p_1} = 1$. For the derivative of the InfoLOOB see below.

The gradient with respect to $\boldsymbol{x}_1$ is

$$\frac{\partial \mathrm{L}_{\mathrm{InfoNCE}}(\boldsymbol{y})}{\partial \boldsymbol{x}_1} = - \tau^{-1} \, \boldsymbol{y} \, + \, \tau^{-1} \, \frac{\exp(\tau^{-1} \, \boldsymbol{x}_1^T \boldsymbol{y})}{\sum_{i=1}^{N} \exp(\tau^{-1} \boldsymbol{x}_i^T \boldsymbol{y})} \, \boldsymbol{y} \quad \text{(A107)}$$

$$= - \tau^{-1} \left( 1 - p_1 \right) \boldsymbol{y} . \quad \text{(A108)}$$

Consequently, the learning rate is scaled by $(1 - p_1)$.

The sum of gradients with respect to $\boldsymbol{x}_1$ and $\boldsymbol{x}_i$ is

$$\frac{\partial \mathrm{L}_{\mathrm{InfoNCE}}(\boldsymbol{y})}{\partial \boldsymbol{x}_1} + \sum_{i=1}^{N} \frac{\partial \mathrm{L}_{\mathrm{InfoNCE}}(\boldsymbol{y})}{\partial \boldsymbol{x}_i} = - \tau^{-1} \, \boldsymbol{y} \, + \, \tau^{-1} \, \boldsymbol{y} \, \mathbf{1}^T \mathrm{softmax} \left( \tau^{-1} \boldsymbol{X}^T \boldsymbol{y} \right) \quad \text{(A109)}$$

$$= - \tau^{-1} \, \boldsymbol{y} + \tau^{-1} \, \boldsymbol{y} = 0 ,$$

where $\mathbf{1}$ is the vector with ones. However, the derivatives with respect to the weights are not zero since the $\boldsymbol{x}_i$ are differently computed.

**InfoLOOB.**

The InfoLOOB loss is

$$\mathrm{L}_{\mathrm{InfoLOOB}} = - \frac{1}{N} \sum_{i=1}^{N} \ln \left( \frac{f(\boldsymbol{x}_i, \boldsymbol{y}_i)}{\frac{1}{N-1} \sum_{j=1, j \neq i}^{N} f(\boldsymbol{x}_j, \boldsymbol{y}_i)} \right) = \frac{1}{N} \sum_{i=1}^{N} \mathrm{L}_{\mathrm{InfoLOOB}}(\boldsymbol{y}_i) , \quad \text{(A110)}$$

where we used

$$\mathrm{L}_{\mathrm{InfoLOOB}}(\boldsymbol{y}_i) = - \ln \left( \frac{f(\boldsymbol{x}_i, \boldsymbol{y}_i)}{\frac{1}{N-1} \sum_{j=1, j \neq i}^{N} f(\boldsymbol{x}_j, \boldsymbol{y}_i)} \right) . \quad \text{(A111)}$$

For the score function $f(\boldsymbol{x}, \boldsymbol{y})$, we use

$$f(\boldsymbol{x}, \boldsymbol{y}) \;=\; \exp(\tau^{-1} \operatorname{sim}(\boldsymbol{x}, \boldsymbol{y}))\,, \tag{A112}$$

$$\operatorname{sim}(\boldsymbol{x}, \boldsymbol{y}) \;=\; \boldsymbol{y}^T \boldsymbol{x} \tag{A113}$$

with $\tau$ as the temperature.

The loss function for this score function is

$$\mathrm{L}_{\mathrm{InfoLOOB}}(\boldsymbol{y}) \;=\; -\,\tau^{-1}\,\boldsymbol{y}^T \boldsymbol{x}_1 \;+\; \tau^{-1}\,\mathrm{lse}\left(\tau^{-1}, \tilde{\boldsymbol{X}}^T \boldsymbol{y}\right)\,, \tag{A114}$$

where lse is the log-sum-exponential function.

The gradient with respect to $\boldsymbol{y}$ is

$$\frac{\partial \mathrm{L}_{\mathrm{InfoLOOB}}(\boldsymbol{y})}{\partial \boldsymbol{y}} \;=\; -\,\tau^{-1}\,\boldsymbol{x}_1 \;+\; \tau^{-1}\,\tilde{\boldsymbol{X}}\,\operatorname{softmax}\left(\tau^{-1}\tilde{\boldsymbol{X}}^T \boldsymbol{y}\right)\,, \tag{A115}$$

which is the positive example $\boldsymbol{x}_1$ that fits to the anchor example $\boldsymbol{y}$ minus the Hopfield network update with state pattern $\boldsymbol{y}$ and stored patterns $\tilde{\boldsymbol{X}}$ and then this difference multiplied by $\tau^{-1}$.

The gradient with respect to $\boldsymbol{x}_1$ is

$$\frac{\partial \mathrm{L}_{\mathrm{InfoLOOB}}(\boldsymbol{y})}{\partial \boldsymbol{x}_1} \;=\; -\,\tau^{-1}\,\boldsymbol{y}\,. \tag{A116}$$

The sum of gradients with respect to $\boldsymbol{x}_1$ and $\boldsymbol{x}_i$ is

$$\frac{\partial \mathrm{L}_{\mathrm{InfoLOOB}}(\boldsymbol{y})}{\partial \boldsymbol{x}_1} \;+\; \sum_i \frac{\partial \mathrm{L}_{\mathrm{InfoLOOB}}(\boldsymbol{y})}{\partial \boldsymbol{x}_i} \;=\; -\,\tau^{-1}\,\boldsymbol{y} \;+\; \tau^{-1}\,\boldsymbol{y}\,\mathbf{1}^T \operatorname{softmax}\left(\tau^{-1}\tilde{\boldsymbol{X}}^T \boldsymbol{y}\right) \tag{A117}$$

$$=\; -\,\tau^{-1}\,\boldsymbol{y} \;+\; \tau^{-1}\,\boldsymbol{y} \;=\; 0\,,$$

where $\mathbf{1}$ is the vector with ones. However, the derivatives with respect to the weights are not zero since the $\boldsymbol{x}_i$ are differently computed.

**Gradients with respect to $\tau^{-1}$.**

The gradient of the InfoNCE loss Eq. (A98) using the similarity Eq. (A100) with respect to $\tau^{-1}$ is

$$\frac{\partial \mathrm{L}_{\mathrm{InfoNCE}}(\boldsymbol{y})}{\partial \tau^{-1}} \;=\; -\,\boldsymbol{y}^T\,\boldsymbol{x}_1 \;+\; \boldsymbol{y}^T\,\boldsymbol{X}\,\operatorname{softmax}\left(\tau^{-1}\boldsymbol{X}^T \boldsymbol{y}\right) \tag{A118}$$

$$=\; -\,\boldsymbol{y}^T\,\left(\boldsymbol{x}_1 \;-\; \boldsymbol{X}\,\operatorname{softmax}\left(\tau^{-1}\boldsymbol{X}^T \boldsymbol{y}\right)\right)\,, \tag{A119}$$

which is the similarity of the anchor $\boldsymbol{y}$ with the difference of the positive example $\boldsymbol{x}_1$ and the Hopfield network update with state pattern $\boldsymbol{y}$ and stored patterns $\boldsymbol{X}$. The gradient of the InfoLOOB loss Eq. (A110) using the similarity Eq. (A112) with respect to $\tau^{-1}$ is

$$\frac{\partial \mathrm{L}_{\mathrm{InfoLOOB}}(\boldsymbol{y})}{\partial \tau^{-1}} \;=\; -\,\boldsymbol{y}^T\,\boldsymbol{x}_1 \;+\; \boldsymbol{y}^T\,\tilde{\boldsymbol{X}}\,\operatorname{softmax}\left(\tau^{-1}\tilde{\boldsymbol{X}}^T \boldsymbol{y}\right) \tag{A120}$$

$$=\; -\,\boldsymbol{y}^T\,\left(\boldsymbol{x}_1 \;-\; \tilde{\boldsymbol{X}}\,\operatorname{softmax}\left(\tau^{-1}\tilde{\boldsymbol{X}}^T \boldsymbol{y}\right)\right)\,. \tag{A121}$$

with the difference that the Hopfield network update is done with stored patterns $\tilde{\boldsymbol{X}}$ instead of $\boldsymbol{X}$.

Without the positive example $\boldsymbol{x}_1$ in the stored patterns $\tilde{\boldsymbol{X}}$, the term $\boldsymbol{x}_1 - \tilde{\boldsymbol{X}}\,\operatorname{softmax}\left(\tau^{-1}\tilde{\boldsymbol{X}}^T \boldsymbol{y}\right)$ in Eq. (A120) will not decrease like the term $\boldsymbol{x}_1 - \boldsymbol{X}\,\operatorname{softmax}\left(\tau^{-1}\boldsymbol{X}^T \boldsymbol{y}\right)$ in Eq. (A118) but grow even larger with better separation of the positive and negative examples.

### A.1.5 INFOLOOB AND INFONCE: PROBABILITY ESTIMATORS

In McAllester & Stratos (2018; 2020) it was shown that estimators of the mutual information by lower bounds have problems as they come with serious statistical limitations. Statistically more justified for

representing the mutual information is a difference of entropies, which are estimated by minimizing the cross-entropy loss. Both InfoNCE and InfoLOOB losses can be viewed as cross-entropy losses.

We sample $N$ pairs independently from $p(\boldsymbol{x}, \boldsymbol{y})$, which gives $Z = \{(\boldsymbol{x}_1, \boldsymbol{y}_1), (\boldsymbol{x}_2, \boldsymbol{y}_2), \dots, (\boldsymbol{x}_N, \boldsymbol{y}_N)\}$. We set $X = \{\boldsymbol{x}_1, \boldsymbol{x}_2, \dots, \boldsymbol{x}_N\}$ and $Y = \{\boldsymbol{y}_1, \boldsymbol{y}_2, \dots, \boldsymbol{y}_N\}$, so that, $Z = X \times Y$. The score function $f(\boldsymbol{x}, \boldsymbol{y})$ is an estimator for $p(\boldsymbol{x}, \boldsymbol{y})$. Then we obtain estimators $\hat{q}$ for the conditional probabilities. $\hat{q}(\boldsymbol{y}_i \mid \boldsymbol{x}_i, Y \setminus \{\boldsymbol{y}_i\})$ is an estimator for $p(\boldsymbol{y}_i \mid \boldsymbol{x}_i)$ and $\hat{q}(\boldsymbol{x}_i \mid \boldsymbol{y}_i, X \setminus \{\boldsymbol{x}_i\})$ an estimator for $p(\boldsymbol{x}_i \mid \boldsymbol{y}_i)$. Each estimator $\hat{q}$ uses beyond $(\boldsymbol{x}_i, \boldsymbol{y}_i)$ additional samples to estimate the normalizing constant. For InfoNCE these estimators are

$$\hat{q}^1(\boldsymbol{y}_i \mid \boldsymbol{x}_i, Y \setminus \{\boldsymbol{y}_i\}) = \frac{f(\boldsymbol{x}_i, \boldsymbol{y}_i)}{\frac{1}{N} \sum_{j=1}^{N} f(\boldsymbol{x}_i, \boldsymbol{y}_j)} \approx \frac{f(\boldsymbol{x}_i, \boldsymbol{y}_i)}{\mathrm{E}_{p(\boldsymbol{y})}\left[f(\boldsymbol{x}_i, \boldsymbol{y})\right]}, \tag{A122}$$

$$\hat{q}^2(\boldsymbol{x}_i \mid \boldsymbol{y}_i, X \setminus \{\boldsymbol{x}_i\}) = \frac{f(\boldsymbol{x}_i, \boldsymbol{y}_i)}{\frac{1}{N} \sum_{j=1}^{N} f(\boldsymbol{x}_j, \boldsymbol{y}_i)} \approx \frac{f(\boldsymbol{x}_i, \boldsymbol{y}_i)}{\mathrm{E}_{p(\boldsymbol{x})}\left[f(\boldsymbol{x}, \boldsymbol{y}_i)\right]} . \tag{A123}$$

The cross-entropy losses for the InfoNCE estimators are

$$\mathrm{L}_{\mathrm{InfoNCE}}^1 = -\frac{1}{N} \sum_{i=1}^{N} \ln \left( \frac{f(\boldsymbol{x}_i, \boldsymbol{y}_i)}{\frac{1}{N} \sum_{j=1}^{N} f(\boldsymbol{x}_i, \boldsymbol{y}_j)} \right) , \tag{A124}$$

$$\mathrm{L}_{\mathrm{InfoNCE}}^2 = -\frac{1}{N} \sum_{i=1}^{N} \ln \left( \frac{f(\boldsymbol{x}_i, \boldsymbol{y}_i)}{\frac{1}{N} \sum_{j=1}^{N} f(\boldsymbol{x}_j, \boldsymbol{y}_i)} \right) . \tag{A125}$$

For InfoLOOB these estimators are

$$\hat{q}^1(\boldsymbol{y}_i \mid \boldsymbol{x}_i, Y \setminus \{\boldsymbol{y}_i\}) = \frac{f(\boldsymbol{x}_i, \boldsymbol{y}_i)}{\frac{1}{N-1} \sum_{j=1, j \neq i}^{N} f(\boldsymbol{x}_i, \boldsymbol{y}_j)} \approx \frac{f(\boldsymbol{x}_i, \boldsymbol{y}_i)}{\mathrm{E}_{p(\boldsymbol{y})}\left[f(\boldsymbol{x}_i, \boldsymbol{y})\right]} , \tag{A126}$$

$$\hat{q}^2(\boldsymbol{x}_i \mid \boldsymbol{y}_i, X \setminus \{\boldsymbol{x}_i\}) = \frac{f(\boldsymbol{x}_i, \boldsymbol{y}_i)}{\frac{1}{N-1} \sum_{j=1, j \neq i}^{N} f(\boldsymbol{x}_j, \boldsymbol{y}_i)} \approx \frac{f(\boldsymbol{x}_i, \boldsymbol{y}_i)}{\mathrm{E}_{p(\boldsymbol{x})}\left[f(\boldsymbol{x}, \boldsymbol{y}_i)\right]} . \tag{A127}$$

The cross-entropy losses for the InfoLOOB estimators are

$$\mathrm{L}_{\mathrm{InfoLOOB}}^1 = -\frac{1}{N} \sum_{i=1}^{N} \ln \left( \frac{f(\boldsymbol{x}_i, \boldsymbol{y}_i)}{\frac{1}{N-1} \sum_{j=1, j \neq i}^{N} f(\boldsymbol{x}_i, \boldsymbol{y}_j)} \right) , \tag{A128}$$

$$\mathrm{L}_{\mathrm{InfoLOOB}}^2 = -\frac{1}{N} \sum_{i=1}^{N} \ln \left( \frac{f(\boldsymbol{x}_i, \boldsymbol{y}_i)}{\frac{1}{N-1} \sum_{j=1, j \neq i}^{N} f(\boldsymbol{x}_j, \boldsymbol{y}_i)} \right) . \tag{A129}$$

The InfoLOOB estimator uses for normalization

$$\mathrm{E}_{p(\boldsymbol{x})}\left[f(\boldsymbol{x}, \boldsymbol{y}_i)\right] \approx \frac{1}{N-1} \sum_{j=1, j \neq i}^{N} f(\boldsymbol{x}_j, \boldsymbol{y}_i) , \tag{A130}$$

$$\mathrm{E}_{p(\boldsymbol{y})}\left[f(\boldsymbol{x}_i, \boldsymbol{y})\right] \approx \frac{1}{N-1} \sum_{j=1, j \neq i}^{N} f(\boldsymbol{x}_i, \boldsymbol{y}_j) , \tag{A131}$$

in contrast to InfoNCE, which uses

$$\mathrm{E}_{p(\boldsymbol{x})}\left[f(\boldsymbol{x}, \boldsymbol{y}_i)\right] \approx \frac{1}{N} \sum_{j=1}^{N} f(\boldsymbol{x}_j, \boldsymbol{y}_i) , \tag{A132}$$

$$\mathrm{E}_{p(\boldsymbol{y})}\left[f(\boldsymbol{x}_i, \boldsymbol{y})\right] \approx \frac{1}{N} \sum_{j=1}^{N} f(\boldsymbol{x}_i, \boldsymbol{y}_j) . \tag{A133}$$

If InfoNCE estimates the normalizing constant separately, then it would be biased. $(\boldsymbol{x}_i, \boldsymbol{y}_i)$ is drawn according to $p(\boldsymbol{x}_i, \boldsymbol{y}_i)$ instead of $p(\boldsymbol{x}_i)p(\boldsymbol{y}_i)$. In contrast, if InfoLOOB estimated the normalizing constant separately, then it would be unbiased.

A.1.6 INFOLOOB AND INFONCE: LOSSES

We have $N$ pairs drawn iid from $p(\boldsymbol{x}, \boldsymbol{y})$, where we assume that a pair $(\boldsymbol{x}_i, \boldsymbol{y}_i)$ is already an embedding of the original drawn pair. These build up the embedding training set $Z = \{(\boldsymbol{x}_1, \boldsymbol{y}_1), (\boldsymbol{x}_2, \boldsymbol{y}_2), \ldots, (\boldsymbol{x}_N, \boldsymbol{y}_N)\}$ that allows to construct the matrices $\boldsymbol{X} = (\boldsymbol{x}_1, \boldsymbol{x}_2, \ldots, \boldsymbol{x}_N)$ of $N$ embedding samples $\boldsymbol{x}_i$ and $\boldsymbol{Y} = (\boldsymbol{y}_1, \boldsymbol{y}_2, \ldots, \boldsymbol{y}_N)$ of $N$ embedding samples $\boldsymbol{y}_i$. We also have $M$ stored patterns $\boldsymbol{U} = (\boldsymbol{u}_1, \ldots, \boldsymbol{u}_M)$ and $K$ stored patterns $\boldsymbol{V} = (\boldsymbol{v}_1, \ldots, \boldsymbol{v}_K)$.

The state vectors $\boldsymbol{x}_i$ and $\boldsymbol{y}_i$ are the queries for the Hopfield networks, which retrieve some vectors from $\boldsymbol{U}$ or $\boldsymbol{V}$. We normalize vectors $\|\boldsymbol{x}_i\| = \|\boldsymbol{y}_i\| = \|\boldsymbol{u}_i\| = \|\boldsymbol{v}_i\| = 1$. The following vectors are retrieved from modern Hopfield networks (Ramsauer et al., 2021):

$$\boldsymbol{U}_{\boldsymbol{x}_i} = \boldsymbol{U} \operatorname{softmax}(\beta \, \boldsymbol{U}^T \boldsymbol{x}_i) , \quad \boldsymbol{U}_{\boldsymbol{y}_i} = \boldsymbol{U} \operatorname{softmax}(\beta \, \boldsymbol{U}^T \boldsymbol{y}_i) , \tag{A134}$$

$$\boldsymbol{V}_{\boldsymbol{x}_i} = \boldsymbol{V} \operatorname{softmax}(\beta \, \boldsymbol{V}^T \boldsymbol{x}_i) , \quad \boldsymbol{V}_{\boldsymbol{y}_i} = \boldsymbol{V} \operatorname{softmax}(\beta \, \boldsymbol{V}^T \boldsymbol{y}_i) \tag{A135}$$

where $\boldsymbol{U}_{\boldsymbol{x}_i}$ denotes an image-retrieved image embedding, $\boldsymbol{U}_{\boldsymbol{y}_i}$ a text-retrieved image embedding, $\boldsymbol{V}_{\boldsymbol{x}_i}$ an image-retrieved text embedding and $\boldsymbol{V}_{\boldsymbol{y}_i}$ a text-retrieved text embedding. The hyperparameter $\beta$ corresponds to the inverse temperature: $\beta = 0$ retrieves the average of the stored pattern, while large $\beta$ retrieve the stored pattern that is most similar to the state pattern (query).

We consider the loss functions

$$\mathrm{L}_{\mathrm{InfoNCE}} = -\frac{1}{N} \sum_{i=1}^{N} \log \frac{\exp(\tau^{-1} \, \boldsymbol{x}_i^T \boldsymbol{y}_i)}{\sum_{j=1}^{N} \exp(\tau^{-1} \, \boldsymbol{x}_i^T \boldsymbol{y}_j)} - \frac{1}{N} \sum_{i=1}^{N} \log \frac{\exp(\tau^{-1} \, \boldsymbol{x}_i^T \boldsymbol{y}_i)}{\sum_{j=1}^{N} \exp(\tau^{-1} \, \boldsymbol{x}_j^T \boldsymbol{y}_i)} , \tag{A136}$$

$$\mathrm{L}_{\mathrm{InfoLOOB}} = -\frac{1}{N} \sum_{i=1}^{N} \log \frac{\exp(\tau^{-1} \, \boldsymbol{x}_i^T \boldsymbol{y}_i)}{\sum_{j \neq i}^{N} \exp(\tau^{-1} \, \boldsymbol{x}_i^T \boldsymbol{y}_j)} - \frac{1}{N} \sum_{i=1}^{N} \log \frac{\exp(\tau^{-1} \, \boldsymbol{x}_i^T \boldsymbol{y}_i)}{\sum_{j \neq i}^{N} \exp(\tau^{-1} \, \boldsymbol{x}_j^T \boldsymbol{y}_i)} , \tag{A137}$$

$$\mathrm{L}_{\mathrm{InfoLOOB}}^{\mathrm{H-UVUV}} = -\frac{1}{N} \sum_{i=1}^{N} \log \frac{\exp(\tau^{-1} \, \boldsymbol{U}_{\boldsymbol{x}_i}^T \boldsymbol{V}_{\boldsymbol{y}_i})}{\sum_{j \neq i}^{N} \exp(\tau^{-1} \, \boldsymbol{U}_{\boldsymbol{x}_i}^T \boldsymbol{V}_{\boldsymbol{y}_j})} - \frac{1}{N} \sum_{i=1}^{N} \log \frac{\exp(\tau^{-1} \, \boldsymbol{U}_{\boldsymbol{x}_i}^T \boldsymbol{V}_{\boldsymbol{y}_i})}{\sum_{j \neq i}^{N} \exp(\tau^{-1} \, \boldsymbol{U}_{\boldsymbol{x}_j}^T \boldsymbol{V}_{\boldsymbol{y}_i})} , \tag{A138}$$

$$\mathrm{L}_{\mathrm{InfoLOOB}}^{\mathrm{H-UUVV}} = -\frac{1}{N} \sum_{i=1}^{N} \log \frac{\exp(\tau^{-1} \, \boldsymbol{U}_{\boldsymbol{x}_i}^T \boldsymbol{U}_{\boldsymbol{y}_i})}{\sum_{j \neq i}^{N} \exp(\tau^{-1} \, \boldsymbol{U}_{\boldsymbol{x}_i}^T \boldsymbol{U}_{\boldsymbol{y}_j})} - \frac{1}{N} \sum_{i=1}^{N} \log \frac{\exp(\tau^{-1} \, \boldsymbol{V}_{\boldsymbol{x}_i}^T \boldsymbol{V}_{\boldsymbol{y}_i})}{\sum_{j \neq i}^{N} \exp(\tau^{-1} \, \boldsymbol{V}_{\boldsymbol{x}_j}^T \boldsymbol{V}_{\boldsymbol{y}_i})} , \tag{A139}$$

where for InfoLOOB the sum $\sum_{j \neq i}$ in the denominator contains only negative examples $j$. We do not consider the loss function $\mathrm{L}_{\mathrm{InfoLOOB}}^{\mathrm{H-UVUV}}$ because of the high variance in the dot product $\boldsymbol{U}_{\boldsymbol{x}_i}^T \boldsymbol{V}_{\boldsymbol{y}_i}$ as elaborated in the following.

Let us consider the dot product between the anchor retrieval with the positive pattern retrieval for the loss functions with Hopfield. In the first term of the loss function Eq. (A138), $\boldsymbol{U}_{\boldsymbol{x}_i}$ is the anchor with $\boldsymbol{V}_{\boldsymbol{y}_i}$ as the positive sample and $\boldsymbol{V}_{\boldsymbol{y}_i}$ with $\boldsymbol{U}_{\boldsymbol{x}_i}$ as the positive sample for the second term, since the anchor also appears in each term of the denominator. Equivalently the same is valid for Eq. (A139), but with positive samples $\boldsymbol{V}_{\boldsymbol{x}_i}$ and $\boldsymbol{U}_{\boldsymbol{y}_i}$ respectively. These dot products can be written as

$$\boldsymbol{U}_{\boldsymbol{x}_i}^T \boldsymbol{V}_{\boldsymbol{y}_i} = \operatorname{softmax}(\beta \, \boldsymbol{U}^T \boldsymbol{x}_i)^T \, \boldsymbol{U}^T \boldsymbol{V} \operatorname{softmax}(\beta \, \boldsymbol{V}^T \boldsymbol{y}_i) , \tag{A140}$$

$$\boldsymbol{U}_{\boldsymbol{x}_i}^T \boldsymbol{U}_{\boldsymbol{y}_i} = \operatorname{softmax}(\beta \, \boldsymbol{U}^T \boldsymbol{x}_i)^T \, \boldsymbol{U}^T \boldsymbol{U} \operatorname{softmax}(\beta \, \boldsymbol{U}^T \boldsymbol{y}_i) , \tag{A141}$$

$$\boldsymbol{V}_{\boldsymbol{x}_i}^T \boldsymbol{V}_{\boldsymbol{y}_i} = \operatorname{softmax}(\beta \, \boldsymbol{V}^T \boldsymbol{x}_i)^T \, \boldsymbol{V}^T \boldsymbol{V} \operatorname{softmax}(\beta \, \boldsymbol{V}^T \boldsymbol{y}_i) . \tag{A142}$$

**High variance of $\boldsymbol{U}_{\boldsymbol{x}_i}^T \boldsymbol{V}_{\boldsymbol{y}_i}$.** To compute the dot product $\boldsymbol{U}_{\boldsymbol{x}_i}^T \boldsymbol{V}_{\boldsymbol{y}_i}$, $M + K$ stored patterns are required ($M$ of the $\boldsymbol{u}_j$ and $K$ of the $\boldsymbol{v}_j$). In contrast, the dot products $\boldsymbol{U}_{\boldsymbol{x}_i}^T \boldsymbol{U}_{\boldsymbol{y}_i}$ and $\boldsymbol{V}_{\boldsymbol{x}_i}^T \boldsymbol{V}_{\boldsymbol{y}_i}$ require only $M$ or respectively $K$ stored patterns. Therefore, $\boldsymbol{U}_{\boldsymbol{x}_i}^T \boldsymbol{V}_{\boldsymbol{y}_i}$ has higher variance than both $\boldsymbol{U}_{\boldsymbol{x}_i}^T \boldsymbol{U}_{\boldsymbol{y}_i}$ and $\boldsymbol{V}_{\boldsymbol{x}_i}^T \boldsymbol{V}_{\boldsymbol{y}_i}$.

**Covariance structure extracted by $\boldsymbol{U}_{\boldsymbol{x}_i}^T \boldsymbol{U}_{\boldsymbol{y}_i}$ and $\boldsymbol{V}_{\boldsymbol{x}_i}^T \boldsymbol{V}_{\boldsymbol{y}_i}$.**

The Jacobian J of the softmax $\boldsymbol{p} = \mathrm{softmax}(\beta \boldsymbol{a})$ is

$$\mathrm{J}(\beta \boldsymbol{a}) \;=\; \frac{\partial \mathrm{softmax}(\beta \boldsymbol{a})}{\partial \boldsymbol{a}} \;=\; \beta \left( \mathrm{diag}(\boldsymbol{p}) - \boldsymbol{p}\boldsymbol{p}^T \right) \;, \tag{A143}$$

which is a symmetric, positive semi-definite matrix with one eigenvalue of zero for eigenvector $\mathbf{1}$. $\mathrm{J}(\beta \boldsymbol{a})$ is diagonally dominant since $|p_i(1 - p_i)| - \sum_{j \neq i} |p_i p_j| = p_i - \sum_j p_i p_j = p_i - p_i = 0$.

Next we give upper bounds on the norm of J.

**Lemma A1.** *For a softmax $\boldsymbol{p} = \mathrm{softmax}(\beta \boldsymbol{x})$ with $m = \max_i p_i(1 - p_i)$, the spectral norm of the Jacobian* J *of the softmax is bounded:*

$$\|\mathrm{J}\|_2 \;\leqslant\; 2\, m\, \beta \,, \tag{A144}$$
$$\|\mathrm{J}\|_1 \;\leqslant\; 2\, m\, \beta \,, \tag{A145}$$
$$\|\mathrm{J}\|_\infty \;\leqslant\; 2\, m\, \beta \,. \tag{A146}$$

*In particular everywhere holds*

$$\|\mathrm{J}\|_2 \;\leqslant\; \frac{1}{2}\, \beta \,. \tag{A147}$$

*If $p_{\max} = \max_i p_i \geq 1 - \epsilon \geq 0.5$, then for the spectral norm of the Jacobian holds*

$$\|\mathrm{J}\|_2 \;\leqslant\; 2\, \epsilon\, \beta \;-\; 2\, \epsilon^2\, \beta \;<\; 2\, \epsilon\, \beta \,. \tag{A148}$$

*Proof.* We consider the maximum absolute column sum norm

$$\|\boldsymbol{A}\|_1 \;=\; \max_j \sum_i |a_{ij}| \tag{A149}$$

and the maximum absolute row sum norm

$$\|\boldsymbol{A}\|_\infty \;=\; \max_i \sum_j |a_{ij}| \,. \tag{A150}$$

We have for $\boldsymbol{A} = \mathrm{J} = \beta \left( \mathrm{diag}(\boldsymbol{p}) - \boldsymbol{p}\boldsymbol{p}^T \right)$

$$\sum_j |a_{ij}| \;=\; \beta \left( p_i(1 - p_i) + \sum_{j, j \neq i} p_i p_j \right) \;=\; \beta\, p_i \left( 1 - 2p_i + \sum_j p_j \right) \tag{A151}$$
$$=\; 2\, \beta\, p_i\, (1 - p_i) \;\leqslant\; 2\, m\, \beta \,,$$

$$\sum_i |a_{ij}| \;=\; \beta \left( p_j(1 - p_j) + \sum_{i, i \neq j} p_j p_i \right) \;=\; \beta\, p_j \left( 1 - 2p_j + \sum_i p_i \right) \tag{A152}$$
$$=\; 2\, \beta\, p_j\, (1 - p_j) \;\leqslant\; 2\, m\, \beta \,.$$

Therefore, we have

$$\|\mathrm{J}\|_1 \;\leqslant\; 2\, m\, \beta \,, \tag{A153}$$
$$\|\mathrm{J}\|_\infty \;\leqslant\; 2\, m\, \beta \,, \tag{A154}$$
$$\|\mathrm{J}\|_2 \;\leqslant\; \sqrt{\|\mathrm{J}\|_1 \|\mathrm{J}\|_\infty} \;\leqslant\; 2\, m\, \beta \,. \tag{A155}$$

The last inequality is a direct consequence of Hölder's inequality.

For $0 \leqslant p_i \leqslant 1$, we have $p_i(1 - p_i) \leqslant 0.25$. Therefore, $m \leqslant 0.25$ for all values of $p_i$.

If $p_{\max} \geq 1 - \epsilon \geq 0.5$ ($\epsilon \leqslant 0.5$), then $1 - p_{\max} \leqslant \epsilon$ and for $p_i \neq p_{\max}$ $p_i \leqslant \epsilon$. The derivative $\partial x(1 - x)/\partial x = 1 - 2x > 0$ for $x < 0.5$, therefore $x(1 - x)$ increases with $x$ for $x < 0.5$. Using $x = 1 - p_{\max}$ and for $p_i \neq p_{\max}$ $x = p_i$, we obtain $p_i(1 - p_i) \leqslant \epsilon(1 - \epsilon)$ for all $i$. Consequently, we have $m \leqslant \epsilon(1 - \epsilon)$. □

For the softmax $\boldsymbol{p} = \mathrm{softmax}(\beta \boldsymbol{a})$ with Jacobian $\partial \mathrm{J} / \partial \boldsymbol{a} = \mathrm{J}(\beta \boldsymbol{a}) = \beta \left( \mathrm{diag}(\boldsymbol{p}) - \boldsymbol{p} \boldsymbol{p}^T \right)$ and for arbitrary $N$-dimensional vectors $\boldsymbol{b}$ and $\boldsymbol{c}$, we have

$$\boldsymbol{b}^T \mathrm{J}(\beta \boldsymbol{a}) \, \boldsymbol{c} \; = \; \beta \, \boldsymbol{b}^T \left( \mathrm{diag}(\boldsymbol{p}) \; - \; \boldsymbol{p} \, \boldsymbol{p}^T \right) \, \boldsymbol{c} \; = \; \beta \, \left( \sum_i p_i \, b_i \, c_i \; - \; \left( \sum_i p_i \, b_i \right) \left( \sum_i p_i \, c_i \right) \right) . \tag{A156}$$

Therefore, $\boldsymbol{b}^T \mathrm{J}(\beta \boldsymbol{a}) \boldsymbol{c}$ is $\beta$ times the covariance between $\boldsymbol{b}$ and $\boldsymbol{c}$ if component $i$ is drawn with probability $p_i$ of the multinomial distribution $\boldsymbol{p}$. In our case the component $i$ is sample $i$.

Using the mean $\hat{\boldsymbol{u}} = 1/M \sum_{i=1}^M \boldsymbol{u}_i$, the empirical covariance of data $\boldsymbol{U}$ is

$$\mathrm{Cov}(\boldsymbol{U}) \; = \; 1/M \, \boldsymbol{U} \, \boldsymbol{U}^T \; - \; \hat{\boldsymbol{u}} \, \hat{\boldsymbol{u}}^T \, , \tag{A157}$$

$$[\mathrm{Cov}(\boldsymbol{U})]_{kl} \; = \; \sum_{i=1}^M 1/M \, u_{ik} \, u_{il} \; - \; \left( \sum_{i=1}^M 1/M \, u_{ik} \right) \left( \sum_{i=1}^M 1/M \, u_{il} \right) . \tag{A158}$$

The weighted covariance (samples $\boldsymbol{u}_i$ are drawn according to $p_i$)

$$\mathrm{Cov}(\boldsymbol{U}) \; = \; \boldsymbol{U} \, \mathrm{J}(\beta \, \boldsymbol{a}) \, \boldsymbol{U}^T \, , \tag{A159}$$

$$[\mathrm{Cov}(\boldsymbol{U})]_{kl} \; = \; \beta \, \left( \sum_{i=1}^M p_i \, u_{ik} \, u_{il} \; - \; \left( \sum_{i=1}^M p_i \, u_{ik} \right) \left( \sum_{i=1}^M p_i \, u_{il} \right) \right) , \tag{A160}$$

which replaces $1/M$ from equal sampling by the $p_i$, that is, $\boldsymbol{u}_i$ is sampled with probability $p_i$.

The next theorem states how to express the dot product $\boldsymbol{U}_{\boldsymbol{x}_i}^T \boldsymbol{U}_{\boldsymbol{y}_i}$ by weighted covariances of the data $\boldsymbol{U}$.

**Theorem A3** (Weighted Covariances). *Using the weighted covariances*

$$\mathrm{Cov}(\boldsymbol{U}, \boldsymbol{y}_i) \; = \; \boldsymbol{U} \, \mathrm{J}^{\mathrm{m}}(\beta \, \boldsymbol{U}^T \boldsymbol{y}_i) \, \boldsymbol{U}^T \, , \quad \mathrm{Cov}(\boldsymbol{U}, \boldsymbol{x}_i) \; = \; \boldsymbol{U} \, \mathrm{J}^{\mathrm{m}}(\beta \, \boldsymbol{U}^T \boldsymbol{x}_i) \, \boldsymbol{U}^T \, , \tag{A161}$$

$$\mathrm{J}^{\mathrm{m}}(\beta \, \boldsymbol{a}) \; = \; \int_0^1 \mathrm{J}(\lambda \, \beta \, \boldsymbol{a}) \, \mathrm{d}\lambda \, , \tag{A162}$$

*where the mean Jacobian $\mathrm{J}^{\mathrm{m}}$ is symmetric, diagonally dominant, and positive semi-definite with spectral norm bounded by $\|\mathrm{J}^{\mathrm{m}}\|_2 \leqslant 0.5\beta$.*

*The dot product $\boldsymbol{U}_{\boldsymbol{x}_i}^T \boldsymbol{U}_{\boldsymbol{y}_i}$ can be expressed by the weighted covariances*

$$\boldsymbol{U}_{\boldsymbol{x}_i}^T \boldsymbol{U}_{\boldsymbol{y}_i} \; = \; \left( \bar{\boldsymbol{u}} \; + \; \mathrm{Cov}(\boldsymbol{U}, \boldsymbol{x}_i) \, \boldsymbol{x}_i \right)^T \, \left( \bar{\boldsymbol{u}} \; + \; \mathrm{Cov}(\boldsymbol{U}, \boldsymbol{y}_i) \, \boldsymbol{y}_i \right) , \tag{A163}$$

*where the mean is $\bar{\boldsymbol{u}} = 1/M \boldsymbol{U} \boldsymbol{1}$.*

*Proof.* We apply the mean value theorem to the softmax with the symmetric, diagonally dominant, positive semi-definite Jacobian matrix $\mathrm{J}^{\mathrm{m}} = \int_0^1 \mathrm{J}(\lambda \boldsymbol{a} + (1 - \lambda) \boldsymbol{a}') \, \mathrm{d}\lambda$:

$$\mathrm{softmax}(\boldsymbol{a}) \; - \; \mathrm{softmax}(\boldsymbol{a}') \; = \; \mathrm{J}^{\mathrm{m}} \, (\boldsymbol{a} \; - \; \boldsymbol{a}') \, . \tag{A164}$$

We set $\boldsymbol{a}' = \boldsymbol{0}$ and use $\beta \boldsymbol{a}$ instead of $\boldsymbol{a}$, which gives:

$$\mathrm{softmax}(\beta \, \boldsymbol{a}) \; = \; 1/M \, \boldsymbol{1} \; + \; \mathrm{J}^{\mathrm{m}}(\beta \, \boldsymbol{a}) \, \boldsymbol{a} \, , \quad \mathrm{J}^{\mathrm{m}}(\beta \, \boldsymbol{a}) \; = \; \int_0^1 \mathrm{J}(\lambda \, \beta \, \boldsymbol{a}) \, \mathrm{d}\lambda \, , \tag{A165}$$

which is exact. We obtain

$$\mathrm{softmax}(\beta \, \boldsymbol{U}^T \boldsymbol{x}_i) \; = \; 1/M \, \boldsymbol{1} \; + \; \mathrm{J}^{\mathrm{m}}(\beta \, \boldsymbol{U}^T \boldsymbol{x}_i) \, \boldsymbol{U}^T \boldsymbol{x}_i \, , \tag{A166}$$

$$\mathrm{softmax}(\beta \, \boldsymbol{U}^T \boldsymbol{y}_i) \; = \; 1/M \, \boldsymbol{1} \; + \; \mathrm{J}^{\mathrm{m}}(\beta \, \boldsymbol{U}^T \boldsymbol{y}_i) \, \boldsymbol{U}^T \boldsymbol{y}_i \, . \tag{A167}$$

The spectral norm of $\mathrm{J}^{\mathrm{m}}$ is bounded by $\|\mathrm{J}^{\mathrm{m}}\|_2 \leqslant 0.5\beta$, since this bound holds for every $\mathrm{J}(\lambda \beta \boldsymbol{a})$ in $\mathrm{J}^{\mathrm{m}}(\beta \, \boldsymbol{a}) = \int_0^1 \mathrm{J}(\lambda \beta \boldsymbol{a}) \, \mathrm{d}\lambda$ according to Lemma A1.

The dot product between the anchor retrieval and the positive sample is:

$$
\begin{aligned}
\boldsymbol{U}_{\boldsymbol{x}_i}^T \boldsymbol{U}_{\boldsymbol{y}_i} &= \operatorname{softmax}(\beta\, \boldsymbol{U}^T \boldsymbol{x}_i)^T\ \boldsymbol{U}^T \boldsymbol{U}\, \operatorname{softmax}(\beta\, \boldsymbol{U}^T \boldsymbol{y}_i) && \text{(A168)} \\
&= \left(1/M\, \boldsymbol{1}\ +\ \mathrm{J^m}(\beta\, \boldsymbol{U}^T \boldsymbol{x}_i)\, \boldsymbol{U}^T \boldsymbol{x}_i\right)^T\ \boldsymbol{U}^T \boldsymbol{U}\ \left(1/M\, \boldsymbol{1}\ +\ \mathrm{J^m}(\beta\, \boldsymbol{U}^T \boldsymbol{y}_i)\, \boldsymbol{U}^T \boldsymbol{y}_i\right) \\
&= \left(1/M\, \boldsymbol{U}\, \boldsymbol{1}\ +\ \boldsymbol{U}\, \mathrm{J^m}(\beta\, \boldsymbol{U}^T \boldsymbol{x}_i)\, \boldsymbol{U}^T \boldsymbol{x}_i\right)^T\ \left(1/M\, \boldsymbol{U}\, \boldsymbol{1}\ +\ \boldsymbol{U}\, \mathrm{J^m}(\beta\, \boldsymbol{U}^T \boldsymbol{y}_i)\, \boldsymbol{U}^T \boldsymbol{y}_i\right) \\
&= \left(\bar{\boldsymbol{u}}\ +\ \operatorname{Cov}(\boldsymbol{U}, \boldsymbol{x}_i)\, \boldsymbol{x}_i\right)^T\ \left(\bar{\boldsymbol{u}}\ +\ \operatorname{Cov}(\boldsymbol{U}, \boldsymbol{y}_i)\, \boldsymbol{y}_i\right)\ ,
\end{aligned}
$$

where we used the mean $\bar{\boldsymbol{u}} = 1/M \boldsymbol{U} \boldsymbol{1}$ and the weighted covariances

$$
\operatorname{Cov}(\boldsymbol{U}, \boldsymbol{y}_i)\ =\ \boldsymbol{U}\, \mathrm{J^m}(\beta\, \boldsymbol{U}^T \boldsymbol{y}_i)\, \boldsymbol{U}^T\ ,\quad \operatorname{Cov}(\boldsymbol{U}, \boldsymbol{x}_i)\ =\ \boldsymbol{U}\, \mathrm{J^m}(\beta\, \boldsymbol{U}^T \boldsymbol{x}_i)\, \boldsymbol{U}^T\ . \tag{A169}
$$

$\square$

The Jacobian $\mathrm{J^m}$ is symmetric, diagonally dominant, and positive semi-definite. The weighted covariance $\operatorname{Cov}(\boldsymbol{U}, .)$ is the covariance if the stored pattern $\boldsymbol{u}_i$ is drawn according to an averaged $p_i$ given by $\mathrm{J^m}(.)$. Analog for weighted covariance $\operatorname{Cov}(\boldsymbol{V}, .)$. When maximizing the dot product $\boldsymbol{U}_{\boldsymbol{x}_i}^T \boldsymbol{U}_{\boldsymbol{y}_i}$, the normalized vectors $\boldsymbol{x}_i$ and $\boldsymbol{y}_i$ are encouraged to agree on drawing the patterns $\boldsymbol{u}_i$ with the same probability $p_i$ to generate similar weighted covariance matrices $\operatorname{Cov}(\boldsymbol{U}, .)$. If subsets of $\boldsymbol{U}$ have a strong covariance structure, then it can be exploited to produce large weighted covariances and, in turn, large dot products of $\boldsymbol{U}_{\boldsymbol{x}_i}^T \boldsymbol{U}_{\boldsymbol{y}_i}$. Furthermore, for a large dot product $\boldsymbol{U}_{\boldsymbol{x}_i}^T \boldsymbol{U}_{\boldsymbol{y}_i}$, $\boldsymbol{x}_i$ and $\boldsymbol{y}_i$ have to be similar to one another to extract the same direction from the covariance matrices. All considerations are analog for $\boldsymbol{V}_{\boldsymbol{x}_i}^T \boldsymbol{V}_{\boldsymbol{y}_i}$.

## A.2 Mutual Information Estimation

We follow the toy experiment discussed in Poole et al. (2019), Belghazi et al. (2018) and Cheng et al. (2020) and experimentally confirm the superior quality of InfoLOOB for mutual information than InfoNCE. The dataset consists of samples $(\boldsymbol{x}_i, \boldsymbol{y}_i)$ drawn jointly from a multivariate Gaussian distribution with correlation $\rho$ where the dimension of the samples $\boldsymbol{x}$ and $\boldsymbol{y}$ is set to $d = 20$. We examine the performance of InfoLoob with and without Hopfield and InfoNCE at estimating mutual information of these samples. Due to the Gaussian distribution, the true value of mutual information can be calculated as $I(\boldsymbol{x}, \boldsymbol{y}) = -\frac{d}{2} \log(1 - \rho^2)$. We set the mutual information true value to the values $(2.0, 4.0, 6.0, 8.0, 10.0, 14.0)$ by varying the value of $\rho$. At each MI true value, we sample data batches 1024 times, with batch size equal to 64, for the training of variational MI estimators. Figure 2 shows that modern Hopfield networks reduce the variance of the model. For models trained on data with mutual information of 10 we observe an average variance of approx. 0.67 for a model without Hopfield and an average variance of approx. 0.33 for a model with Hopfield. For models trained on data with mutual information of 14 we observe an average variance of approx. 1.00 for a model without Hopfield and an average variance of approx. 0.48 for a model with Hopfield.

In Figure A1 we show the performance of our method InfoLOOB with and without Hopfield at estimating mutual information as well as InfoNCE. As expected estimates of InfoNCE have estimates that saturate at log(batch size). InfoLOOB without Hopfield exhibits good estimates of high mutual information while InfoLOOB with Hopfield accomplishes both - good estimates of high mutual information with a decreased variance.

## A.3 Experiments

### A.3.1 Ablation studies

As mentioned in the main paper, CLOOB has two new main components compared to CLIP: (1) the InfoLOOB objective instead of the InfoNCE objective and (2) the modern Hopfield networks. To assess which of the new main components of CLOOB have led to the performance increase over CLIP, we performed ablation studies on the CC dataset. The results are reported in Table A1. First, we enhanced CLIP by replacing the InfoNCE objective with InfoLOOB (see column CLIP InfoLOOB). Next, we added modern Hopfield networks to the CLIP architecture and used retrieved embeddings instead of the original embeddings, while keeping the InfoNCE objective (see column Hopfield InfoNCE). Finally, we add modern Hopfield networks to CLIP and replace the InfoNCE objective

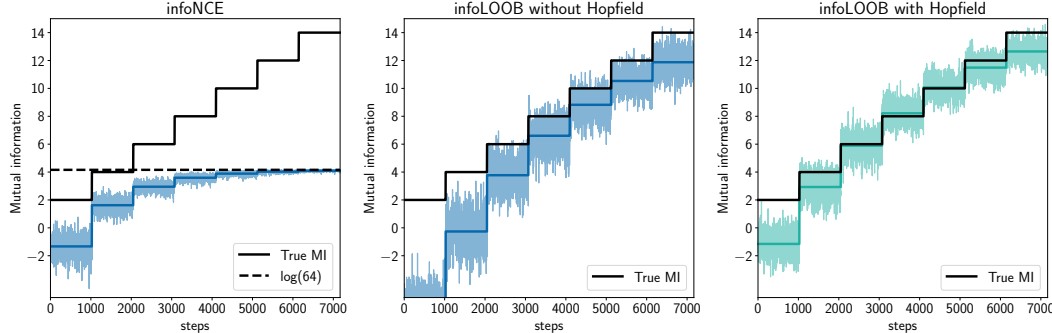

Figure A1: The estimated mutual information of the InfoNCE objective saturates at the batch size induced bound. The InfoLOOB objective trained with the same batch size with samples from the same correlated Gaussian distributions following (Belghazi et al., 2018; Poole et al., 2019; Cheng et al., 2020) is not limited by that bound and better estimates higher mutual information but suffers from higher variance. This is remedied by incorporating the modern Hopfield network.

Table A1: Influence of loss functions and Hopfield retrieval. InfoLOOB increases the performance of CLIP in most of the tasks. The InfoNCE loss is not suited for the Hopfield approach as it saturates leading to a worse performance. Hopfield with InfoLOOB strongly improves the performance in 7 out of 8 datasets compared to both CLIP models.

| | CLIP | | Hopfield | |
| Dataset | InfoNCE | InfoLOOB | InfoNCE | InfoLOOB |
|---|---|---|---|---|
| Birdsnap | 1.94 | 2.37 | 1.67 | **2.53** |
| Country211 | 0.62 | 0.63 | 0.54 | **0.76** |
| Flowers102 | 13.04 | 13.03 | 11.53 | **14.24** |
| GTSRB | **7.28** | 4.39 | 5.76 | 5.86 |
| UCF101 | 21.00 | 19.14 | 20.56 | **22.29** |
| Stanford Cars | 0.90 | 1.33 | 1.24 | **1.37** |
| ImageNet | 20.31 | 22.13 | 19.04 | **24.21** |
| ImageNetV2 | 20.63 | 21.65 | 18.97 | **23.80** |

with InfoLOOB (see column Hopfield InfoLOOB). As shown in Table A1 the InfoLOOB objective increases the performance of CLIP in the majority of the datasets. We attribute this increase to the fact that InfoLOOB suffers less than InfoNCE from the "explaining away" problem. However, InfoLOOB is even more effective for higher mutual information, that is, a richer covariance structure. Hopfield networks amplify the covariance structure in their retrieved embeddings. Though, this amplified covariance structure is disadvantageous for InfoNCE, as the saturation effect is stronger. The stronger saturation effect is caused by a richer covariance structure through Hopfield networks, which in turn leads to higher similarity between anchor and positive. Therefore, we see a performance drop when combining modern Hopfield networks with InfoNCE. Concluding, modern Hopfield networks are a perfect match for InfoLOOB as they yield higher mutual information. Therefore, CLOOB strongly improves the performance on 7 out of 8 zero-shot transfer learning tasks compared to CLIP.

For CLIP with InfoNCE, the hyperparameter $\tau^{-1}$ is a learnable parameter. For the other experiments, we use a fixed $\tau^{-1}$ of 30. The value for $\tau^{-1}$ was determined via hyperparameter search (see Section A.3.2).

In contrast to CLIP, we use a learning rate scheduler with restarts (Loshchilov & Hutter, 2017) to be more flexible regarding the number of total training epochs and enable training up to a plateau. To investigate the influence of the learning rate scheduler, we performed experiments with and without restarts. Table A2 shows the zero-shot performance for the different downstream tasks for CLIP and CLOOB respectively. For both CLIP and CLOOB, the performance at the majority of the tasks either increases or remains roughly the same with restarts.

Table A2: Influence of learning rate scheduler. For most of the tasks the performance either increases or remains roughly the same with restarts for both CLIP and CLOOB.

| Dataset | CLIP | | CLOOB | |
|---|---|---|---|---|
| | w/o restarts | w/ restarts | w/o restarts | w/ restarts |
| Birdsnap | **2.10** | 1.94 | **2.64** | 2.53 |
| Country211 | **0.71** | 0.62 | 0.63 | **0.76** |
| Flowers102 | 11.00 | **13.04** | 11.50 | **14.24** |
| GTSRB | 6.16 | **7.28** | 5.05 | **5.86** |
| UCF101 | 19.05 | **21.00** | 21.97 | **22.29** |
| Stanford Cars | **1.29** | 0.90 | 1.22 | **1.37** |
| ImageNet | 20.19 | **20.31** | 23.29 | **24.21** |
| ImageNet V2 | 20.53 | **20.63** | 22.97 | **23.80** |

Table A3: Datasets used for zero-shot and linear probing. In the case of several train or test sets per dataset we report the total number of samples. It should be noted that at the time of this work some Birdsnap images were not accessible anymore.

| Dataset | Classes | Train size | Test size | Evaluation metric |
|---|---|---|---|---|
| Birdsnap | 500 | 38,411 | 1,855 | accuracy |
| Country211 | 211 | 42,200 | 21,100 | accuracy |
| Flowers102 | 102 | 2,040 | 6,149 | class-weighted accuracy |
| GTSRB | 43 | 26,640 | 12,630 | accuracy |
| ImageNet | 1,000 | 1,281,167 | 50,000 | accuracy |
| ImageNet V2 | 1,000 | 1,281,167 | 30,000 | accuracy |
| Stanford Cars | 196 | 8,144 | 8,041 | accuracy |
| UCF101 | 101 | 28,747 | 11,213 | accuracy |

### A.3.2 HYPERPARAMETERS

The hyperparameter search was done on a validation split of CC with about 15,000 samples. For the hyperparameter $\tau^{-1}$ several values were considered (14.3, 30, 50, 70), where 30 leads to the best results for both YFCC and CC. Analogously to CLIP, we use the Adam optimizer (Kingma et al., 2014) with decoupled weight decay regularization (Loshchilov & Hutter, 2019). The weight decay is only applied to weights that are not gains or biases. As proposed in OpenCLIP (Ilharco et al., 2021) weight decay was set to 0.1. Different choices of weight decay (0.2 or 0.05), did not lead to a relevant performance change. We use the same learning rate of $1 \times 10^{-3}$ for CC and $5 \times 10^{-4}$ for YFCC as used in OpenCLIP. For the hyperparameter $\beta$ we considered values in the range of 5 to 20. A value of 8 resulted in the best performance for CC and 14.3 for YFCC. The batch size for CC was reduced to 512 due to computational restraints which did not result in performance losses. The batch size for YFCC was kept at 1024 as reported by OpenCLIP since a reduction resulted in a significant drop in performance. The learning rate scheduler for all experiments is cosine annealing with warmup and hard restarts (Loshchilov & Hutter, 2017) with a cycle length of 7 epochs. For models trained on YFCC the warmup was set to 10000 steps and for models trained on CC to 20000 steps.

### A.3.3 DATASETS

For pretraining we consider two datasets, Conceptual Captions (CC) (Sharma et al., 2018) and YFCC100M (Thomee et al., 2016). The **CC** dataset consists of 2.9 million images and corresponding high-quality captions. Images and their corresponding notations for CC have been gathered via an automated process from the web and therefore represent a wide variety of styles. Raw descriptions of images are collected from the *alt-text* HTML attribute. Both images and texts are filtered such that only image-text pairs above a certain quality threshold are part of this dataset. The dataset we refer to as **YFCC** is a subset of the Yahoo Flickr Creative Commons 100 Million (YFCC100M) dataset. It was created by filtering for images which contain natural language descriptions and/or titles in English resulting in 15 million image-caption pairs. The textual descriptions contain less

useful information than CC because they are not filtered by quality. Occasionally they also contain metadata like camera settings or web addresses.

We evaluate and compare our method on several downstream classification tasks. We evaluate on the same set of datasets as CLIP reported for a model trained on YFCC. This set contains Birdsnap (Berg et al., 2014), Country211 (Radford et al., 2021), Flowers102 (Nilsback & Zisserman, 2008), GTSRB (Stallkamp et al., 2011), UCF101 (Soomro et al., 2012), Stanford Cars (Krause et al., 2013) and ImageNet (Deng et al., 2009). Additionally, we include ImageNet V2 in our analysis (Recht et al., 2019). Table A3 shows an overview of training and test set sizes, number of classes and the applied evaluation metric. In the case of several test sets per dataset the metric is calculated for every set individually and the average performance is reported. The set size in Table A3 corresponds to the total number of samples across all test and training sets of a dataset respectively.

**Birdsnap** contains images of North American bird species, however our dataset is smaller than reported in CLIP as some samples are no longer available. The **Country211** dataset was published in CLIP and is a small subset of the YFCC100m dataset. It consists of photos that can be assigned to 211 countries via GPS coordinates. For each country 200 photos are sampled for the training set and 100 for testing. For the **Flowers102** images of 102 flower categories commonly occuring in the United Kingdom were collected. Several classes are very similar and there is a large variation in scale, pose and lighting. The German Traffic Sign Recognition Benchmark (**GTSRB**) was a challenge held at the IJCNN 2011. The dataset contains images of german traffic signs from more than 40 classes. Note that two versions of this dataset exist, one used for the challenge and an official dataset released after the competition. For CLIP the linear probing classifiers were trained using the competition training set but tested on the official test set. **Stanford Cars** contains images of 196 car models at the level of make, model and year (e.g. Tesla Model S Sedan 2012). **UCF101** (Soomro et al., 2012) is a video dataset with short clips for action recognition consisting of three training sets and three test sets. We follow the procedure reported in CLIP and extract the middle frame of every video to assemble the dataset. The **ImageNet** Large Scale Visual Recognition Challenge was held from 2012 through 2017 and is one of the most widely used benchmarks for object detection and localization. Several years later **ImageNet V2** assembled three new test sets with images from the same 1,000 classes to test for generalization of models optimized for the original ImageNet benchmark. Every test set comprises 10,000 samples.

### A.3.4 ZERO-SHOT EVALUATION

Class names for all downstream tasks were adopted from CLIP, that is, among other changes special characters like hyphens or apostrophes were removed. Furthermore, some class names of the datasets were slightly changed (e.g. "`kite`" to "`kite (bird of prey)`" in ImageNet). For zero-shot evaluation, we use the same prompt templates as published in CLIP. Depending on the dataset the number of prompts can vary from one prompt (e.g. "`a photo of a {label}, a type of bird.`" for Birdsnap) up to 80 prompts for ImageNet covering various settings (e.g. "`a cropped photo of a {label}.`", "`a origami {label}.`"). In case of several prompts an average embedding over all prompt embeddings is calculated. Figure A2 shows the zero-shot results for all evaluation tasks with the ResNet-50x4 model reported in Table 4.

### A.3.5 LINEAR PROBING

We try to follow the evaluation procedure in Radford et al. (2021) as closely as possible. We note one difference with respect to the implementation: Instead of scikit-learn's logistic regression using the L-BFGS solver, we use cuML's logistic regression classifier with L-BFGS algorithm to utilize GPUs for efficiency. All hyperparameters are the same as described in Radford et al. (2021), the maximum number of iterations was set to 1000, and the L2 regularization strength $\lambda$ was determined by using a parametric binary search.

We tried to reproduce the CLIP results with the correspondingly published models, however, failed to produce the exact numbers. This could be due to several factors:

- The train and validation split. Same as in Radford et al. (2021) , we use the provided validation set to perform the hyperparameter search. When there is none provided, we use a random half of the training dataset for validation.

Table A4: Linear probing results for the reimplementation of CLIP and CLOOB using different ResNet architectures trained on YFCC. The performance of CLOOB scales with increased encoder size

| Dataset | CLIP RN-50 | CLOOB RN-50 | CLOOB RN-101 | CLOOB RN-50x4 |
|---|---|---|---|---|
| Birdsnap | 50.9 | **56.2** | 58.1 | 62.2 |
| Country211 | 19.5 | **20.6** | 21.8 | 24.2 |
| Flowers102 | 94.8 | **96.1** | 96.1 | 96.2 |
| GTSRB | **82.5** | 78.9 | 77.9 | 80.6 |
| UCF101 | **75.2** | 72.3 | 72.8 | 75.3 |
| Stanford Cars | 36.2 | **37.7** | 39.0 | 44.3 |
| ImageNet | **66.9** | 65.7 | 67.0 | 69.7 |
| ImageNet V2 | **60.2** | 58.7 | 60.3 | 62.2 |

- In case of a tie in the validation score, we use the maximal $\lambda$ for the strongest regularization. We note though that we came closer to reproducing the results published in CLIP when using the mean $\lambda$ over all ties when these exist.

- For the Birdsnap dataset, the resources that we have got online at the time of this writing could be different from the resources that CLIP's authors obtained at the time.

Linear probing evaluation of YFCC-pretrained models is shown in Table A4. Comparing our reimplementation of CLIP and CLOOB with ResNet-50 encoders, we observe mixed results. The reason for this effect might be attributed to the observed task-dependence of multimodal models (Devillers et al., 2021). Another potential reason is that the benefit of the restrictions to more reliable patterns that occur in both modalities does not directly translate to an evaluation of just the encoding part of one modality. Again, as expected in self-supervised training, increasing the capacity of the CLOOB models benefits accuracy.

## A.4 REVIEW OF MODERN HOPFIELD NETWORKS

We briefly review continuous modern Hopfield networks that are used for deep learning architectures. They are continuous and differentiable, therefore they a work with gradient descent in deep architectures. They retrieve with one update only, therefore they can be activated like other deep learning layers. They have exponential storage capacity, therefore they can tackle large problems. Hopfield networks are energy-based, binary associative memories, which popularized artificial neural networks in the 1980s (Hopfield, 1982; 1984). Associative memory networks have been designed to store and retrieve samples. Their storage capacity can be considerably increased by polynomial terms in the energy function (Chen et al., 1986; Psaltis & Cheol, 1986; Baldi & Venkatesh, 1987; Gardner, 1987; Abbott & Arian, 1987; Horn & Usher, 1988; Caputo & Niemann, 2002; Krotov & Hopfield, 2016). In contrast to these binary memory networks, we use continuous associative memory networks with very high storage capacity. These modern Hopfield networks for deep learning architectures have an energy function with continuous states and can retrieve samples with only one update (Ramsauer et al., 2021; 2020). Modern Hopfield Networks have been successfully applied to immune repertoire classification (Widrich et al., 2020) and chemical reaction prediction (Seidl et al., 2021).

We assume a set of patterns $\{\boldsymbol{u}_1, \ldots, \boldsymbol{u}_N\} \subset \mathbb{R}^d$ that are stacked as columns to the matrix $\boldsymbol{U} = (\boldsymbol{u}_1, \ldots, \boldsymbol{u}_N)$ and a state pattern (query) $\boldsymbol{\xi} \in \mathbb{R}^d$ that represents the current state. The largest norm of a stored pattern is $M = \max_i \|\boldsymbol{u}_i\|$. Continuous modern Hopfield networks with state $\boldsymbol{\xi}$ have the energy

$$\mathrm{E} = -\beta^{-1} \log \left( \sum_{i=1}^{N} \exp(\beta \boldsymbol{u}_i^T \boldsymbol{\xi}) \right) + \beta^{-1} \log N + \frac{1}{2} \boldsymbol{\xi}^T \boldsymbol{\xi} + \frac{1}{2} M^2 . \tag{A170}$$

For energy E and state $\boldsymbol{\xi}$, the update rule

$$\boldsymbol{\xi}^{\text{new}} = f(\boldsymbol{\xi}; \boldsymbol{U}, \beta) = \boldsymbol{U} \, \boldsymbol{p} = \boldsymbol{U} \, \text{softmax}(\beta \boldsymbol{U}^T \boldsymbol{\xi}) \tag{A171}$$

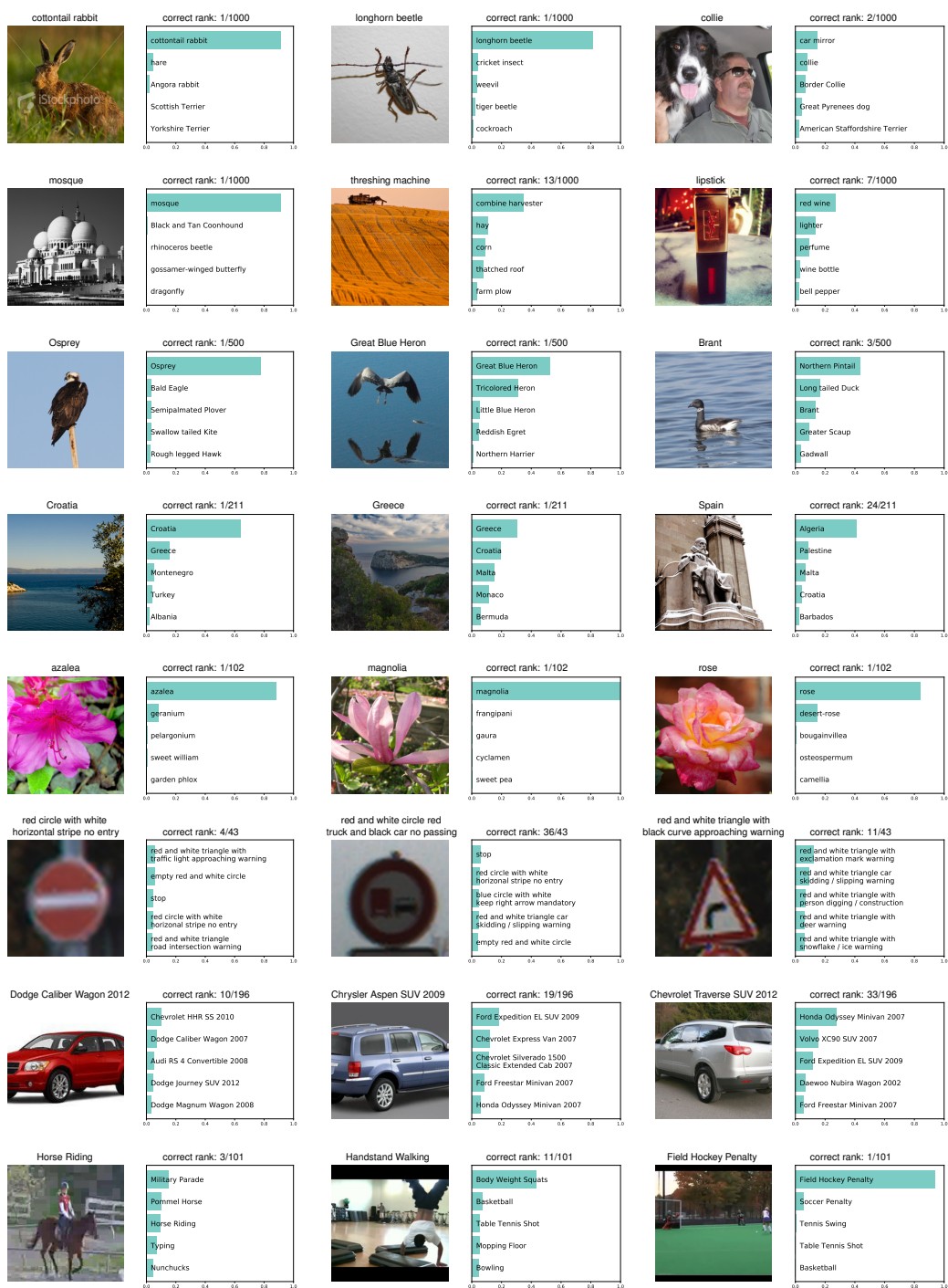

Figure A2: Visualization of zero-shot classification of three examples from each dataset. The following datasets are used (top to bottom): ImageNet, ImageNet V2, Birdsnap, Country211, Flowers102, GTSRB, Stanford Cars and UCF101. The ground truth label is displayed above the picture. The bar plots show the softmax values of the top 5 classes.

has been proven to converge globally to stationary points of the energy E, which are almost always local minima (Ramsauer et al., 2021). The update rule Eq. (A171) is also the formula of the well-known transformer attention mechanism (Ramsauer et al., 2021), therefore Hopfield retrieval and transformer attention coincide.

The *separation* $\Delta_i$ of a pattern $\boldsymbol{u}_i$ is defined as its minimal dot product difference to any of the other patterns: $\Delta_i = \min_{j,j\neq i} \left(\boldsymbol{u}_i^T \boldsymbol{u}_i - \boldsymbol{u}_i^T \boldsymbol{u}_j\right)$. A pattern is *well-separated* from the data if $\Delta_i \geq \frac{2}{\beta N} + \frac{1}{\beta} \log\left(2(N-1)N\beta M^2\right)$. If the patterns $\boldsymbol{u}_i$ are well separated, the iterate Eq. (A171) converges to a fixed point close to a stored pattern. If some patterns are similar to one another and, therefore, not well separated, the update rule Eq. (A171) converges to a fixed point close to the mean of the similar patterns. This fixed point is a *metastable state* of the energy function and averages over similar patterns.

The next theorem states that the update rule Eq. (A171) typically converges after one update if the patterns are well separated. Furthermore, it states that the retrieval error is exponentially small in the separation $\Delta_i$.

**Theorem A4** (Modern Hopfield Networks: Retrieval with One Update). *With query $\boldsymbol{\xi}$, after one update the distance of the new point $f(\boldsymbol{\xi})$ to the fixed point $\boldsymbol{u}_i^*$ is exponentially small in the separation $\Delta_i$. The precise bounds using the Jacobian $\mathrm{J} = \frac{\partial f(\boldsymbol{\xi})}{\partial \boldsymbol{\xi}}$ and its value $\mathrm{J}^m$ in the mean value theorem are:*

$$\|f(\boldsymbol{\xi}) - \boldsymbol{u}_i^*\| \;\leqslant\; \|\mathrm{J}^m\|_2 \, \|\boldsymbol{\xi} - \boldsymbol{u}_i^*\|\,, \tag{A172}$$

$$\|\mathrm{J}^m\|_2 \;\leqslant\; 2\,\beta\,N\,M^2\,(N-1)\exp(-\beta\,(\Delta_i - 2\,\max\{\|\boldsymbol{\xi} - \boldsymbol{u}_i\|, \|\boldsymbol{u}_i^* - \boldsymbol{u}_i\|\}\,M))\,. \tag{A173}$$

*For given $\epsilon$ and sufficient large $\Delta_i$, we have $\|f(\boldsymbol{\xi}) - \boldsymbol{u}_i^*\| < \epsilon$, that is, retrieval with one update. The retrieval error $\|f(\boldsymbol{\xi}) - \boldsymbol{u}_i\|$ of pattern $\boldsymbol{u}_i$ is bounded by*

$$\|f(\boldsymbol{\xi}) - \boldsymbol{u}_i\| \;\leqslant\; 2\,(N-1)\,\exp(-\beta\,(\Delta_i - 2\,\max\{\|\boldsymbol{\xi} - \boldsymbol{u}_i\|, \|\boldsymbol{u}_i^* - \boldsymbol{u}_i\|\}\,M))\,M\,. \tag{A174}$$

For a proof see (Ramsauer et al., 2021).

The main requirement of modern Hopfield networks to be suited for contrastive learning is that they can store and retrieve enough embeddings if the batch size is large. We want to store a potentially large set of embeddings. We first define what we mean by storing and retrieving patterns from a modern Hopfield network.

**Definition A1** (Pattern Stored and Retrieved). *We assume that around every pattern $\boldsymbol{u}_i$ a sphere $\mathrm{S}_i$ is given. We say $\boldsymbol{u}_i$ is stored if there is a single fixed point $\boldsymbol{u}_i^* \in \mathrm{S}_i$ to which all points $\boldsymbol{\xi} \in \mathrm{S}_i$ converge, and $\mathrm{S}_i \cap \mathrm{S}_j = \emptyset$ for $i \neq j$. We say $\boldsymbol{u}_i$ is retrieved for a given $\epsilon$ if iteration (update rule) Eq. (A171) gives a point $\tilde{\boldsymbol{x}}_i$ that is at least $\epsilon$-close to the single fixed point $\boldsymbol{u}_i^* \in \mathrm{S}_i$. The retrieval error is $\|\tilde{\boldsymbol{x}}_i - \boldsymbol{u}_i\|$.*

As with classical Hopfield networks, we consider patterns on the sphere, i.e. patterns with a fixed norm. For randomly chosen patterns, the number of patterns that can be stored is exponential in the dimension $d$ of the space of the patterns ($\boldsymbol{u}_i \in \mathbb{R}^d$).

**Theorem A5** (Modern Hopfield Networks: Exponential Storage Capacity). *We assume a failure probability $0 < p \leqslant 1$ and randomly chosen patterns on the sphere with radius $M := K\sqrt{d-1}$. We define $a := \frac{2}{d-1}(1 + \ln(2\beta K^2 p(d-1)))$, $b := \frac{2K^2\beta}{5}$, and $c := \frac{b}{W_0(\exp(a+\ln(b)))}$, where $W_0$ is the upper branch of the Lambert W function (Olver et al., 2010, (4.13)), and ensure $c \geq \left(\frac{2}{\sqrt{p}}\right)^{\frac{4}{d-1}}$. Then with probability $1-p$, the number of random patterns that can be stored is*

$$N \;\geq\; \sqrt{p}\,c^{\frac{d-1}{4}}\,. \tag{A175}$$

*Therefore it is proven for $c \geq 3.1546$ with $\beta = 1$, $K = 3$, $d = 20$ and $p = 0.001$ ($a + \ln(b) > 1.27$) and proven for $c \geq 1.3718$ with $\beta = 1$, $K = 1$, $d = 75$, and $p = 0.001$ ($a + \ln(b) < -0.94$).*

For a proof see (Ramsauer et al., 2021).

This theorem justifies to use continuous modern Hopfield networks for using retrieved embeddings instead of the original embeddings for large batch sizes. Even for hundreds of thousands of embeddings, the continuous modern Hopfield network is able to retrieve the embeddings if the dimension of the embeddings is large enough.

## A.5 FURTHER RELATED WORK

Multiple works have proposed improvements to InfoNCE. Joint Contrastive Learning (JCL) studies the effect of sampling multiple positives for each anchor. (Cai et al., 2020). Sampling negatives around each positive leads to higher bias but lower variance than InfoNCE (Wu et al., 2021). InfoNCE has been generalized to C-InfoNCE and WeaC-InfoNCE, which are conditional contrastive learning approaches to remove undesirable information in self-supervised representations (Tsai et al., 2021). ProtoNCE is a generalized version of the InfoNCE, which pushes representations to be closer to their assigned prototypes (Li et al., 2021). ProtoNCE combines contrastive learning with clustering. SimCSE employs InfoNCE for contrastive learning to learn sentence embeddings (Gao et al., 2021). InfoNCE has been extended to video representation learning (Han et al., 2020).

Many follow up works have been based on the CLIP model. The CLIP model is used in Vision-and-Language tasks (Shen et al., 2021). The CLIP model guided generative models via an additional training objective (Bau et al., 2021; Galatolo et al., 2021; Frans et al., 2021) and improved clustering of latent representations (Pakhomov et al., 2021). It is used in studies of out of distribution performance (Devillers et al., 2021; Milbich et al., 2021; Miller et al., 2021), of fine-tuning robustness (Wortsman et al., 2021), of zero-shot prompts (Zhou et al., 2021) and of adversarial attacks to uncurated datasets (Carlini & Terzis, 2021). It stirred discussions about more holistic evaluation schemes in computer vision (Agarwal et al., 2021). Multiple methods utilize the CLIP model in a straightforward way to perform text-to-video retrieval (Fang et al., 2021; Luo et al., 2021; Narasimhan et al., 2021).

