# OpenReview forum: "CLOOB: Modern Hopfield Networks with InfoLOOB Outperform CLIP"
_ICLR.cc/2022/Conference — ICLR 2022 Submitted_

### Official Review · Reviewer_jmHN · 2021-10-29

**Correctness:** 3
**Technical Novelty And Significance:** 2
**Empirical Novelty And Significance:** 3
**Recommendation:** 5
**Confidence:** 4

**Main Review:**

Strengths:

1. This paper is well written and easy to follow. There is almost no typo in the paper.

2. The proposed method is interesting. The introduction of Hopfield networks into contrastive learning to reduce the high variance of InfoLOOB is well motivated. In particular, CLOOB is easy to be implemented in practical applications.

3. The experiments on zero-shot datasets are well extensive and truly demonstrate the advantages of the proposed CLOOB when compared with the latest method CLIP. I appreciate the effectiveness of the proposed method in real-world applications.

4. The content of the appendix is substantial. The proof seems sound and correct, after I checked it.

Weaknesses:

1. The novelty of the proposed method seems not big enough. In my opinion, the proposed method is just to add modern Hopfield networks on top of the encoder to transform the original embedding to an informative one.  The total objective of the loss function in Eq. 13 is still the common-used  InfoLOOB loss. From this perspective, it seems like that the proposed method is just a feature-engineering algorithm, whose novelty, in my opinion, is not convincing enough. However, it is still glad to see that such simple algorithm is efficient in practical applications.

2. My major concern is the originality of the theoretical results of this paper. As far as I can tell, both Theorem 1 and Theorem 2 are the results, or can be directly obtained, from previous work. For example, the InfoNCE as the lower bound of mutual information (MI) is well-known result in the work of Poole2019ICML (cf Eq. 12 in “On Variational Bounds of Mutual Information” ), the InfoLOOB as the upper bound of MI can also be directly derived from Eq. 13 in Poole2019ICML. Therefore, in my opinion, there is no need to present the whole detailed proof of these theoretical results in Appendix, as such proof technique is not originated from the authors. Rigorously speaking, Theorem 1 and Theorem 2 in the main paper need citations as they are just the theoretical results from previous works. Authors of CLOOB could not use such theoretical results as their own.

3.  In view of Weakness 2, the theoretical contribution of this paper is limited. In my opinion, authors should give more theoretical analysis of their proposed CLOOB method, such as whether the introduction of Hopfield networks can theoretically reduce the variance of the learned model, instead of presenting the well-known lower and upper bounds on MI as their main theoretical results.

**Summary Of The Paper:**

This paper proposed a new contrastive learning method called CLOOB, which minimized the leave-one-out upper bound (InfoLOOB) on mutual information with the modern Hopfield networks. Concretely, Hopfield networks replace the original embeddings by retrieved embeddings in the InfoLOOB objective.  The retrieved embeddings are more robust as they capture the common covariance structure of all sample embeddings, leading to a stable performance of InfoLOOB. Extensive experiments on several zero-shot datasets show that the proposed CLOOB method outperforms the well-known CLIP model across all considered architectures.

**Summary Of The Review:**

I appreciate the effectiveness of the proposed method. However, considering the limited theoretical contribution of this work, and insufficient novelty of the proposed method, I could only give a borderline score (incline to rejection) for this paper.

---

> ### Author Response · Authors · 2021-11-23
> **Response to Reviewer jmHN (Part 2)**
>
> **“Originality of the theoretical results of this paper” (Theorem 1):**
> The reviewer is right that Poole et al. (2019) already provided a proof for Theorem 1. Therefore we have written: “For a proof see Poole et al. (2019) and the proof of Theorem A1 in the Appendix”. In the Appendix, we have written “Next, we present a theorem that shows this bound, where we largely follow Poole et al. (2019) in the proof. In contrast to Poole et al. (2019), we do not use the NWJ bound Nguyen et al. (2010)”. We directly proof Theorem 1 without a detour over NWJ bound. Furthermore the proof is presented as one sequence of inequalities, where one immediately can see the used inequalities. Agreed, this contribution is minor. Since we cite Poole et al. (2019) and never claimed that it is our proof, we cannot follow the reviewer’s comment: “Rigorously speaking, Theorem 1 and Theorem 2 in the main paper need citations as they are just the theoretical results from previous works. Authors of CLOOB could not use such theoretical results as their own.”
>
> **“Originality of the theoretical results of this paper” (Theorem 2):**
> The bound for score function $f(x,y)$ in the theorem is novel. InfoLOOB is not an upper bound on the mutual information if a neural network is used as a scoring function. Therefore, we had to use the idea of under-sampling, which was not used in the cited papers. We even have to introduce a new distribution for under-sampling. The theorem states that InfoLOOB is an upper bound if positives are under-sampled, that is $\tilde p(x | y)$ draws the positives for $y$ with lower probability than $p(x)$. The idea of under-sampling is new for these proofs. The reviewer is right, InfoLOOB as an upper bound for probabilities, that is, for $f(x,y)=p(y|x)$ was already mentioned in Poole et al. (2019). However, Poole et al. (2019) does not use score functions for InfoLOOB, which are needed if using neural networks.
>
> **“Give more theoretical analysis of their proposed CLOOB method”:**
> Following the suggestion of the reviewer, we included new analysis, which justifies the maximization of the InfoLOOB bound for contrastive learning. Maximizing the InfoLOOB bound is not intuitive as it was introduced as an upper bound on the mutual information. Still maximizing the InfoLOOB bound leads to a good approximation of the mutual information, in particular for high mutual information.
>
> Our new analysis shows in particular:
> - that InfoLOOB with neural networks as scoring function is bounded from above.
> - that InfoLOOB with neural networks as scoring function has two terms which makes it differ from the mutual information.
>
> The first term is the Kullback-Leibler divergence between the variational $q(x|y)$ and the posterior $p(x|y)$ similar to InfoNCE. For $q(x|y)=p(x|y)$, we have $f(y|x)=p(y|x)$. The second term is governed by the difference between the mean $E(f(x,y))$ and the empirical mean $1/(N-1) \sum_i f(x,y)$. Hoeffding's inequality bounds this difference as we demonstrate in our additional analysis. Therefore, the second term is negligible. The KL term is dominant and the relevant term, therefore maximizing InfoLOOB leads to $f(y|x) \approx p(y|x)$.
>
> The new analysis is included in the appendix and its result mentioned in the main paper.

---

> > ### Comment · Reviewer_jmHN · 2021-11-26
> > **Further Comments on Authors' Rebuttal**
> >
> > I am a researcher in machine learning theory, so I have to admit that I have strict requirements for the quality of a theoretical academic paper, especially the novelty of the theoretical results and the applications of existing theoretical results. Therefore, I would still maintain my original views, and my detailed responses are as follows.
> >
> > Part 1
> > 1.  While modern Hopfield networks have been designed to store patterns and then retrieve them, CLOOB utilizes them in a completely novel way. \
> > A: Directly using existing neural network models to a new application scenario, in my opinion, is not  novel enough (but i am happy to see its effectiveness in contrastive learning). I feel sorry that I am unable to point out explicitly the limitations of applying the existing Hopfield networks to contrastive learning in the current submitted work, but there is a good example that applied the idea of GAN to the design of domain adaptation algorithm (i.e. “Conditional Adversarial Domain Adaptation” in NeurIPS 2018), whose novelty of using GAN model to the domain adaptation is big enough. I hope this may give motivations  to the authors.
> >
> > 2. InfoLOOB was never used as an objective for contrastive learning.\
> > A: Honestly speaking, applying existing generalization bounds as the objective for contrastive learning is not novel. In my opinion, only when the generalization bounds (or say the theoretical results) is derived ORIGINALLY by yourself, setting such bounds as the objective function for certain problems will make the work novel enough. For example, Germain et al. (ICML2009, “PAC-Bayesian Learning for Linear Classifiers”) used their derived PAC-Bayesian generalization bounds (e.g. Thm 2.1) to directly guide the learning of linear classifiers,  Bartlett et al. (NeurIPS 2017, “Spectrally-normalized margin bounds for neural networks”) applied their proposed specially-normalized bounds (e.g. Thm 1.1) to the training of deep neural network,  Kuzborskij et al. (ICML2018, “Data-Dependent Stability of Stochastic Gradient Descent”) set their original stability-based generalization bound (e.g. Thm 4 and Prop 1) as objective for transfer learning. All of the aforementioned published works originally proposed novel generalization bounds and set them as the minimization objective for certain problems. Therefore, the novelty of their works is big enough. In summary, using existing InfoLOOB for contrastive learning is not novel enough, even though it leads to effective performance.
> >
> > Part 2
> > 1. We directly prove Theorem 1 without a detour over NWJ bound. Furthermore the proof is presented as one sequence of inequalities, where one immediately can see the used inequalities. Agreed, this contribution is minor.\
> > A:  Thanks. I noticed that you have cited the work of Poole et al. (2019)  in your original version. My major concern is that the novelty of the theoretical contributions of this work is not big enough.  For Theorem1, as the authors also admitted that the contribution is minor, I believe that such theoretical result (i.e. Thm 1) should be exclusive out of the theoretical contributions of this paper. Therefore, from the point of the rigorously theoretical novelty of this paper, Theorem 1 can be ignored.
> >
> > 2. The bound for score function $f(x,y)$ in the theorem is novel. InfoLOOB is not an upper bound on the mutual information if a neural network is used as a scoring function.\
> > A: As I have claimed in my original reviews, the theoretical contributions of this work is not big enough (although there maybe exist slightly different theoretical results for specific settings). Even though the function $f(x,y)$  is the scoring function  instead of the probability for neural networks, the proof of Theorem 2 (i.e. Eq.A50) is just a simple calculation of mutual information (or KL-divergence).
> >
> > 3. Following the suggestion of the reviewer, we included new analysis, which justifies the maximization of the InfoLOOB bound for contrastive learning. Maximizing the InfoLOOB bound is not intuitive as it was introduced as an upper bound on the mutual information. \
> > A: I appreciate authors’ effort to add more theoretical results about their proposed models. Considering that the contribution of Theorem 1 is minor and the theoretical contribution of Theorem 2 is not so big, I suggest authors to present their new theoretical analysis as a Proposition in the main text since such theoretical analysis is their own original result. In the current version, I have to admit that it is still hard for me to find the added theoretical results in the main text.

---

> > > ### Author Response · Authors · 2021-11-26
> > > **Response to Reviewer jmHN**
> > >
> > > We thank the reviewer for their additional comments on our paper. Despite their focus on theoretical machine learning, we hope that the reviewer still appreciates our development of a new method and the related experimental work as they already acknowledged in the first review.
> > >
> > > **To clarify a possible misunderstanding, this paper was never meant as a theoretical paper.** This is why we stated our contributions, as we wrote:
> > > “Our contributions are
> > > - a)  we introduce a new contrastive learning method called CLOOB,
> > > -b)  we propose InfoLOOB as an objective for contrastive learning
> > > - c)  we propose to use modern Hopfield networks to increase stability and to reinforce correlations for contrastive learning,
> > > - d)  we show theoretical properties of the InfoLOOB objective and loss function.”
> > >
> > > The main contribution is the new method, followed by the use of InfoLOOB as an objective, the use of Hopfield networks to reinforce correlations, and then the theoretical properties of the objective.
> > > We did not mention showing theoretical properties of the InfoLOOB *bound* in our contributions. However, we mentioned our contribution to theoretical properties concerning the InfoLOOB *objective* and *loss* function.
> > >
> > > Thanks for your suggestion to integrate the findings of the new analysis in the main paper - we will follow your advice.

---

> ### Author Response · Authors · 2021-11-23
> **Response to Reviewer jmHN (Part 1)**
>
> Thank you for your comments which helped to improve the paper. We addressed all your comments and followed all your suggestions.
>
> **“The novelty of the proposed method seems not big enough” (modern Hopfield networks):**
> While modern Hopfield networks have been designed to store patterns and then retrieve them, CLOOB utilizes them in a completely novel way. However, modern Hopfield networks have never been used to extract correlations in the data for contrastive learning. In this paper, we introduce this novel idea for extraction of covariance structures from retrieved patterns that originate from co-occurrences of embedding features in the memory. To use modern Hopfield networks to enhance the covariance structure in the data is a novel approach and might enter other machine learning methods.
>
> **“The novelty of the proposed method seems not big enough” (InfoLOOB):**
> To the best of our knowledge, InfoLOOB was never used as an objective for contrastive learning. Therefore, we do not believe that it is a commonly used loss function. While Poole et al. (2019) described the InfoLOOB bound, it was never considered in this paper as an objective for contrastive learning. Cheng et al. (2020) reported that they tested it without success but did not report any results. On 13th October 2021 (after the ICLR submission deadline), the ArXiv paper “Decoupled Contrastive Learning” (https://arxiv.org/abs/2110.06848) appeared, which exclusively focuses on the new InfoLOOB objective for unimodal contrastive learning (image-to-image). We now cite this paper, whose main novelty is the InfoLOOB objective. To use the InfoLOOB objective was not obvious, since it is an upper bound on the mutual information when using probabilities. However, when using approximations with neural networks, InfoLOOB is no longer an upper bound on mutual information. This justifies maximizing the InfoLOOB objective when using neural networks as scoring functions. At the time of submission, InfoLOOB was a novel objective for contrastive learning.

---

### Official Review · Reviewer_vr8w · 2021-11-01

**Correctness:** 3
**Technical Novelty And Significance:** 3
**Empirical Novelty And Significance:** 3
**Recommendation:** 6
**Confidence:** 4

**Main Review:**

Strengths:

1) Writing: The paper is generally carefully written, and easy to follow and understand. The reviewer enjoys reading the paper, though did not check the theorems in detail.

2) Novelty: The proposed CLOOB method is novel. It combines the InfoLOOB objective with modern Hopfield network for the training of a CLIP-like model. This has not been investigated before. Since CLIP is a hot topic in the field, it will be interesting for the community to know how CLIP can be further enhanced.

3) Experiments: The authors have conducted comprehensive experiments on zero-shot transfer learning, in order to demonstrate the effectiveness of CLOOB when compared with CLIP.

Weaknesses:

1) Method:

a) Typically, people will try to maximize an MI lower bound, and minimize an MI upper bound. In this paper, the authors try to maximize an MI upper bound. More discussion and rationales behind this design will be appreciated.

b) Since the proposed method is general, besides CLIP, did the authors try to apply it to self-supervised visual representation learning, such as MoCo and SimCLR etc.

c) The authors use InfoLOOB as the objective. Since InfoLOOB is an upper bound, a natural question is that why not also try using CLUB (Cheng et al., ICML 2020) as the objective?

d) What exactly is a modern Hopfield network? Eqn. (9) and (10) tried to explain this, but they are not clear enough to me. Why they are better? What's the original application they are proposed for? Since this is a core part of the proposed method, a more detailed description is needed.

2) Experiments:

a) From results in Figure 1, it is not easy to see clearly variance of InfoLOOB is reduced via modern Hopfield network. Is there any quantitative measure on the variance?

b) It would be interesting to also test ViT backbones, instead of ResNet backbones.

c) The ablation study in Sec. 4.1 seems important, maybe the authors should consider move them back to the main text if space permits.

d) In Table 3, what are the performance of CLIP under stronger RN-101 and RN-50x4 backbones? It will be interesting to see how the performance gap changes while making the visual backbone stronger.


Typo:

Figure 2, caption, 4-th and 5-th row, the "image embedding U_{y_i}" should be "text embedding"?

**Summary Of The Paper:**

By using the InfoNCE loss for model training, CLIP has achieved great success. In this paper, the authors propose CLOOB, short for "Contrastive Leave One Out Boost", where modern Hopfield networks are used together with the InfoLOOB objective. InfoLOOB is a leave-one-out upper bound of mutual information, and modern Hopfield networks replace the original embeddings by retrieved embeddings. Results show that CLOOB outperforms CLIP at zero-shot transfer learning across multiple architectures and datasets.

**Summary Of The Review:**

In summary, this paper proposes a novel method for CLIP-like model training, and results demonstrate the effectiveness of the proposed method on zero-shot transfer learning. However, the reviewer also thinks that this paper can be made stronger (see details in my comments), therefore, the reviewer decides to give a Borderline Accept recommendation at this moment.

---

> ### Author Response · Authors · 2021-11-23
> **Response to Reviewer vr8w (Part 3)**
>
> **“Experiments a)”:**
> We agree, a quantitative measure was missing. We added values for the variance in the main paper and in the appendix.
>
> We now write:
> “Figure 2 shows that modern Hopfield networks reduce the variance of the model. The average variances are reduced from 0.67 to 0.33 for MI 10 and from 1.00 to 0.48 for MI 14 (more details in Appendix A.2).”
>
> **“Experiments b)”:**
> For CC and YFCC, CLIP models with ViT backbones perform far worse than CLIP models with the much smaller ResNet-50 backbones. First experiments indicate that CLOOB with ViT backbones performs about as poorly as CLIP with ViT backbones. It seems that these datasets are not large enough to perform a valid comparison of the methods. Our observations are in concordance with findings of others concerning the ViT requirement for sufficiently large datasets [1][2][3].
>
> - [1] A. Dosovitskiy et al. ‘An Image is Worth 16x16 Words: Transformers for Image Recognition at Scale’, ICLR, 2020.
> - [2] H. Touvron et al. ‘Training data-efficient image transformers & distillation through attention’, ICML, 2021.
> - [3] X. Chen et al. ‘An Empirical Study of Training Self-Supervised Vision Transformers’, ICCV, 2021.
>
> **“Experiments c)”:**
> We thank the reviewer for acknowledging the importance of the ablation study. We followed the suggestion of the reviewer. We moved the table comparing different loss functions and modern Hopfield networks back to the main text as new Table 2 after making more space by rewriting.
>
> We now elaborate more on this topic by writing in the main paper :
> “Table 2 shows that the InfoLOOB objective increases the performance of CLIP in the majority of the datasets. The reason is that InfoLOOB suffers less than InfoNCE from the “explaining away” problem. However, InfoLOOB is more effective for higher mutual information, that is, for a richer covariance structure. Hopfield networks amplify the covariance structure by retrieved embeddings. For InfoLOOB, however, this amplification is disadvantageous as the saturation effect is increased by higher similarity between anchor and positive. Thus, combining modern Hopfield networks with InfoNCE leads to a performance drop. Combining Hopfield and InfoLOOB into CLOOB strongly improves the performance on 7 out of 8 zero-shot transfer learning tasks”
>
> **“Experiments d)”:**
> We agree with the reviewer and did runs with these backbones. These experiments confirmed the effectiveness of CLOOB for large backbones. We consistently obtain superior results when compared to CLIP.
>
> **“Typo: Figure 2, caption” (Figure 1 in the new version):**
> Thanks a lot for pointing that out. The caption is indeed confusing as we did not explicitly address the modality where the query comes from. We improved the caption by terming
> - ${\bf U}_{{\bf x}_i}$ the image-retrieved image embedding,
> - ${\bf U}_{{\bf y}_i}$ the text-retrieved image embedding,
> - ${\bf V}_{{\bf x}_i}$ the image-retrieved text embedding and
> - ${\bf V}_{{\bf y}_i}$ the text-retrieved text embedding.
>
> With these newly defined terms it should be more clear that the modern Hopfield network ${\bf U}$ stores images. However, a text embedding ${\bf y}_i$ queries the modern Hopfield network ${\bf U}$ for images that fit to the text. The text query ${\bf y}_i$ finds image embeddings ${\bf x}_j$ that are similar to the text embedding ${\bf y}_i$.
>
> We now write in the paper in the caption of Figure 1 (old Figure 2):
> "The image embedding $x_i$ and the text embedding $y_i$ retrieve the embeddings $U_{x_i}$ and $U_{y_i}$,
> respectively, from a modern Hopfield network that stores image embeddings $U= (U_1,\ldots,U_M)$ (green boxes at the left block). The image-retrieved image embedding $U_{x_i}$ serves as anchor in order to contrast the positive text-retrieved image embedding $U_{y_i}$ with the negative text-retrieved image embedding $U_{y_j}$ for $j \ne i$.
> Analog, for the second modern Hopfield network that stores text embeddings $V = (V_1,\ldots,V_K)$ (green boxes at the right block)."

---

> > ### Comment · Reviewer_vr8w · 2021-11-29
> > **Thanks for the rebuttal**
> >
> > Thanks the authors for the detailed rebuttal. These comments addressed my concerns properly, and I appreciate the reviewers spending time on the additional experiments. Generally, I am still positive about the paper. Though the training objective and the modern Hopfield network are not new, their combination and use in the context of CLIP-like training is novel, and empirical results have demonstrated this. I am hesitate about whether to increase the score further, due to the concerns raised by other reviewers. So, for now, I will keep my score as 6.

---

> ### Author Response · Authors · 2021-11-23
> **Response to Reviewer vr8w (Part 2)**
>
> **“What exactly is a modern Hopfield network?”:**
> The reviewer is absolutely right. The explanation of modern Hopfield networks was very superficial to non-existing.
> We now write in the paper:
> “Hopfield networks are energy-based, binary associative memories, which popularized artificial neural networks in the 1980s (Hopfield,1982; 1984). Associative memory networks have been designed to store and retrieve samples. Their storage capacity can be considerably increased by polynomial terms in the energy function (Chen et al., 1986; Psaltis & Cheol, 1986; Baldi & Venkatesh, 1987; Gardner, 1987; Abbott & Arian, 1987; Horn & Usher, 1988; Caputo & Niemann, 2002; Krotov & Hopfield, 2016). In contrast to these binary memory networks, we use continuous associative memory networks with very high storage capacity. These modern Hopfield networks for deep learning architectures have an energy function with continuous states and can retrieve samples with only one update (Ramsauer et al., 2021; 2020). Modern Hopfield Networks have already been successfully applied to immune repertoire classification (Widrich et al., 2020) and chemical reaction prediction (Seidl et al., 2021)”
>
> We also added to the paper:
> “Modern Hopfield networks is a new concept for contrastive learning.
> In bioinformatics the covariance structure in a sequence is reinforced by first retrieving similar sequences from a database and then aligning them. Conserved regions are characterized by high local covariance in the alignment (Dickson & Gloor, 2012; Kreth & Fodor, 2014). Modern Hopfield networks detect high covariances of embedded features, which is conveyed by the retrieved sample that corresponds to an alignment.”
>
> Furthermore, we give a review of modern Hopfield networks in the Appendix A.4.

---

> ### Author Response · Authors · 2021-11-23
> **Response to Reviewer vr8w (Part 1)**
>
> Thank you for a very elaborate and profound review and suggestions that helped us to improve our paper a lot. We addressed all your comments and followed all your suggestions. Please let us expand a bit on your comments.
>
> **“Maximize an MI upper bound”:**
> The reviewer has a good insight and makes a valid and very relevant point. InfoLOOB with a neural network as a scoring function is not an upper bound on the mutual information. InfoLOOB is an upper bound with probabilities, that is, for $f(x,y)=p(y|x)$ (Theorem 2). InfoLOOB with a neural network as a scoring function can be made to an upper bound only if under-sampling positives (Theorem 2). Under-sampling means that $\tilde p(x | y)$ draws the positives for $y$ with lower probability than $p(x)$. The idea of under-sampling is completely new.
>
> **“More discussion and rationales behind this design will be appreciated”:**
> We follow the suggestion of the reviewer to elaborate more on the rationale behind the maximization of the InfoLOOB. Towards this end, we performed an additional analysis:
> - It shows that InfoLOOB with neural networks as scoring function is bounded from above.
> - It shows that InfoLOOB with neural networks as scoring function has two terms which makes it differ from the mutual information.
>
> The first term is the Kullback-Leibler divergence between the variational $q(x|y)$ and the posterior $p(x|y)$ similar to InfoNCE. For $q(x|y)=p(x|y)$, we have $f(y|x)=p(y|x)$. The second term is governed by the difference between the mean $E(f(x,y))$ and the empirical mean $1/(N-1) \sum_i f(x,y)$. Hoeffding's inequality bounds this difference as we demonstrate in our additional analysis. Therefore, the second term is negligible. The KL term is dominant and the relevant term, therefore maximizing InfoLOOB leads to $f(y|x) \approx p(y|x)$.
>
> The new analysis is included in the appendix and its result mentioned in the main paper.
>
> **“Application to self-supervised visual representation learning”:**
> The suggestion of the reviewer to extend CLOOB to image-image is a very good idea. In our future work we want to extend the work in the ArXiv paper “Decoupled Contrastive Learning” (https://arxiv.org/abs/2110.06848) by not only using the new InfoLOOB objective but also modern Hopfield networks, that is, using CLOOB. Our focus was on zero-shot transfer learning capabilities, which is not the aim of image-image contrastive learning.
>
> **“CLUB as the objective”:**
> Again a valid point. In Cheng et al. (2020) CLUB has been used for minimizing the mutual information as usual for an upper bound. However, learning fails when maximizing the CLUB objective. There is a collapse to a single mode both without and with modern Hopfield networks. After extensive experimentation, we were not able to solve this problem. While CLUB averages over the negatives, InfoLOOB uses a log-sum-exponential (LSE). The LSE avoids a collapse since the sample that is most similar to the anchor is strongly pushed away from the anchor and even away from the average. The LSE leads to a more uniform distribution of features on the hypersphere. Our observations are in agreement with the findings in “Understanding Contrastive Representation Learning through Alignment and Uniformity on the Hypersphere” (https://arxiv.org/abs/2005.10242).
>
>
> We now write in the paper:
> “Contrastive Log-ratio Upper Bound (CLUB), another upper bound on the mutual information, was only used for minimizing it (Cheng et al., 2020). Maximizing CLUB failed in experiments, because the embedding distribution was not uniform as known for similar objectives (Wang & Liu, 2021). Uniform embedding distributions are required for successful contrastive learning (Wang & Isola, 2020).”

---

### Official Review · Reviewer_BgtC · 2021-11-02

**Correctness:** 4
**Technical Novelty And Significance:** 2
**Empirical Novelty And Significance:** 3
**Recommendation:** 5
**Confidence:** 4

**Main Review:**

**Strengths**
 - The paper is well written, with clear motivation, careful algorithm derivation, and detailed experiments + ablations. The comparison between InfoLOOB and InfoNCE losses (Section 2) are carefully made and technical sound (myself didn't rigorously go through the proof, though)
 - Good empirical improvements over CLIP.

**Weaknesses**
 - Contribution / novelty is small: neither InfoLOOB nor Modern Hopfield Network is new
 - In addition to image-text contrastive learning, would love to see how it applies to other contrastive learning tasks (e.g. image-image)

**Summary Of The Paper:**

This paper presents a new contrastive learning algorithm that comes with two improvements: a) replacing InfoNCE loss with InfoLOOB loss, b) replacing normal deep neural network with Modern Hopfield Network. When applied to image<->text contrastive learning, the new algorithm shows good improvements over baseline (CLIP) on a number of zero-shot image classification benchmarks.

**Summary Of The Review:**

While empirical results are good, the overall contribution of the paper seems fall below the bar. It simply applies a combination of two existing techniques to image-text contrastive learning, and doesn't show how it broadly applies to other contrastive learning tasks.

---

> ### Author Response · Authors · 2021-11-23
> **Response to Reviewer BgtC**
>
> We thank you for your comments and your suggestion to expand our work which thereby improved the paper. We addressed all your comments and followed all your suggestions.
>
> **“Neither InfoLOOB nor Modern Hopfield Network is new”:**
> To the best of our knowledge, InfoLOOB was never used as an objective for contrastive learning. While Poole et al. (2019) described the InfoLOOB bound, it was never considered in this paper as an objective for contrastive learning. Cheng et al. (2020) reported that they tested it without success but did not report any results. On 13th October 2021 (after the ICLR submission deadline), the ArXiv paper “Decoupled Contrastive Learning” (https://arxiv.org/abs/2110.06848) appeared, which exclusively focuses on the new InfoLOOB objective for unimodal contrastive learning (image-to-image). We now cite this paper, whose main novelty is the InfoLOOB objective. To use the InfoLOOB loss was not obvious, since it is an upper bound on the mutual information when using probabilities. However, when using approximations with neural networks, InfoLOOB is no longer an upper bound on mutual information. This justifies maximizing the InfoLOOB objective when using neural networks as scoring functions. At the time of submission, InfoLOOB was a novel objective for contrastive learning.
>
> While modern Hopfield networks have been designed to store patterns and then retrieve them, CLOOB utilizes them in a completely novel way. However, modern Hopfield networks have never been used to extract correlations in the data for contrastive learning. In this paper, we introduce this novel idea for extraction of covariance structures from retrieved patterns that originate from co-occurrences of embedding features in the memory. To use modern Hopfield networks to enhance the covariance structure in the data is a novel approach and might enter other machine learning methods.
>
>
> **“Other contrastive learning tasks (e.g. image-image)”:**
> The suggestion of the reviewer to extend CLOOB to image-image is a very good idea. In our future work we want to extend the work in the ArXiv paper “Decoupled Contrastive Learning” (https://arxiv.org/abs/2110.06848) by not only using the new InfoLOOB objective but also modern Hopfield networks, that is, using CLOOB. Our focus was on zero-shot transfer learning capabilities, which is not the aim of image-image contrastive learning.

---

> > ### Comment · Reviewer_BgtC · 2021-11-29
> > **Comments on Authors' Rebuttal**
> >
> > I would like to thank the authors for providing additional context on the novelty aspect, including pointing to a very recent arxiv paper. However, I still feel the paper falls below the novelty bar of ICLR, and therefore I'm keeping my original rating. I would be ok if all other reviewers or AC thinks otherwise (on the novelty aspect).

---

### Official Review · Reviewer_z4BR · 2021-11-03

**Correctness:** 3
**Technical Novelty And Significance:** 2
**Empirical Novelty And Significance:** 3
**Recommendation:** 8
**Confidence:** 3

**Main Review:**

Positive points:
- The method is clear and the paper well written and structured.
- The proposed CLOOB approach outperforms the chosen baseline on most tasks, demonstrating a clear performance gain.
- Ablation study A1 does seem to demonstrate that the objective introduced has strong synergies with the Hopfield network approach. This justifies the usefulness of the different components.

Concerns:
- The ablation study seems to indicate that InfoLOOB does not work well using the CLIP framework. Could the authors expand on why they thing this is the case?
- While it is true that the authors design a learning algorithm that functions well against their chosen baseline, overall most of the components (the InfoLOOB, modern Hopfield networks, ...) are not intrinsically novel. This does not detract from the performance gains presented in the paper however.

**Summary Of The Paper:**

The paper introduces an unsupervised learning technique called: "Contrastive leave-one out boost", which relies on a contrastive learning objective called "InfoLOOB". The authors compare this objective in detail with the classical InfoNCE objective, and compare the performance of their method with CLIP on a set of zero-shot transfer learning tasks (pre-training occurs on two datasets, conceptual captions and YFCC).

**Summary Of The Review:**

The paper is well written and interesting. The authors combine different components to design CLOOB, a learning algorithm that shows performance gains compared to the chosen baseline. Despite this, the components used in the approach are not novel, they are the result of previous work, meaning that the technical contributions mainly consist in finding ways to adequately combine Hopfield networks and the InfoLOOB objective. I would still recommend acceptance as I do not feel lack of technical novelty alone should detract from the empirical merits of this paper.

---

> ### Author Response · Authors · 2021-11-23
> **Response to Reviewer z4BR**
>
> Thank you for your thorough review and helpful comments which were very useful to improve the paper. We address your points individually.
>
> **“InfoLOOB does not work well using the CLIP framework”:**
> The reviewer raises an important topic. In the new version we elaborate more on this topic. We moved the ablation study table into the main paper as new Table 2. We explain more clearly that the InfoLOOB objective increases the performance of CLIP in the majority of the datasets when describing the results in Table 2. We now also discuss in more detail why modern Hopfield networks are key for leveraging InfoLOOB.
>
> We now write in the new version in the main paper:
> “Table 2 shows that the InfoLOOB objective increases the performance of CLIP in the majority of the datasets. The reason is that InfoLOOB suffers less than InfoNCE from the “explaining away” problem. However, InfoLOOB is more effective for higher mutual information, that is, for a richer covariance structure. Hopfield networks amplify the covariance structure by retrieved embeddings. For InfoLOOB, however, this amplification is disadvantageous as the saturation effect is increased by higher similarity between anchor and positive. Thus, combining modern Hopfield networks with InfoNCE leads to a performance drop. Combining Hopfield and InfoLOOB into CLOOB strongly improves the performance on 7 out of 8 zero-shot transfer learning tasks”
>
> **“Components of CLOOB are not novel” (InfoLOOB):**
> To the best of our knowledge, InfoLOOB was never used as an objective for contrastive learning. While Poole et al. (2019) described the InfoLOOB bound, it was never considered in this paper as an objective for contrastive learning. Cheng et al. (2020) reported that they tested it without success but did not report any results. On 13th October 2021 (after the ICLR submission deadline), the ArXiv paper “Decoupled Contrastive Learning” (https://arxiv.org/abs/2110.06848) appeared, which exclusively focuses on the new InfoLOOB objective for unimodal contrastive learning (image-to-image). We now cite this paper, whose main novelty is the InfoLOOB objective. To use the InfoLOOB objective was not obvious, since it is an upper bound on the mutual information when using probabilities. However, when using approximations with neural networks, InfoLOOB is no longer an upper bound on mutual information. This justifies maximizing the InfoLOOB objective when using neural networks as scoring functions. At the time of submission, InfoLOOB was a novel objective for contrastive learning.
>
> **“Components of CLOOB are not novel” (modern Hopfield networks):**
> While modern Hopfield networks have been designed to store patterns and then retrieve them, CLOOB utilizes them in a completely novel way. However, modern Hopfield networks have never been used to extract correlations in the data for contrastive learning. In this paper, we introduce this novel idea for extraction of covariance structures from retrieved patterns that originate from co-occurrences of embedding features in the memory. To use modern Hopfield networks to enhance the covariance structure in the data is a novel approach and might enter other machine learning methods.

---

> > ### Comment · Reviewer_z4BR · 2021-12-06
> > **Response to author commnents**
> >
> > I would like to thank the authors for responding to my comments. I feel they have addressed my first question concerning the relationship between CLIP and InfoLOOB. On the other hand, despite their comments, I believe my concerns re. the novelty still stand, to some extent.
> > As I do not believe a paper that achieves good results and is interesting should be penalized for a perceived lack of novelty of each individual component, and because I appreciate the authors clarifying the first point I mentioned, I am raising my score to an accept rating.
> >
> > Edit: To clarify further, as this point has been raised by other reviewers. Novelty is good, but at the heart of machine learning is the idea of solving problems. My personal opinion is that the paper proposes an interesting solution to an important problem. If we were to always reject ideas that re-use past components we, as a field, would be missing out on quite a few advances. For this reason, I would recommend acceptance, despite the comments about novelty raised by other reviewers.

---

> ### Comment · Area_Chair_uynK · 2021-12-04
> **Please respond to the authors' rebuttal**
>
> Dear reviewer, though this is late in the review process, please still try to respond to the authors' rebuttal as soon as possible. Thanks!

---

> > ### Comment · Reviewer_z4BR · 2021-12-06
> > **Response to AC's comments.**
> >
> > Thank you for the message, I have responded to the author's comments. My apologies for the delayed response.

---

### Author Response · Authors · 2021-11-23
**General Response**

We thank all reviewers for their time and for their constructive feedback. We were pleased to see that the reviewers found the paper well written and appreciated the detailed and thorough experiments. We thank the reviewers for valuing the contribution via InfoLOOB and modern Hopfield networks and recognizing CLOOB’s effectiveness.

The feedback of reviewers helped a lot to improve our paper. We answered all questions and comments of the reviewers and changed the paper accordingly. Furthermore we followed all their suggestions in the new version of the paper.

We provide answers, clarifications, and improvements of the paper in the individual responses to the respective reviewers. Further, we uploaded a revision of our paper in OpenReview. The main changes and responses are given in the following:

**Concerning the novelty:**
- The InfoLOOB is a novel objective (not a novel bound) at the time of submission.
- The application of modern Hopfield networks to reinforce the covariance structure is novel. We are not aware of a similar approach.

**Concerning experiments:**
- We performed additional experiments with larger backbone architectures: These experiments confirmed the effectiveness of CLOOB for large backbones. We consistently obtain superior results when compared to CLIP.
- We performed experiments with ViT backbones. Both CLOOB and CLIP underperformed compared to the much smaller ResNet50, since CC and YFCC are not large enough to obtain proper performance of ViT architectures.

**Concerning theory:**
- Theorem 1 and its proof are known (and cited) but proved in a slightly different way.
- Theorem 2 with score function and under-sampling is completely new (for probabilities it is known)
- We included new analysis, which justifies the maximization of InfoLOOB for contrastive learning even if it is an upper bound on the mutual information.

**Concerning the structure of the paper:**
- We improved the review of modern Hopfield networks.
- We added text parts on modern Hopfield networks.
- We included a new section in the appendix dedicated to the justification of the InfoLOOB objective.

---

### Decision · Program_Chairs · 2022-01-20

**Decision:**

Reject

**Comment:**

Four experts reviewed the paper and provided mixed recommendations. All reviewers found the experimental results strong, but they have different views about the technical novelty. Three reviewers considered the technical novelty as a weakness of the paper, but Reviewer z4BR was less concerned about it than the other two. After AC carefully read the paper and the authors' responses, AC agreed with the reviewers that the combination of InfoLOOB and modern Hopfield networks, which were both existing works, is incremental despite the empirical results. Besides, AC agreed with Reviewer jmHN that the theoretical results are not significant enough and could be moved to Appendices. While the empirical results are strong, they could not answer how the trend would change with bigger models and bigger datasets. Hence, while the paper clearly has merit, the decision is not to recommend acceptance. The authors are encouraged to consider the reviewers' comments when revising the paper for submission elsewhere.